Resource

# Single-nucleus multi-omics of human stem cell-derived islets identifies deficiencies in lineage specification

Punn Augsornworawat [1,2], Nathaniel J. Hogrebe [1], Matthew Ishahak [1], Mason D. Schmidt [1], Erica Marquez[1], Marlie M. Maestas[1], Daniel A. Veronese-Paniagua[1], Sarah E. Gale[1], Julia R. Miller[1,2], Leonardo Velazco-Cruz[1] & Jeffrey R. Millman [1,2] ✉

Insulin-producing β cells created from human pluripotent stem cells have potential as a therapy for insulin-dependent diabetes, but human pluripotent stem cell-derived islets (SC-islets) still differ from their in vivo counterparts. To better understand the state of cell types within SC-islets and identify lineage specification deficiencies, we used single-nucleus multi-omic sequencing to analyse chromatin accessibility and transcriptional profiles of SC-islets and primary human islets. Here we provide an analysis that enabled the derivation of gene lists and activity for identifying each SC-islet cell type compared with primary islets. Within SC-islets, we found that the difference between β cells and awry enterochromaffin-like cells is a gradient of cell states rather than a stark difference in identity. Furthermore, transplantation of SC-islets in vivo improved cellular identities overtime, while long-term in vitro culture did not. Collectively, our results highlight the importance of chromatin and transcriptional landscapes during islet cell specification and maturation.

The development of methods to differentiate hPSCs into islet-like clusters has the potential to generate an unlimited number of insulin-producing stem cell-derived β (SC-β) cells for the treatment of insulin-dependent diabetes[1-3]. This process utilizes temporal combinations of small molecules and growth factors[4-6], microenvironmental cues[7] and other sorting or aggregation[4,8-14] approaches to drive cells through several intermediate progenitor cell types. The resulting SC-β cells possess many features of primary human β cells, including the expression of β-cell-specific markers, glucose-responsive insulin secretion and the ability to reverse severe diabetes in animal models[5,6,15-18]. As a result, these cells have the potential to provide a functional cure for human patients with type 1 diabetes. However, these stem cell-derived islets (SC-islets) are a heterogeneous tissue that also contains stem cell-derived α (SC-α) and δ (SC-δ) cells, as well as an endocrine cell type of intestinal identity denoted here as stem cell-derived enterochromaffin (SC-EC) cells[19]. The presence of this awry population suggests that there are inefficiencies in lineage specification during these directed differentiation protocols[9]. Correcting this aberrant signalling could enhance specification to a β-cell identity and further improve the function of SC-β cells.

The specific pattern of gene expression and chromatin state within a cell directs differentiation and maintains its final identity[13,14,20-23]. Thus, characterizing both the transcriptional and chromatin landscape of cells during this differentiation process can provide insight into the degree that a particular cell type resembles its in vivo analogue. Single-cell RNA sequencing has been used to characterize the transcriptional environment of SC-islets, demonstrating that SC-β cells express many, but not all, of the important genes found in human primary β cells[9,16,17,24,25]. Studies have also begun to investigate the chromatin state within pancreatic cell differentiations using bulk approaches[26-29],

[1]Division of Endocrinology, Metabolism and Lipid Research, Washington University School of Medicine, MSC 8127-057-08, St. Louis, MO, USA. [2]Department of Biomedical Engineering, Washington University in St. Louis, St. Louis, MO, USA. ✉e-mail: jmillman@wustl.edu

and recent work has demonstrated the advantages of studying primary human islets using single-cell assay for transposase-accessible chromatin (ATAC) sequencing[30–32]. ATAC sequencing can describe whether particular chromatin regions are in an open, accessible state that is ready to be transcribed or interacted with, providing valuable information about cell identity that is missing when only transcriptional data are investigated. This data can define, for instance, the chromatin accessibility of promoters for specific genes that are capable of being transcribed. Furthermore, it can describe the chromatin accessibility of the DNA sequence motifs where specific transcription factors bind to help activate the transcription of different genes. Importantly, modulating the chromatin states of particular genes and motifs that are mis-expressed in in vitro-derived cell types are likely to drive cells closer to the identity of their in vivo counterparts. However, a comprehensive index of the combined chromatin accessibility and transcriptional signatures of each in vitro-differentiated SC-islet cell type is currently lacking.

In this Resource, we have provided and analysed the transcriptional and chromatin landscapes of SC-islets at the single-cell level, identifying important genes and motifs for each cell type. Interestingly, SC-EC and SC-β cells formed a gradient of cell identities, with subpopulations exhibiting characteristics of the other. Primary human islets had a more defined chromatin state than SC-islets, the latter of which had open chromatin regions associated with other lineages. Transplantation into mice for 6 months closed many of these accessible chromatin regions and improved lineage-specific gene expression, while extended in vitro culture did not have the same effect. We identified and modulated chromatin regulators important for SC-β cell identity, highlighting the importance of the chromatin landscape. Our findings improve the characterization of cellular identities within SC-islets and provide a resource to guide the development of strategies to improve SC-β cell differentiation.

## Results

### Multi-omic SC-islet analysis improves cell identity resolution

We analysed transcriptional and chromatin features of cells produced through directed differentiation of hPSCs to pancreatic islets to define cell identity and investigate lineage specification deficiencies rigorously[7,33] (Extended Data Fig. 1a,b and Supplementary Table 1). These SC-islets were processed and sequenced using single-nucleus multi-omics, obtaining both gene expression (messenger RNA) and chromatin accessibility (ATAC) information for each cell (Fig. 1a and Supplementary Table 2). Analysis of gene expression and chromatin accessibility both individually and in combination enabled us to identify specific islet cell types, including SC-β cells (Fig. 1b,c, Extended Data Fig. 1c,d and Supplementary Tables 3 and 4; 29,526 cells from 3 independent differentiations). We were able to identify two subpopulations of SC-EC cells (denoted as SC-EC1 and SC-EC2) only when using the integrated analysis of both mRNA and ATAC data (Fig. 1b). SC-β, SC-α and SC-δ populations had strong chromatin peaks represented by their respective *INS*, *GCG* and *SST* gene regions, as expected (Fig. 1d). Unexpectedly, however, the *INS* gene region had open accessibility across all detected cell types, including non-endocrine cell types to some extent. In contrast, *GCG* and *SST* gene regions had relatively few chromatin peaks outside of SC-α and SC-δ cells, respectively.

We found distinct transcription factor-binding motif groups enriched in each specific cell type (Fig. 1e, Extended Data Fig. 1e and Supplementary Table 5). Within the endocrine population, SC-β cells had enrichment of motifs that correspond to known β-cell-associated transcription factors[9,25,34,35]. Interestingly, this included enrichment of the MAFA binding motif, but the expression of the transcription factor itself remained at very low levels in SC-β cells, as previously described[19,36]. Motifs that were specifically enriched in SC-EC cells include GATA6, CDX2 and LMX1A, which are known intestinal markers[37,38]. Interestingly, SC-β and SC-EC cells possessed shared enrichment

of certain motifs, such as PDX1, PAX4 and NKX6-1, that are important in human primary β cells[31] (Fig. 1e).

In an effort to better characterize these cell populations within SC-islets, we cross-referenced transcription factor DNA-binding motif chromatin accessibility and gene expression information to identify active transcription factors that have both high expression and that can access their target binding motifs to promote transcription of other genes (Supplementary Table 5). Here we highlight the top ten identified active transcription factors enriched in specific endocrine populations compared with the average of the other endocrine populations (Fig. 1f). Highly active transcription factors in SC-β cells include RFX1, PDX1 and PAX6, which are important for β-cell identity[31,39]. Other highly active transcription factors, such as MAFG, EBF1, ISX and PLAGL2, have not been highlighted previously, demonstrating the utility of using both mRNA and chromatin accessibility data for identifying cell types. In contrast, SC-EC cells have co-enrichment of LMX1B, LMX1A, MNX1 and GATA6. Notably, the SC-β and SC-EC populations both have high activities of NKX6-1 and PAX4, suggesting that they share common features essential for their identities. This similarity is further demonstrated with motif chromatin accessibility plots, where NKX6-1 and PDX1 were enriched in both SC-β and SC-EC populations (Fig. 1e,g). In contrast, binding motifs for PAX6 were enriched in SC-β cells compared with SC-EC cells while LMX1A was enriched in SC-EC cells compared with SC-β cells. Unexpectedly, we found a mismatch in RNA expression and motif accessibility across cell types for certain transcription factors, such as LMX1B and ISX (Fig. 1f,g). We also investigated transcription factors specifically implicated during endocrine cell development by RNA expression and motif accessibility (Extended Data Fig. 1h). Of note, we observed FEV being upregulated in the SC-EC population, HNF1A in all endocrine populations, and PDX1 expressed predominantly in SC-EC and SC-β cells by both RNA and chromatin motif accessibility. We also observed a mismatch of NEUROG3 in the final SC-β cell population, where there was no detectable RNA but enriched motif accessibility. Collectively, this multi-omic analysis has generated important lists of genes to better classify the cell types found within SC-islets, providing better resolution of cell identity and insights into the relative importance of different transcription factor activity within each cell type.

### SC-EC and SC-β cells show gradient, not distinct, identities

The serotonin-producing SC-EC cells comprise an awry cell population that arises during in vitro SC-islet differentiation protocols but that does not positively contribute to tissue function[9,40,41] (Fig. 1b). Detection of substantial amounts of serotonin occurred only at the end of this differentiation protocol (Extended Data Fig. 2a). While it is not known how serotonin affects in vitro differentiation to SC-β cells, a study has reported its involvement in regulating β-cell mass during pregnancy[42,43]. Although both enterochromaffin and β cells arise from a definitive endoderm precursor during in vivo development, enterochromaffin cells originate from an intestinal lineage, while islet cells differentiate from a distinct pancreatic origin. During in vitro differentiation of SC-islets, however, they appear to share a common progenitor lineage, and how this enterochromaffin cell population emerges is not well understood[9]. Interestingly, the combined mRNA/ATAC clustering analysis allowed us to resolve two distinct SC-EC populations that were adjacent to the SC-β cell population (Fig. 1a), suggesting that there may be multiple or a gradient of cell states between SC-EC and SC-β cells. Given the apparent similarity of SC-β and the two SC-EC cell populations in our multi-omic clustering (Fig. 1b), we performed a trajectory analysis to detect differences in both gene expression and chromatin accessibility between SC-β and SC-EC cells (Fig. 2a, Extended Data Fig. 2b and Supplementary Table 6). As expected, enterochromaffin cell identity genes were most highly expressed in the SC-EC side of the trajectory map and β-cell identity genes were most highly expressed towards the SC-β cell side of the trajectory map (Fig. 2b,c and Extended Data Fig. 2c). ATAC peaks around SC-β cell and SC-EC cell marker genes

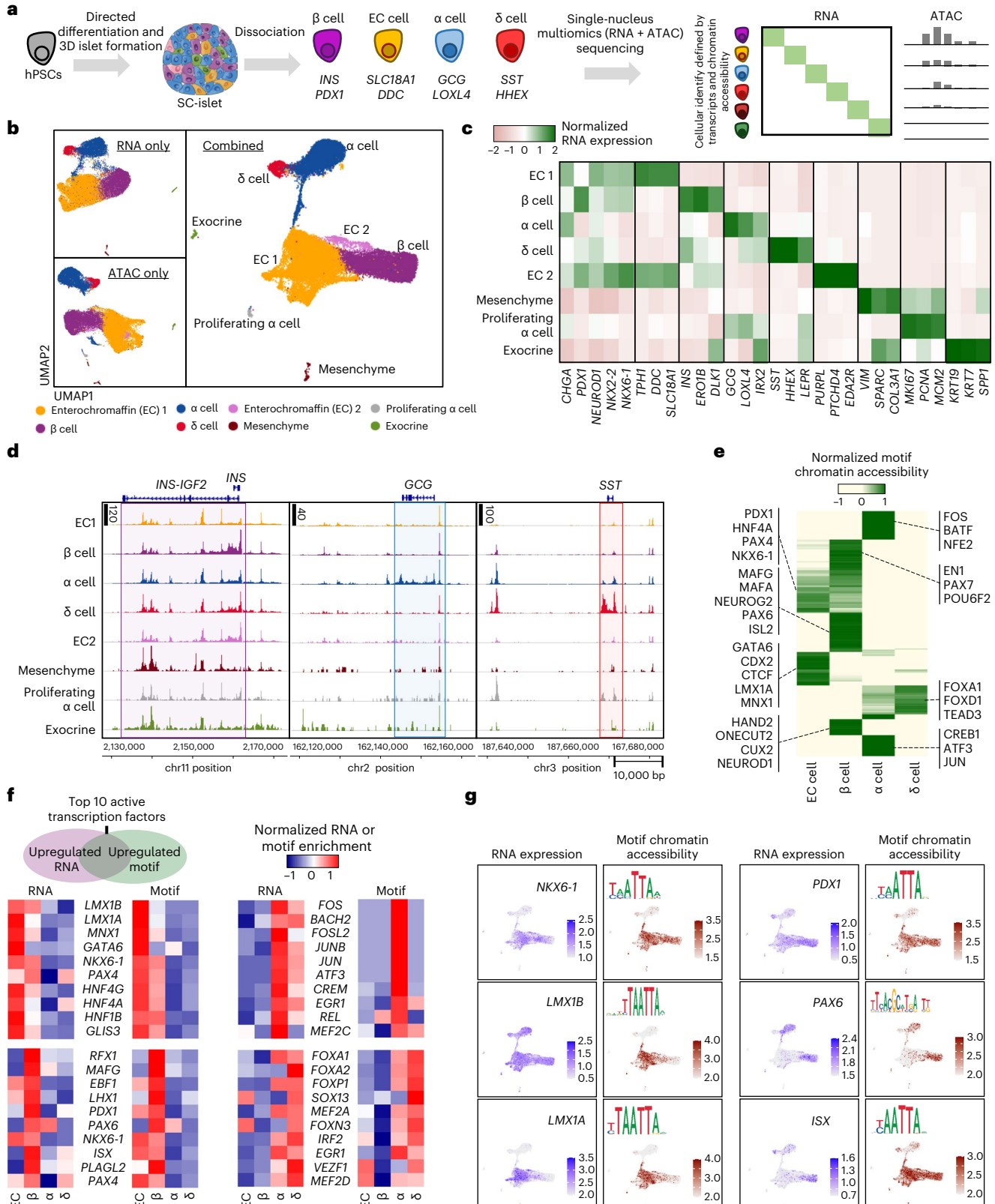

**Fig. 1 | Multi-omic profiling of SC-islets shows unique chromatin accessibility signatures in endocrine cell types. a**, Schematic of SC-islet differentiation and multi-omic sequencing. **b**, UMAPs showing identified cell types in SC-islets 2 weeks into stage 6 using both and either chromatin accessibility (ATAC) or gene (mRNA) information (29,526 cells from 3 independent differentiations; integration of all samples). **c**, Heat map showing gene expression of markers associated with each cell type. **d**, ATAC plots showing chromatin accessibility of SC-islet cell types around the *INS*, *GCG* and *SST* genomic regions. **e**, Heat map showing the top 200 variable DNA-binding motif accessibility within endocrine cell populations and highlighting markers for each cell type. **f**, Heat maps highlighting gene expression and ATAC motif accessibility of top ten active transcription factors co-enriched with both features in SC-β, SC-α, SC-δ and SC-EC cells. **g**, UMAP showing gene expression and motif accessibility of selected transcription factors associated with SC-β cells or SC-EC cells. SC, stem cell derived; EC, enterochromaffin.

also highlight differences in chromatin accessibility patterns, some of which were predicted to be *cis*-regulatory elements (Extended Data Fig. 2d). Differential motif chromatin accessibility analysis identified LMX1A, GATA6 and CTCF motifs to be enriched in SC-EC cells, while NEUROD1, HAND2, MAFA, PAX6, ONECUT2 and RFX2 motifs were enriched in SC-β cells (Fig. 2b,c).

While key identity genes and motifs were enriched on their respective sides of the trajectory map, their expression was a mixture of SC-β and SC-EC cell identities across the trajectory (Fig. 1b). To further probe the transcriptional and chromatin landscape of cells along this trajectory map, we performed subclustering analyses separately on the SC-β cell population and the SC-EC cell populations (Fig. 2d). Differential expression analyses revealed that each of the SC-β cell subpopulations had distinguishing gene expression and motif accessibility features (Fig. 2d,e, Extended Data Fig. 2e,f and Supplementary Table 7), including SC-β cells with *TPH1* expression and more open chromatin accessibility of the SC-EC cell-associated DNA-binding motifs for LMX1A and CTCF. The high *INS*-expressing SC-β cell population displayed greater motif accessibility to β-cell-associated transcription factors, such as PAX6, and less motif accessibility to EC-cell transcription factors, such as LMX1A. Similarly, differential expression analyses demonstrated that the four SC-EC cell subpopulations also had distinguishing gene expression and motif accessibility features, including SC-EC cells with elevated *INS* expression and more open chromatin accessibility of the SC-β cell-associated DNA-binding motifs for PAX6 and MAFG. Additionally, we performed trajectory analysis using another tool[44] that produced consistent results (Extended Data Fig. 2g). Collectively, these data suggest that the SC-EC and SC-β cell populations produced from in vitro differentiation form a continuum of cell states rather than exhibiting clear exclusivity of gene expression and chromatin accessibility, suggesting deficiencies in fully specifying each endocrine cell population in vitro.

Because changes in chromatin accessibility influence which genes can be expressed in a cell and consequently its identity, chromatin regulators could influence where a cell lies on this gradient between SC-β and SC-EC cells during in vitro differentiation. To this end, our integrated multi-omic analysis identified the chromatin remodeller CCCTC-binding factor (CTCF) as the transcription factor binding motif having the greatest increase in accessibility in SC-EC cells compared with SC-β cells (Fig. 2c). Because CTCF is a chromatin remodeller that is involved in other developmental and differentiation processes[45], we sought to further examine its impact on pancreatic differentiation. Unfortunately, we were unsuccessful in also knocking down its expression despite trying several approaches, which we would hypothesize could improve differentiation to SC-β cells. Instead, we utilized a doxycycline-inducible VP64-p65-Rta CRISPR activation (CRISPRa)

stem cell line[46] to increase transcription of *CTCF* during differentiation (Fig. 2f and Extended Data Figs. 2h,i and 3a). Upregulation of *CTCF* during stage 5 endocrine cell induction[7,33] resulted in drastic reductions in the expression of β-cell identity markers, insulin content and glucose-stimulated insulin secretion along with notable upregulation of intestinal lineage/EC-cell-associated genes (Fig. 2g,h and Extended Data Fig. 3a–e). *CTCF* overexpression had the most pronounced impact during the endocrine induction stage, indicating its specific role in selecting between SC-EC and SC-β cell fates (Extended Data Fig. 3f). Furthermore, single-nucleus multi-omic sequencing demonstrated that *CTCF* overexpression caused increased accessibility of the CTCF binding motif and decreased accessibility of β-cell-associated transcription factor binding motifs (Fig. 2i, Extended Data Fig. 3g–l and Supplementary Table 8; 12,467 cells from 2 datasets, 1 of each condition from the same differentiation batch). These results demonstrate that elevated *CTCF* expression disrupts the development of SC-β cells and redirects pancreatic progenitors towards an intestinal EC-like cell fate.

## Multi-omic analysis defines distinct primary islet cell types

To compare the transcriptional and chromatin signatures identified in our SC-islets to their in vivo counterparts, we sequenced and characterized primary human islets (Fig. 3a, Extended Data Fig. 4a,b and Supplementary Table 2; 30,202 cells from 4 separate donors). We identified ten distinct populations that included both pancreatic endocrine, exocrine and other minority cell populations (Fig. 3b,c, Extended Data Fig. 4c,d and Supplementary Tables 3 and 4). Unlike SC-islets (Fig. 1b,c and Extended Data Fig. 1c), clusters representing all cell types appeared very distinct and exhibited robust expression of their respective gene markers by gene expression and chromatin accessibility. In addition, the chromatin accessibility profiles demonstrated that primary islet endocrine cells contained distinct peak signals around the *INS*, *GCG*, *SST* and *PPY* genomic regions of β cells, α cells, δ cells and PP cells, respectively (Fig. 3d). We also performed chromatin motif accessibility analyses in primary islets and identified distinguishing features across these endocrine types (Fig. 3e, Extended Data Fig. 4e and Supplementary Table 5). Notably, primary β cells had accessible binding sites for PDX1, NKX6-1, MAFA and ISX, similar to SC-β cells. Surprisingly, motifs identified as enriched in SC-EC cells, such as LMX1A and MNX1, were enriched in β cells when compared with other endocrine populations. Furthermore, re-clustering analysis of the primary β cells identified three subpopulations that displayed unique gene expression and accessible motif signatures (Extended Data Fig. 4f–i and Supplementary Table 7), consistent with previous studies[25,31,47,48].

Similar to our multi-omic analysis of SC-islets, we also examined transcription factor activity in primary islet endocrine cells as assessed by both relative increases in mRNA transcripts and chromatin

**Fig. 2 | SC-EC and SC-β cells have unique and common transcriptional and chromatin accessibility signatures. a**, UMAP showing the trajectory of cells from the SC-β, SC-EC1 and SC-EC2 populations (subset of 21,317 cells from 3 independent differentiations; integration of all samples). **b**, Trajectory heat map showing dynamic changes of gene expression and motif accessibility enriched in β and EC groups. **c**, Volcano plots showing differential gene expression analysis (left) and differential motif accessibility analysis (right), highlighting relevant genes associated with SC-β and SC-EC cell populations. Statistical significance assessed by two-sided Wilcoxon rank sum test for RNA expression and two-sided logistic regression for motif chromatin accessibility. **d**, UMAP showing subpopulations by reclustering SC-β or SC-EC cell populations. Violin plots show gene marker expressions of *INS* and *TPH1*, highlighting off-target genes. **e**, Heat map showing DNA-binding motif accessibility associated with SC-β and SC-EC cells in subpopulations. Selected transcription factors plotted to show distribution of cells with target or off-target motif accessibility. **f**, Schematic of CRISPRa experiment for overexpression of *CTCF* in differentiating pancreatic progenitor cells. **g**, qPCR analysis of differentiated SC-islets with *CTCF* overexpression during endocrine induction, plotting mean ± s.e.m.

(*n* = 4 biologically independent samples), showing expression differences in doxycycline treated compared with untreated control (*INS*, P = 0.0010; *IAPP*, P = 1.5 × 10⁻⁷; *ISL1*, P = 3.0 × 10⁻⁵) and EC cells (*SLC18A1*, P = 1.6 × 10⁻⁴; *FEV*, P = 0.0019; *DDC*, P = 1.2 × 10⁻⁴; *TPH1*, P = 0.0047; *LMX1A*, P = 4.6 × 10⁻⁴). Control represents cells without doxycycline treatment. Statistical significance was assessed by unpaired two-sided *t*-test. **h**, ICC quantification of cells expressing C-peptide protein (P = 5.3 × 10⁻⁶) and SLC18A1 protein (P = 3.1 × 10⁻⁴) with or without *CTCF* overexpression, plotting mean ± s.e.m. (control; *n* = 6 biologically independent samples, doxycycline; *n* = 7 biologically independent samples). Control represents cells without doxycycline treatment. Statistical significance was assessed by unpaired two-sided *t*-test. **i**, Volcano plots showing differential motif chromatin accessibility analysis of SC-endocrine population comparing control and *CTCF* overexpression (12,467 cells from 2 independent biological samples, 1 of each condition from the same differentiation batch; integration of all samples). Statistical significance was assessed by logistic regression. Control represents cells without doxycycline treatment. SC, stem cell derived; EC, enterochromaffin cells; CRISPRa, CRISPR activation.

accessibility of the corresponding DNA-binding motif (Fig. 3f,g). For β cells, NKX6-1 was the only transcription factor that was shared in the top ten active factor list with SC-β cells (Figs. 1f and 3f). While some of the active transcription factors were shared with the list from SC-islets, this analysis illustrates that primary islet cells have a unique transcriptional and chromatin landscape when compared with their stem cell-derived counterparts, highlighting specific deficiencies in lineage specification during directed differentiation protocols.

## Primary islet chromatin is more restricted than in SC-islets
Previous studies have demonstrated that β cells derived from in vitro differentiation of hPSCs are functionally and transcriptionally different

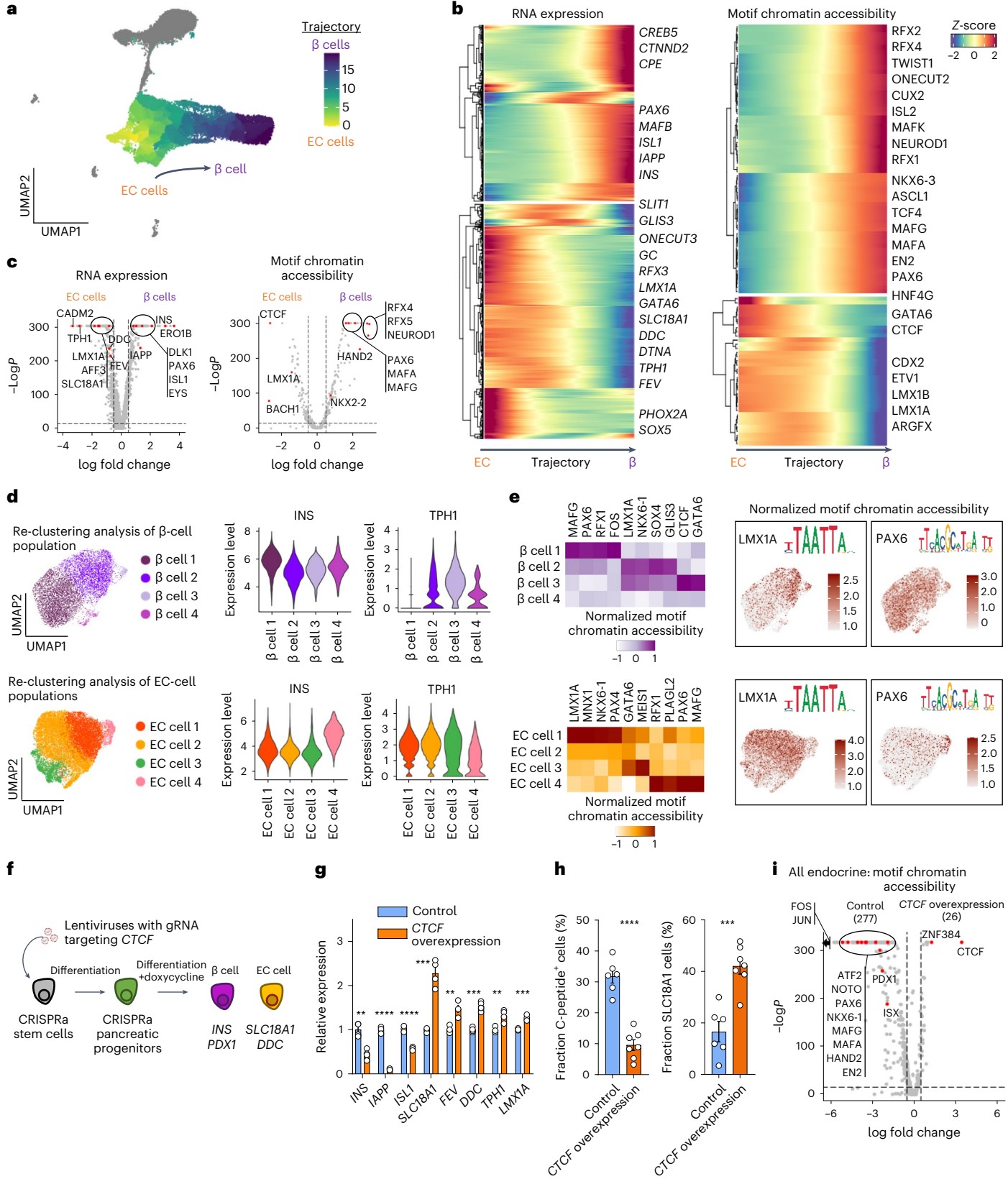

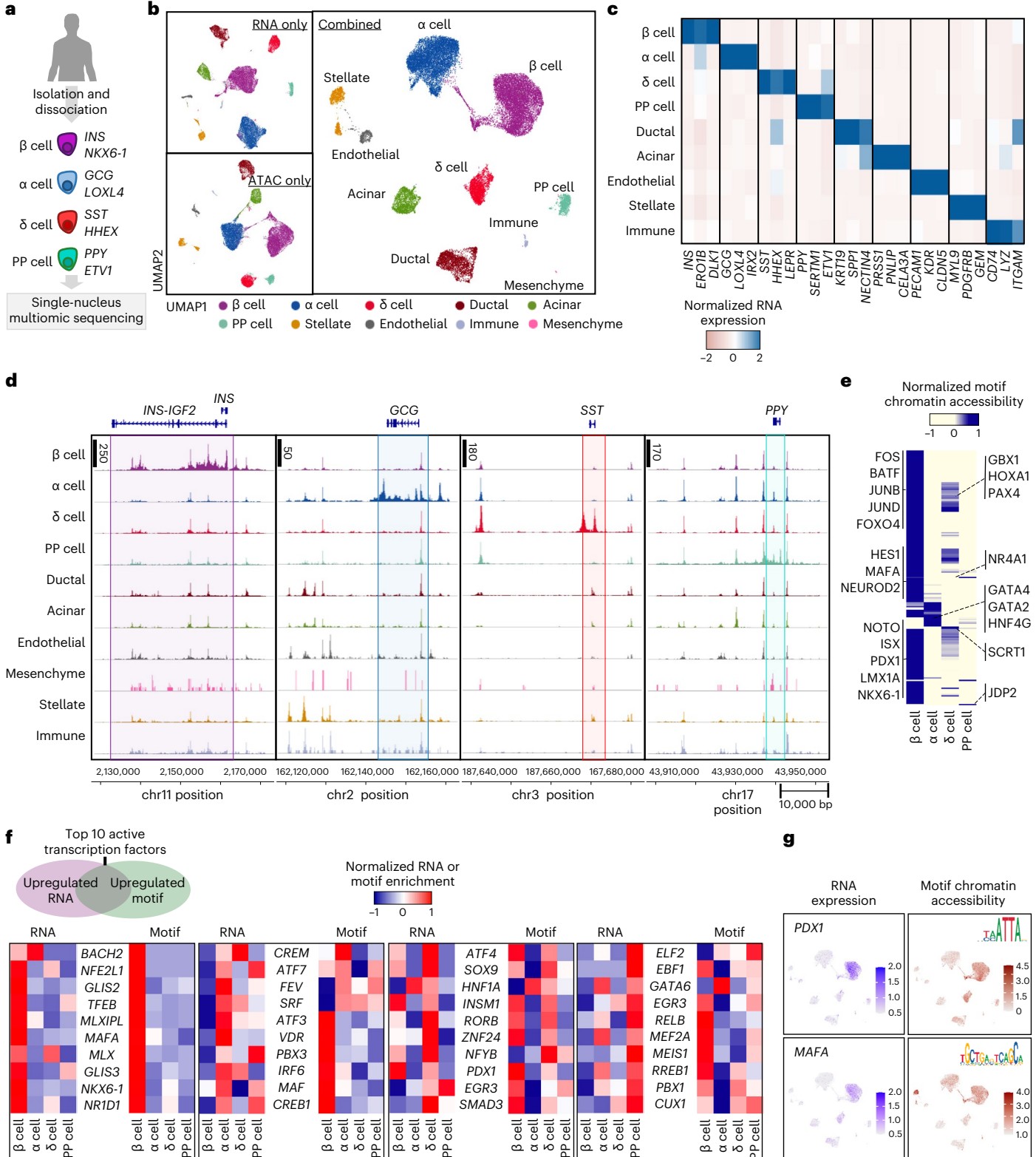

**Fig. 3 | Multi-omic profiling of human adult primary islets shows unique chromatin accessibility signatures in endocrine cell types. a**, Schematic of human adult primary islets for mutiomic sequencing. **b**, UMAPs showing identified cell types in primary human islets using both and either chromatin accessibility (ATAC) or gene (mRNA) information (30,202 cells from 4 independent biological donors; integration of all samples). **c**, Heat map showing gene expression of markers associated with each cell type. **d**, ATAC plots showing chromatin accessibility primary islet cell types around the *INS*, *GCG*, *SST* and *PPY* genomic regions. **e**, Heat map showing the top 200 variable DNA-binding motif accessibility within endocrine cell populations and highlighting motif markers for each cell type. **f**, Heat maps highlighting gene expression and ATAC motif accessibility of top ten active transcription factors co-enriched with both features in primary β, α, δ and PP cells. **g**, UMAP showing gene expression and motif accessibility of selected transcription factors associated with primary β cells. PP, pancreatic polypeptide.

than their in vivo counterparts[5,6,9,16,17,19]. We compared the transcriptional and chromatin landscapes of SC-islets and primary islets using single-nucleus multi-omic sequencing to better understand these differences and similarities (Fig. 4a; 47,566 cells from 5 samples; 3 SC-islets, 2 primary islets). Both SC-islets and primary islets contained β, α and δ cells (Extended Data Fig. 5a,b). However, only SC-islets contained EC-cells, while only primary islets contained PP cells, consistent with prior analysis[9]. In general, SC-islet cell types were transcriptionally most similar to their primary cell counterparts (Fig. 4b). Surprisingly, promoter chromatin accessibility did not show this trend, as cell origin (in vitro derived versus primary) was a greater factor in determining similarity rather than cell type. In general, promoter accessibility of SC-islet cells was open across cell types for endocrine and non-endocrine identity genes and lacked cell type distinctiveness (Fig. 4c,d), even when compared with additional primary islets (Extended Data Fig. 5c–e; 52,317 cells from 5 samples; 3 SC-islets, 2 primary islets). Primary β cells had greater chromatin accessibility of the *INS* gene compared with SC-β cells (Fig. 4e). While SC-α and SC-δ cells also shared chromatin accessibility peaks around the *INS* gene, this was not observed in primary α and δ cells. Chromatin accessibility analysis of *MAFA* and *UCN3* revealed predicted *cis*-regulatory elements that differed considerably between SC-β and primary β cells (Fig. 4f).

To explore similarities and differences in endocrine cell identity between SC-islets and primary islets, we analysed differential gene expression and motif chromatin accessibility in β-, α- and δ-cell subpopulations (Fig. 4g, Extended Data Fig. 5f and Supplementary Table 9). The mRNA data from the multi-omic analysis yielded results consistent with previous RNA-only studies, demonstrating SC-β cells had lower expression of *IAPP* and other maturation genes[9,16,25,49,50]. *ARID1B*, a chromatin regulator[51], had higher expression in SC-β cells. Surprisingly, *ONECUT2* had increased mRNA expression and DNA-binding motif chromatin accessibility in SC-β cells compared with primary β cells, despite this being a gene whose increased expression is associated with adult human β cells compared with juvenile β cells[34]. The chromatin accessibility data demonstrated that SC-β cells had more enriched motifs that are associated with off-target or progenitor cell states compared with primary β cells. Primary β cells showed enriched motifs linked to FOS/JUN family genes, expression of which is related to better function following transplantation[16]. MAFA surprisingly had similar chromatin accessibility of the associated DNA-binding motif between SC-β and primary β cells despite low mRNA expression and chromatin accessibility of the gene in SC-β cells (Fig. 4f,g). Thus, the role of MAFA in β-cell identity may be regulated by the chromatin state of the associated gene and not accessibility of its binding motif. These findings reveal substantial differences in mRNA expression and chromatin accessibility between SC-β cells and primary β cells. SC-β cells display more off-target cell type features and lack key adult primary β-cell identity signatures, whereas primary β cells have a more restricted chromatin state.

We compared β, α and δ cells from SC-islets and primary islets by analysing the activity of transcription factors in each cell type from both origins (Fig. 4h and Extended Data Fig. 5g). For SC-β and primary β cells, several transcription factors had comparable mRNA expression and motif accessibility, while others exhibited differences, such as enriched RFX1 motif accessibility, reduced *MAFA* RNA expression and reduced NFE2L1 and BACH2 motif accessibility in SC-β cells. We identified the top ten transcription factors with the greatest differential activity in SC-β cells compared with primary β cells and found that many of these factors were enriched in SC-β cells, indicating ambiguity in their cell identity due to incorrect or incomplete cell fate specification. Our analyses reveal fundamental differences in mRNA expression, chromatin accessibility and transcription factor DNA-binding motifs between cell types in SC-islets and their in vivo counterparts.

## In vivo, not in vitro, time enhances SC-β cell signatures

SC-β cells can acquire improved phenotypes with weeks in vitro or months after transplantation[4,9,17,18,27,52]. We performed multi-omic sequencing on SC-islets that underwent extended in vitro culture or transplantation into mice to characterize the chromatin and transcriptional changes that occur over time under these conditions. SC-islets cultured in vitro were sequenced periodically for up to 12 months (Fig. 5a and Extended Data Fig. 6a,b; 40,332 cells from 5 datasets, 1 from each timepoint). While SC-β cells displayed increased *INS* transcript with time, insulin secretion improved only until week 4 in vitro (Extended Data Fig. 6c,d). SC-α, SC-δ and SC-EC cells increased *GCG*, *SST* and *TPH1* expression, respectively, until week 4 but declined by month 6 in vitro (Extended Data Fig. 6c). We found reductions in both the number and identity features for SC-EC cells starting at month 6 and SC-α cell by month 12 (Extended Data Fig. 6e,f). Drastic shifts in chromatin accessibility were observed broadly across all endocrine cell types at months 6 and 12 (Extended Data Fig. 6g).

In SC-β cells, expression of many β-cell transcription factors and the accessibility of their associated motifs were upregulated at week 4 of in vitro culture but decreased in the long term (months 6 and 12) (Fig. 5b–d). Expression and promoter accessibility of other β-cell genes, including *UCN3*, increased, while those associated with off-target identities, such as SC-EC cell, diminished with long-term culture (Fig. 1b,d and Supplementary Table 10). Similarly, ATAC peaks demonstrated accessibility around the *INS* genomic region increased in SC-β cells but diminished in SC-α and SC-δ cells (Fig. 5c). Collectively, multi-omic analysis of in vitro cultured SC-islets revealed that, while the transcription of some important β-cell genes increased over time, the chromatin state of these cells became much more restricted in long-term culture. While this restriction helped to decrease off-target populations, many crucial β-cell transcription factors were also downregulated, possibly leading to the observed decrease in function.

In parallel, we also performed multi-omic sequencing of SC-islets transplanted for 6 months in mice (Fig. 5e and Extended Data Fig. 7a,b; 7,162 cells from 3 datasets, each dataset used 3 mice). Gene expression and promoter accessibility information demonstrated that endocrine identities became even more distinct than before transplantation (Extended Data Fig. 7c–e). Motif accessibility was also well defined for each cell type, with transplanted SC-β and SC-EC cells no longer displaying shared open motifs, such as for PDX1 (Extended Data Fig. 7f,g). Comparison of these 6 month transplanted SC-islets with those before transplantation revealed that transplanted SC-islets acquired gene expression and chromatin accessibility signatures that were more similar to their primary cell counterparts (Extended Data Fig. 8a–c and Supplementary Table 11). These improvements in cell identity were also reflected in the chromatin accessibility around the *INS* gene, which exhibited diminished peaks in non-β cell populations (Fig. 5f). No major shifts in composition were seen, suggesting this was set by the end of the in vitro differentiation process and that transdifferentiation was likely not occuring (Extended Data Fig. 8a). Gene sets and motif accessibility for many β-cell markers were upregulated in transplanted SC-β cells, whereas immature and off-target populations markers decreased (Fig. 5g and Extended Data Fig. 8d–f). Collectively, transplanted SC-β cells resembled primary β cells more in gene expression, promoter accessibility and motif accessibility, indicating that transplantation enhances both transcriptional and chromatin landscapes.

To compare 6 month in vitro and 6 month transplanted SC-islets, we integrated these datasets with the 2 week in vitro data (Fig. 6a and Extended Data Fig. 9a,b). Overall, transplanted SC-islet cells displayed greater enrichment of genes and motifs for transcription factors associated with their cell types, demonstrating that both time and environment are necessary for maturation (Extended Data Fig. 9c). In particular, transplanted SC-β cells exhibited several upregulated and

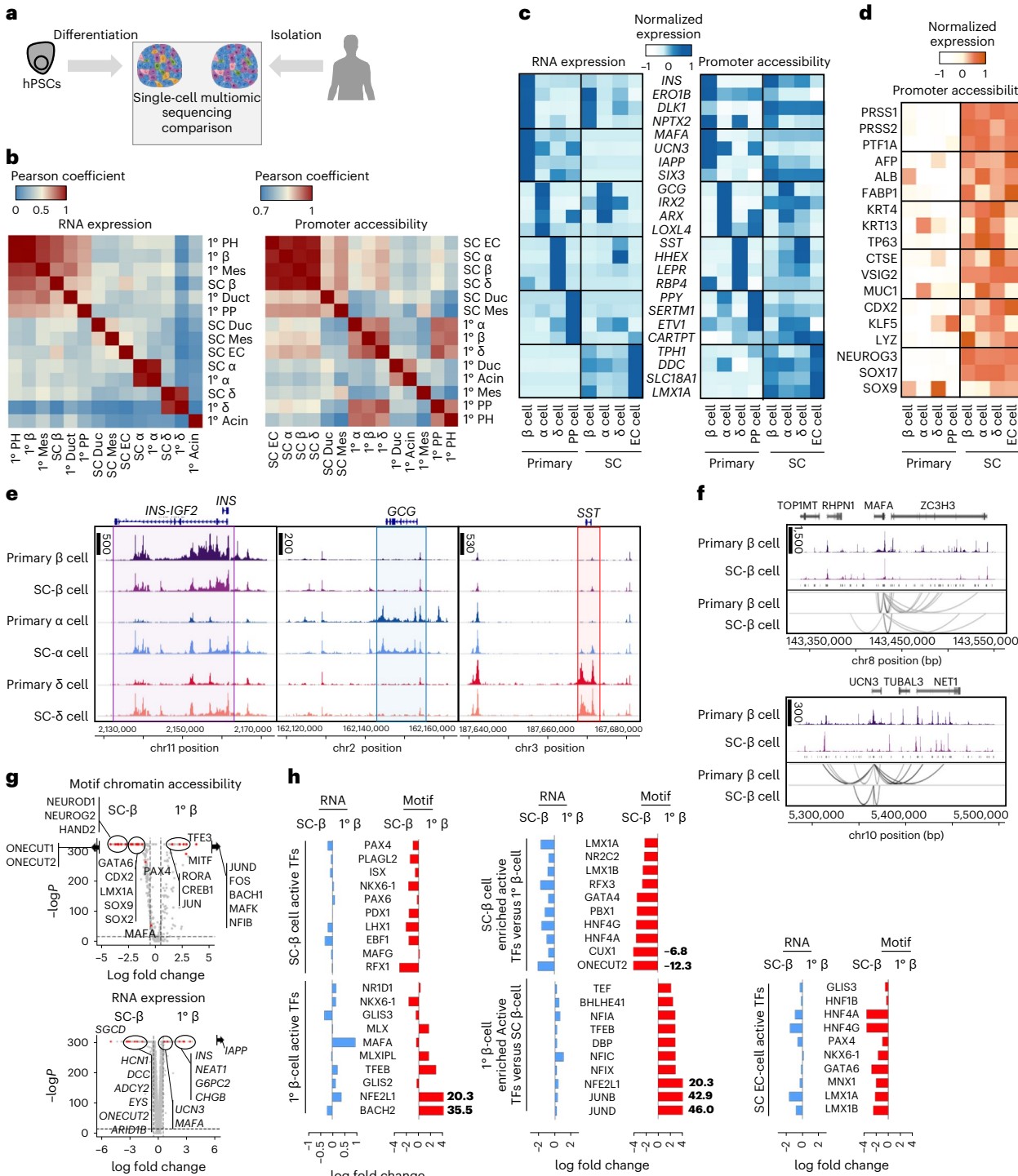

**Fig. 4 | Comparative analysis of SC-islets and primary human islets shows differences in chromatin accessibility signatures associated with islet identity. a**, Schematic showing comparisons of SC-islet and primary human islets (47,566 cells from 5 independent biological samples; all 3 SC-islets, 2 representative primary islets). **b**, Pearson correlation analysis comparing cell types in SC-islets and primary islets using gene expression and ATAC promoter accessibility. **C**, Heat map highlighting and comparing identity (β, adult β, α, δ, PP and EC) associated gene expression and ATAC promoter accessibility of endocrine cells in SC-islets and primary islets. **d**, Heat map highlighting and comparing off-target identity (exocrine, hepatic, oesophagus, stomach, intestinal and pancreatic progenitor) associated gene expression and ATAC promoter accessibility in SC-islet cells and primary islet cells. **e**, ATAC plots comparing chromatin accessibility around the *INS*, *GCG* and *SST* genomic regions in β, α and δ cells from SC-islets and primary islets. **f**, ATAC peaks from

SC-β cells and 1° β cells showing chromatin accessibility around β-cell identity marker, *MAFA* (top) and *UCN3* (bottom). Peaks were linked and analysed to depict differences in the number of *cis*-regulatory elements. **g**, Volcano plots showing differential motif accessibility analysis (top) and differential gene expression analysis (bottom) comparing SC-β cells and primary β cells. Statistical significance was assessed by two-sided Wilcoxon rank sum test for RNA expression and two-sided logistic regression for motif chromatin accessibility. **h**, Bar graphs showing fold change differences between SC-β cells and primary β cells, showing gene expression and motif accessibility of identified transcription factors (TFs) associated with SC-β cells, primary β cells and SC-EC cells. SC, stem cell derived; EC, enterochromaffin cells; PH, polyhormonal cells; Mes, mesenchyme; Duc, ductal; PP, pancreatic polypeptide cells; Acin, acinar cells.

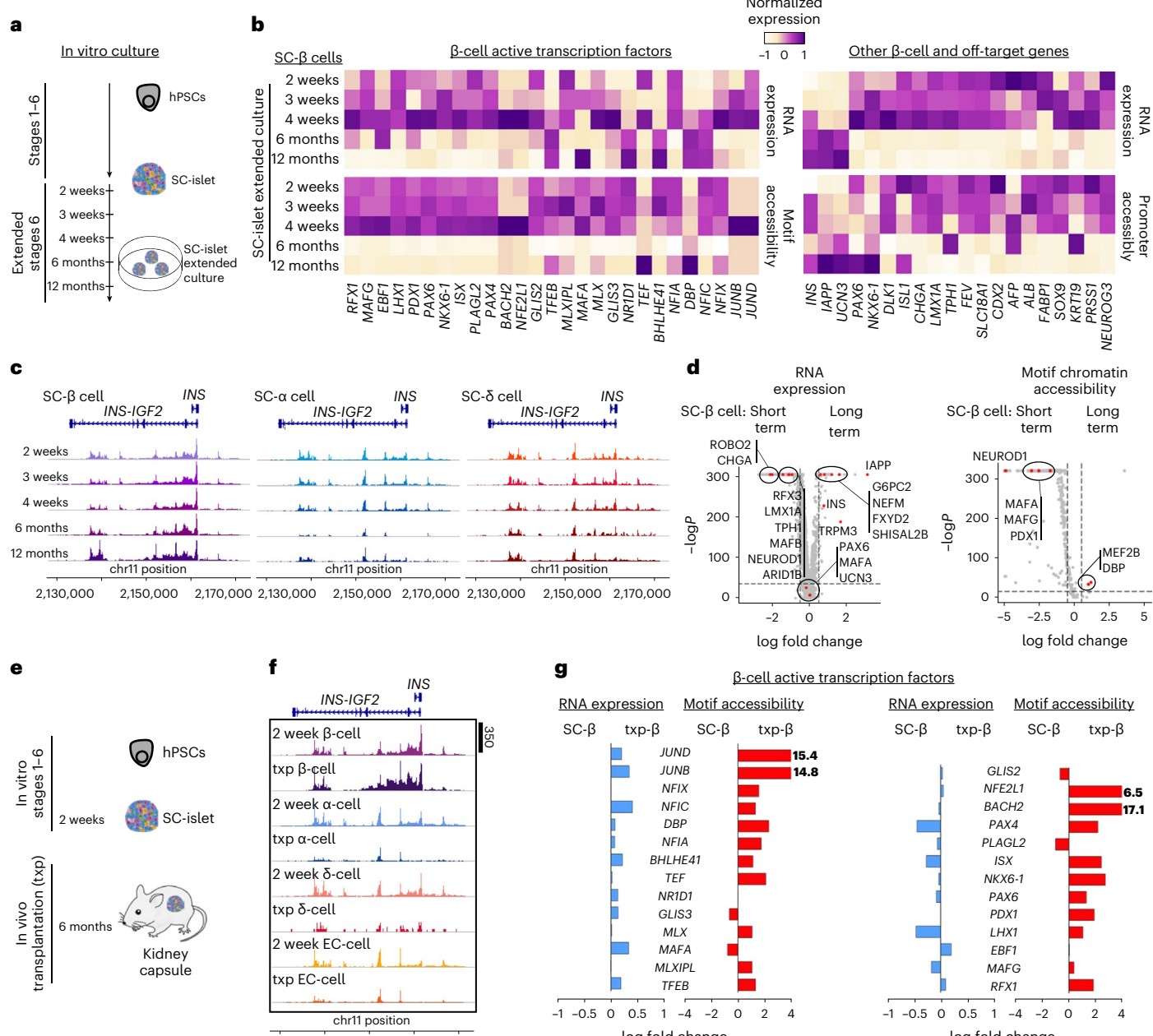

**Fig. 5 | Interrogating SC-islet identity with time in vitro and after transplantation. a**, Schematic of SC-islet in vitro culture with extended time course. **b**, Heat map showing gene expression, motif chromatin accessibility, or ATAC promoter accessibility of gene markers and transcription factors of SC-β cells cultured in vitro with extended time (2, 3 and 4 weeks and 6 and 12 months; 40,332 cells from 5 independent biological samples, 1 from each timepoint). **c**, Graphs comparing time course of chromatin accessibility around the *INS* genomic region in SC-β, α and δ cells. Peak signals around the *INS* gene in SC-β cell increases over time, but decreases in SC-α and δ cells. **d**, Volcano plots showing differential gene expression analysis (left) and differential motif accessibility analysis (right) comparing SC-β cells cultured short term (weeks 2, 3 and 4)

and SC-β cells cultured long term (months 6 and 12). Statistical significance was assessed by two-sided Wilcoxon rank sum test for RNA expression and two-sided logistic regression for motif chromatin accessibility. **e**, Schematic of differentiated SC-islets maintained long-term by in vivo transplantation. **f**, ATAC chromatin accessibility around the *INS* genomic region showing increase of peak signals in SC-β cells after transplantation and decrease in SC-α and δ cells (36,688 cells from 6 independent biological samples; 3 in vitro SC-islets; 3 in vivo SC-islets; integration of all samples). **g**, Bar graphs showing fold change differences of β-cell-associated active transcription factors in SC-β cells and transplanted SC-β cells. SC, stem cell derived; Txp, transplant; EC, enterochromaffin cells.

downregulated genes and motifs compared with their 6 month in vitro counterparts, allowing us to compile a list of markers relevant for β-cell identity from both environments (Fig. 6b,c, Extended Data Fig. 9d and Supplementary Table 12). In vitro culture closed many chromatin regions, including some associated with β-cell identity. In vivo time opened chromatin regions linked with cell identity while restricting others, enabling transplanted SC-β cells to acquire a chromatin and

transcriptional landscape closer to primary β cells. Furthermore, SC-EC cells also retained expression of identity genes when transplanted, in contrast to during 6 month in vitro culture.

Throughout our analysis, the chromatin regulator *ARID1B* emerged several times as being differentially expressed in β cells (Figs. 4g and 5d and Extended Data Figs. 8c and 10a). Knockdown of *ARID1B* during the in vitro culture of SC-islets increased β-cell identity markers, including

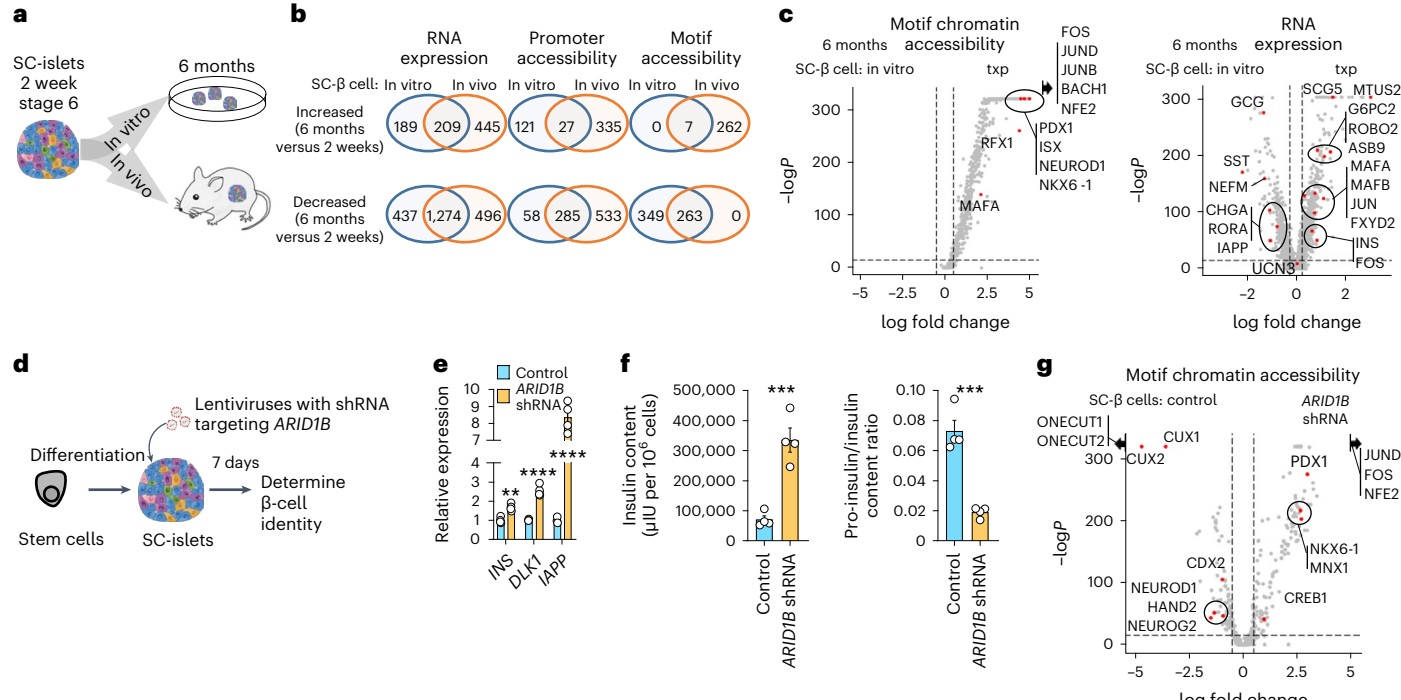

**Fig. 6 | Comparative analysis of 6 months in vitro and in vivo SC-β cells and improving in vitro maturation by *ARID1B* gene knockdown. a**, Schematic showing comparison of changes associated with 6 month in vitro culture and 6 month in vivo transplants. **b**, Number of genes, by gene expression or ATAC promoter accessibility, or motif chromatin accessibility, downregulated or upregulated in 6 months in vitro and in vivo SC-β cells. Number of features were determined by differential gene, promoter accessibility or motif accessibility analysis comparing 2 week SC-β cells with 6 month SC-β cells in vitro or 6 month SC-β cells in vivo (24,491 cells from 5 independent biological samples; week 2 representative SC-islets, month 6 SC-islets, and 3 samples of month 6 in vivo SC-islets). **c**, Volcano plots showing differential motif accessibility analysis (left) and differential gene expression analysis (right) comparing 6 months SC-β cells from in vitro and in vivo SC-islets. Statistical significance was assessed by two-sided Wilcoxon rank sum test for RNA expression and two-sided logistic

regression for motif chromatin accessibility. **d**, Schematic of SC-islets transfected with shRNA lentivirus for *ARID1B* gene knockdown. **e**, qPCR plots of SC-islets with ARID1B shRNA showing mean ± s.e.m. ($n = 4$ biologically independent samples) of expression of β-cell-associated genes, *INS* ($P = 0.0014$), *DLK1* ($P = 4.3 \times 10^{-5}$) and *IAPP* ($P = 3.0 \times 10^{-6}$). Statistical significance was assessed by unpaired two-sided *t*-test. **f**, Protein quantification plot showing mean ± s.e.m. ($n = 4$ biologically independent samples) of human insulin content ($P = 7.6 \times 10^{-4}$) by ELISA (left), and pro-insulin/insulin ratio ($P = 3.6 \times 10^{-4}$) by ELISA (right). Statistical significance was assessed by unpaired two-sided *t*-test. **g**, Volcano plot from single-nucleus multi-omics comparing motif chromatin accessibility of SC-β cells from control and *ARID1B* shRNA condition (21,969 cells from 2 independent biological samples, 1 of each condition from the same differentiation batch; integration of all samples). Statistical significance was assessed by two-sided logistic regression. SC, stem cell derived; Txp, transplant; EC, enterochromaffin cells.

gene expression and motif accessibility (Fig. 6d–g, Extended Data Fig. 10b–j and Supplementary Table 13; 21,969 cells from 2 datasets, 1 of each condition from the same differentiation batch). These findings demonstrate that the chromatin landscape is important for SC-β cell identity and that targeting chromatin regulators can further improve SC-β cell maturation in vitro.

## Discussion

SC-β cells have the potential to functionally cure type 1 diabetes[53] but do not perfectly match the transcriptional and functional features of primary β cells. An increased understanding of the deficiencies in lineage specification could improve SC-islet differentiation to prevent non-endocrine cell generation and increase SC-β cell function. Our single-nucleus multi-omic approach provided a more robust definition of cell types than using either data type in isolation and identified important genes and chromatin signatures for SC-islet cell type specification. Comparison with primary human islets allowed for identification of deficiencies in the chromatin and transcriptional landscape of SC-islet cell types. Although the analysis of only four islet donors may not encompass all possible variability across the general population, these data still offer important insights into the chromatin and transcriptional landscape of SC-islet cell types and helped identify differences from their primary counterparts. These differentially expressed

genes and chromatin accessibility signatures identified here can be targeted to improve SC-islet cell differentiation.

Our multi-omic analysis revealed important insights about the gene expression and chromatin state dynamics of SC-islet cell types compared with primary human islets. While SC-islet cell types were less distinct from each other by chromatin accessibility compared with primary cells, this difference was mainly driven by continued open chromatin accessibility of genes expressed by progenitor cell types, such as *NEUROG3* (ref. 54) and *GP2* (ref. 14), and alternative cell fates that were closed in primary cells. After transplantation into mice, SC-islet cell types developed more distinguished transcriptional and chromatin accessibility signatures that matched their respective cell identities[16,17]. However, extended in vitro culture broadly restricted access to chromatin regions, including those associated with a β-cell identity, and the underlying mechanism responsible for this difference in cellular identity remains unclear. These findings may have implications for other in vitro differentiation systems where immaturity is commonly observed[55,56].

Previous research suggested that SC-EC and SC-β cells are distinct populations that arise from the same pancreatic progenitor population during SC-islet differentiation[9]. However, our multi-omic study revealed that these cells form a continuum of cell types with varying degrees of both enterochromaffin and β-cell features, rather than

distinct cell populations. This suggests that current directed differentiation methodologies are insufficient for fully specifying each endocrine cell type. Furthermore, our data support the idea that chromatin accessibility is a major regulator of fate decisions between SC-EC and SC-β cells, particularly by CTCF. Interestingly, a recent publication[57] using a different single-cell approach suggested that SC-EC cells resemble a pre-β cell population in the pancreas. Transplanted SC-islets retained SC-EC cells after 6 months in vivo, but this population became less similar to β cells over time. As SC-EC cells may be detrimental[9] to SC-β cell function, understanding how to reduce or eliminate them is desired. Analysis tools that take account of both chromatin and mRNA[44] could lead to insights on cell fate decisions using our datasets.

SC-islets have been studied due to the success of primary islet transplantation in patients with type 1 diabetes[58–60]. Several areas of investigation are being pursued for which multi-omic assessment and the dataset presented here could be beneficial. Since the early SC-islet reports[5,6,61], groups have focused on changing medium composition or culture methods to improve SC-islet function[4,7,8,17,62,63]. Controlled assembly of three-dimensional SC-islet aggregates and circadian entrapment have also shown improvements[4,8,12,27]. Proliferation, purification and increased generation efficacy of progenitors have been pursued to improve final differentiation to SC-islets[10,14,64–69]. Genetic engineering to lessen immune recognition[70] and materials for immune protection have also been reported[63,71–80]. SC-islets have been generated from patients with diabetes via induced pluripotent stem cells (iPSCs) for diabetes pathology study and as an autologous cell source for β-cell replacement therapy[15,81–87]. A multi-omic approach could help determine how these modulations affect cell identity within SC-islets with greater resolution, facilitating finer control of differentiation parameters. Further multi-omic sequencing of iPSC-derived SC-islets can reveal the impact of iPSC derivation methods and starting somatic cell types on differentiation strategies.

Our study highlights the important role of chromatin regulators in the generation and final identity of islet cell types, as evidenced by the drastic differences in chromatin accessibility between cell types and within a given cell type from different origins. By modulating the chromatin regulators CTCF and ARID1B, we were able to alter SC-β cell identity, demonstrating the potential for better control of the chromatin landscape to improve SC-islet differentiation protocols. Our comprehensive indexing of single-cell transcriptional and chromatin accessibility states in SC-islets provides a valuable resource for further development of these protocols, as both aspects of islet cell identity will probably need to be targeted for enhanced differentiation strategies. Further comparisons with other multi-omic approaches[57] may yield additional insights into SC-islet identity and biology.

## Online content

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

## Methods

### Stem cell culture and differentiation

The HUES8 human embryonic stem cell (hESC) line was provided by Douglas Melton (Harvard University)[5]. The H1 hESC line was provided by Lindy Barrett (Broad Institute) with permission from WiCell containing doxycycline-inducible dCas9-VPR transgene in the AAVS1 locus (CRISPRa system)[46]. All hESC work was approved by the Washington University Embryonic Stem Cell Research Oversight Committee (approval no. 15-002) with appropriate conditions and consent. mTeSR1 (StemCell Technologies; 05850) was used for the culture of undifferentiated stem cells. All cell culture was maintained in a humidified incubator at 5% $CO_2$ and 37 °C. Cells were passaged every 4 days by washing cell with phosphate-buffered saline (PBS) and incubating with TrypLE at 0.2 ml cm$^{-2}$ (Gibco; 12-604-013) for 10 min or less at 37 °C. Dispersed cells were then mixed with an equal volume of mTeSR1 supplemented with 10 μM Y-27632 (Pepro Tech; 129382310MG). Cells were counted on Vi-Cell XR (Beckman Coulter) and spun at 300$g$ for 3 min at room temperature (RT). The supernatant was aspirated, and cells were seeded at a density of $0.8 \times 10^5$ cm$^{-2}$ for propagation onto Matrigel (Corning; 356230)-coated plates in mTeSR1 supplemented with 10 μM Y-27632. After 24 h, medium was replaced daily with mTeSR1 without Y-27632. SC-islet differentiation was performed as described previously[7,33]. Briefly, hESCs were seeded at a density of $6.3 \times 10^5$ cells cm$^{-2}$. Twenty-four hours later, the mTeSR1 was replaced with differentiation medium supplemented with small molecules and growth factors as outlined in Supplementary Table 1.

### SC-islet and primary islet cell culture

After 7 days in stage 6 of the differentiation protocol, cells were dispersed from the culture plate with TrypLE at 0.2 ml cm$^{-2}$ (Gibco; 12-604-013) for 10 min or less at 37 °C. The cells were mixed with an equal volume of stage 6 enriched serum-free medium (ESFM), centrifuged at 300$g$, and resuspended in ESFM at a concentration of 1 million cells ml$^{-1}$. Five millilitres of this solution were pipetted in each well of a six-well plate and placed on an orbital shaker (Orbi-Shaker $CO_2$, Benchmark Scientific) at 100 r.p.m. to form SC-islet clusters. These clusters were maintained by aspirating and replacing 3 ml of ESFM every 2–3 days. SC-islets in long-term culture were similarly maintained with ESFM for up to 1 year without passaging. Primary human islets were acquired as clusters and shipped from Prodo Laboratories, which required consent from the donor relatives for use in research. Consent information can be found on their website (https://prodolabs.com/human-islets-for-research). These islets have been refused for human islet transplants and meet specific criteria for research use. Donor details can be found in Supplementary Table 2. Our study consists of four donors. Upon arrival, islets were transferred into a six-well plate on an orbital shaker at 100 r.p.m. and maintained with 4 ml per well of CMRL1066 Supplemented medium (Corning; 99-603-CV) with 10% heat-inactivated foetal bovine serum (Gibco; 26140-079). Primary human islets were submitted for sequencing within 2 days after arrival.

### Mouse transplantations and SC-islet cell retrieval

Mice that were 7 weeks old, male, and with the NOD.Cg-*Prkdc*$^{scid}$ *Il*2*rg*$^{tm1wjl}$/SzJ (NSG) background (Jackson Laboratories; 005557) were randomly assigned to experimental groups. Mice were housed in an ambient facility with 30–70% humidity and a 12 h light/dark cycle and were fed a chow diet. Animal studies were performed by unblinded individuals in accordance with the Washington University International Care and Use Committee guidelines (approval 21-0240). Mice were anaesthetized using isoflurane and injected with ~5 × 10$^6$ SC-islet cells under the kidney capsule. At 6 months post-transplantation, mice were killed, and the kidney transplanted with SC-islets was removed. Sliced kidney samples were placed into a solution of 2 mg ml$^{-1}$ collagenase D (Sigma; 11088858001) in RPMI (Gibco; 1187-085). A total of nine mice were used to produce three samples. Each sample consisted of pooling three transplanted kidneys (one kidney from each mouse) to achieve sufficient cell numbers required for sequencing. The tissue was incubated for 40 min at 37 °C before diluting with PBS, mechanically disrupting with a pipette, and filtered through a 70 μm strainer (Corning; 431751). The flowthrough was centrifuged, and the remaining cell pellet was resuspended in MACS buffer (0.05% bovine serum albumin (BSA) in PBS). The Miltenyi mouse cell isolation kit (Miltenyi; 130-104-694; LS column, 130-042-401) was used to remove excess mouse cells. The flowthrough was centrifuged, and resuspended in PBS with 0.04% BSA for nuclei processing and sequencing. Data collection and analysis were not performed blind to the conditions of the experiments. No animals or data points associated with transplantation were excluded.

### Single-nucleus sample preparation and sequencing

Cells were processed and delivered to the McDonnell Genome Institute at Washington University for library preparation and sequencing. Samples were processed by dispersing cells into a single-cell suspension using TrypLE for 10 min at 37 °C and quantified for viability using the Vi-Cell XR (Beckman Coulter). Before proceeding, all samples were ensured to have >90% viability to minimize dead cell carry over in sequencing. Single-cell suspension samples were processed into nuclei according to the 10X Multiome ATAC + Gene Expression (GEX) protocol (CGOOO338). Cell samples were collected and washed with PBS (with 0.04% BSA), lysed with chilled Lysis Buffer for 4 min, washed three times with wash buffer and resuspended with 10x nuclei buffer at 3,000–5,000 nuclei μl$^{-1}$. Nucleus samples were processed using the Chromium 10x genomics instrument, with a target cell number of 7,000–1,0000. The 10x Single Cell Multiome ATAC + Gene Expression v1 kit was used according to the manufacturer's instructions for library preparations. Tape station figures for single-nucleus ATAC libraries can be found in Supplementary Fig. 1. Sequencing of the library was performed using the NovaSeq 6000 System (Illumina).

### Processing and filtering of multi-omic sequencing data

Multi-omic sequenced files were processed for demultiplexing and analysed using Cell Ranger ARC v2.0. Genes were mapped and referenced using human reference genome GRCh38. RStudio 1.3.1093 (R version 4.0.3) was used to perform analyses. Datasets were analysed using Seurat 4.01 (ref. 88) and Signac 1.3.0 (ref. 89). For ATAC data, peaks were called using MACS2 and the genomic positions were mapped and annotated with reference human genome EnsDb.Hsapeins.v86 and hg38. Dead cells were removed by excluding cells with very low RNA yields. (Nuclei samples do not contain valid cytoplasmatic mitochondrial content.) Low-quality cells including doublets, dead cells and poor sequencing depth cells were removed by filtering out cells with low RNA counts (nCount_RNA <1,000) and low ATAC counts (nCount_ATAC <1,000); high RNA counts (>40,000–50,000) and high ATAC counts (>40,000–50,000); nucleosome signal >1.25 and transcription start site (TSS) enrichment <2. For transplanted samples, excess mice cells from host mice were removed by filtering out cells expressing kidney marker *TTC36* (ref. 16), which is shared in both mouse and human. Highly expressed TTC36 cells (>0.0005 normalized counts) were excluded. Details on filtering can be found in Supplementary Table 2 and Supplementary Fig. 2.

### Dataset normalization, integration and assay build

Gene expression data were processed with SCT transform. ATAC data were processed with 'RunTFIDF' and 'RunSVD'. Integration of datasets were performed by anchoring using 'FindIntegrationAchors'. 'rpca' was used as a reduction method, with 'SCT' normalization to correct for batch differences for RNA expression. For integrated ATAC data, 'lsi' reduction was used along with 'RunTFIDF' and 'RunSVD' for batch correction. 'FindMultimodalNeighbors' with gene expression based 'pca' and ATAC based 'lsi' reductions were used to generate a joint neighbour graph. 'FindClusters' with SLM algorithm was used to identify clusters.

Promoter accessibilities were determined from ATAC data using 'Gene-Activity' (2,000 base pairs upstream of the transcription start sites[89,90]). ATAC peaks were called using MACS2 and linked using LinkPeaks and Cicero to determine *cis*-regulatory elements[89,91]. 'RunChromVAR' of chromVAR 1.12.0 package[92] and JASPAR version 2020 database[93] were used to compute motif enrichment. Motif sequence IDs were converted to transcription factor motifs using 'ConvertMotifID'. Promoter accessibility and motif enrichment information were incorporated as assay data in the integrated Seurat object files.

### Analysis of multi-omic datasets

Cell types in the integrated datasets were identified by performing differential expression analysis. 'FindMarkers' using the 'wilcox' test method was used to determine highly expressed feartures. 'LR' test method in 'FindMarkers' was used to determine upregulated motif accessibility. 'ConveragePlot' of the Signac package (for example, Figs. 1d and 4f) was used to generate ATAC peaks with *cis*-regulatory element information. Uniform manifold approximation and projection (UMAP) and violin plots were generated using 'FeaturePlot' and 'VlnPlot' (for example, Figs. 1g and 2d). Heat maps were generated using average values using 'AverageExpression' and 'heatmap.2' (for example, Fig. 1c). Top active transcription factors were assessed for top upregulated transcription factor gene expression and motif enrichment by computing the average fold change of selected populations or conditions compared to other cell populations. Shared upregulated transcription factors by both gene expression and motif enrichment assays were determined as active transcription factors (for example, Fig. 1f). Volcano plots were generated using 'EnhanvedVolcano' (for example, Fig. 2c). Gene set enrichment analysis was performed using 'DEenrichRPlot' of enrichR package with databases from Gene Ontology, KEGG_2021_Human and MSigDB_Hallmark_2020.

### Trajectory analysis

Trajectory analysis was performed using SeuratWrappers and the Monocle3 1.0 package[94]. 'Subset' was used to isolate specific populations (enterochromaffin cells and β cells) from the integrated dataset. A monocle compatible cds file was generated using 'as.cell_data_set'. Pseudotime was computed and determined using the 'cluster_cells' and 'learn_graph' functions. Trajectories were re-established by selecting the initial node pseudo-timepoint using 'order_cells'. Dynamic analysis of gene expression and motif enrichment were obtained by conducting differential expression feature analyses along the pseudotime trajectory. 'graph_test' was used with 'neighbor_graph' parameter set to 'principal_graph'. Genes or motifs of high differential expression along pseudotime were determined by excluding features with low Moran's *I* scores (morans_I >0.05). Expression values were *Z*-scored and plotted using 'Heatmap' with *K* means parameter ($k_m = 2$). Pseudotime information from this analysis was incorporated into the Seurat object using 'AddMetaData'.

MultiVelo[44] package running on Jupyter Notebook (Python) was used to perform alternative trajectory analysis. ATAC and RNA files were imported using 'sc.read_10x_mtx' and 'scv.read'. Aggregated peaks around gene regions were computed using 'mv.aggregate_peaks_10x', and mapped with RNA and ATAC information using 'pd.Index'. RNA counts and ATAC peaks were normalized using 'scv.pp.filter_and_normalize' and 'mv.tfidf_norm' respectively. Smoothing of gene peak aggregates was performed using 'mv.knn_smooth_chrom', followed with 'mv.recover_dynamics_chrom' to execute the multi-omic dynamic model to predict cell state trajectories. Pseudotime was computed using 'mv.latent_time'. Genes of interest were plotted along pseudotime using 'mv.dynamic_plot'.

### Transduction of *CTCF* gRNA in CRISPRa line

CRISPRa genetic engineering of the H1 dCas9-VPR line[46] was performed using custom guide RNAs (gRNAs, MilliporeSigma, Supplementary Table 1). gRNAs were resuspended to a final concentration of 100 µM in water. Primers were phosphorylated and ligated together by adding T4 ligation buffer and T4 Polynucleotide kinase enzyme (NEB; B0202A and M0201S) and running on a thermocycler under the following conditions: 37 °C for 30 min; 95 °C for 5 min; and ramp down to 25 °C at 5 °C min⁻¹. Oligos were then diluted with 90 µl of ultrapure water. These oligos were then inserted into the single guide RNA (sgRNA) library backbone (Lenti sgRNA(MS2)_puro) using the Golden Gate reaction. This was achieved by adding a 25 ng µl⁻¹ plasmid backbone to a master mix of Rapid Ligase Buffer 2X (Enzymatics: B1010L), Fast Digest Esp31 (Thermo: FD0454), dithiothreitol (Promega: PRP1171), BSA (NEB: B9000S), T7 DNA ligase, (Enzymatics: L6020L) and the diluted gRNA oligos in a total reaction volume of 25 µl. The Golden Gate assembly reaction was then performed in a thermocycler under the following conditions: 15 cycles of 37 °C for 5 min, 20 °C for 5 min with final hold at 4 °C. Lenti sgRNA (MS2) puromycin optimized backbone was a generous gift from Feng Zhang (Addgene plasmid no. 73797). This final plasmid was then transfected into STBL3 following the same methods as described below in the 'Lentiviral design, preparation and transduction' section.

To transfect the CRISPRa H1 dCas9-VPR stem cell line, lentiviral particles containing gRNA was added at a multiplicity of infection of 5 with polybrene (5 µg ml⁻¹) in culture for 24 h. At confluency, transfected stem cells were passaged and cultured with medium containing puromycin (1 µg ml⁻¹) for selection. To induce CRISPRa expression, doxycycline (MilliporeSigma) was added at 1 µg ml⁻¹ for 7 days during stage 5 of the differentiation protocol.

### Real-time PCR

Cells were lysed directly with RLT buffer from the RNeasy Mini Kit (74016; Qiagen) followed by RNA extraction following the manufacturer's instructions. Complementary DNA was synthesized from the RNA using the High-Capacity cDNA Reverse Transcription Kit (Applied Biosystems; 129382310MG) on a T100 thermocycler (BioRad). PowerUp SYBR Green Master Mix (Applied Biosystems; A257411) was used to run samples on the Quant Studio 6 Pro (Applied Biosystems), and results were analysed using ΔΔCt methodology. The housekeeping genes TBP and GUSB were both used for normalization. Primer sequences used in this paper are listed in Supplementary Table 1. qPCR data were collected from the QuantStudio6 Pro using Design & Analysis 2.6.0.

### ICC

Fluorescence images were taken on a Zeiss Cell Discoverer confocal 7 microscope. For immunocytochemistry (ICC), cells were fixed in 4% paraformaldehyde (Electron Microscopy Science; 15714) for 30 min at RT. For staining, fixed cells were incubated in ICC solution (PBS (Fisher; MT21040CV), 0.1% Triton X (Acros Organics; 327371000) and 5% donkey serum (Jackson Immunoresearch; 01700-121)) for 30 min at RT. Samples were subsequently treated with primary and secondary antibodies in ICC solution overnight at 4 °C and 2 h at RT, respectively. DAPI (Invitrogen; D1306) was used for nuclear staining. Samples were incubated in DAPI for 12 min at RT, washed with ICC solution and stored in PBS until imaging. Antibody details and dilutions can be found in Supplementary Table 1. ImageJ was used for analysis. Quantification was performed by manual counting of cells from analysed fluorescence images and can be found in Supplementary Fig. 3.

### Flow cytometry

Cells were single-cell dispersed by washing with PBS and adding 0.2 ml TrypLE cm⁻² for 10 min at 37 °C. Cells were washed with PBS, centrifuged and fixed by resuspending the cells in 4% paraformaldehyde at 4 °C for 30 min. After another PBS wash, samples were treated with ICC solution for 45 min at RT. Primary antibodies were prepared in ICC solution and incubated on cells overnight at 4 °C. Samples were washed with PBS and incubated for 2 h with secondary antibodies in ICC at 4 °C. Antibody

details and dilutions can be found in Supplementary Table 1. Cells were washed twice with PBS and filtered before running on the LSR Fortessa flow cytometer (BD Bioscience) using BD FACSDiva. FlowJo v10.8.0 (Becton, Dickinson, and Company) was used for analysis. The used gating strategy can be found in Supplementary Fig. 4.

## Hormone content

Hormone content was measured by collecting SC-islets, rinsing with PBS and incubating in an acid–ethanol solution for 48 h at −20 °C. Samples were neutralized with 1 M Tris buffer (Millipore Sigma; T6066). Hormone measurements were made using enzyme-linked immunosorbent assay (ELISA) kits: human insulin ELISA (ALPCO; 80-INSHU-E01.1), somatostatin EIA (Phoenix Pharmaceuticals; EK-060-03), glucagon ELISA (Crystal Chem; 81520), serotonin ELISA (ALPCO; 17-SERHU-E01-FST) and human pro-insulin ELISA (Mercodia; 10-1118-01). ELISAs were performed according to the manufacturer instructions. Results were normalized to cell counts performed on Vi-Cell XR (Beckman Coulter).

## Lentiviral design, preparation and transduction

Gene knockdown of *ARID1B* was performed using pLKO.1 TRC plasmids containing short hairpin RNA (shRNA) sequences targeting *ARID1B* and *GFP* (Control) (Supplementary Table 1). Glycerol stocks were grown, and plasmid DNA was isolated using Qiagen Mini-prep kit (Qiagen; 27115). Plasmid DNA was transfected into One Shot Stbl3 Chemically Competent *Escherichia coli* (Invitrogen; C737303) and spread on an agar plate. After 18 h, one colony was selected, cultured and DNA extracted using the Qiagen Maxi-prep-plus kit (Qiagen; 12981). Viral particles were generated using Lenti-X 293T cells (Takara; 632180) cultured in Dulbecco's modified Eagle medium with 10% heat-inactivated foetal bovine serum (MilliporeSigma; F4135) and 0.01 mM sodium pyruvate (Corning; 25-000-CL) in 10 cm plates (Falcon; 353003). Confluent cells were transfected with 6 μg of shRNA plasmid, 4.5 μg of psPAX2 (Addgene; 12260; gift from Didier Trono) and 1.5 μg pMD2.G (Addgene; 12259; gift from Didier Trono) packaging plasmids in 600 μl of Opti-MEM (Life Technologies; 31985-070) and 48 μl of Polyethylenimine 'Max' MW 40,000 Da (Polysciences; 24765-2) per plate. Medium was refreshed after 16 h. Virus-containing supernatant was collected at 96 h post transfection and concentrated using Lenti-X concentrator (Takara; 631232). Collected lentivirus was titred using Lenti-X GoStix Plus (Takara; 631280). Lentiviral transduction was initiated at the beginning of stage 6 with a multiplicity of infection of 5 for 24 h.

## Statistics and reproducibility

No statistical methods were used to pre-determine sample sizes, but our sample sizes are similar to those reported in previous publications[5,9,16,31,33]. This study contains multi-omic sequencing datasets of SC-islets from three independent differentiation batches (three datasets), human islets from four donors (four datasets), SC-islets after 3 weeks, 4 weeks, 6 months and 12 months into stage 6 with one replicate each (five datasets), CTCF CRISPRa differentiations of one replicate from each condition (control and doxycycline-induced CTCF overexpression; two datasets), and ARID1BshRNA SC-islets of one replicate from each condition (control and shARID1B knockdown; two datasets). Information on datasets used and sample details can be found in Supplementary Table 2. Cell selection criteria or exclusion methods for single-cell analysis can be found in Supplementary Fig. 2. Statistical significance from the multi-omic analyses was calculated using the Wilcoxon rank sum test for RNA expression, logistic regression for motif chromatin accessibility and Bonferroni correction to account for multiple testing. For in vitro experiments, we performed unpaired or paired parametric *t*-tests (two-sided) and one-way ANOVA with Tukey's multiple comparison testing to determine significance. Data distribution was assumed to be normal, but this was not formally tested. All in vitro experiment data points presented are biological replicates and can be found in Source Data. Significant values are marked on the basis of *P* values using non-significant (NS) >0.05, *<0.05, **<0.01, ***<0.001 and ****<0.0001.

## Reporting summary

Further information on research design is available in the Nature Portfolio Reporting Summary linked to this article.

## Data availability

Sequencing data that support the findings of this study have been deposited in the Gene Expression Omnibus (GEO) under accession code GSE199636. Source data are provided with this paper. GRCh38 human genome[95], MSigDB[96] and JASPAR2020[93] databases were used. All other data supporting the findings of this study are available from the corresponding author on reasonable request.

## Code availability

Codes used for analysing single-nucleus multi-omic sequencing are available on https://github.com/punnaug. No custom codes or mathematical algorithms were developed or used in this study.

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

## Acknowledgements

This work was funded by NIH (R01DK114233 and R01DK127497 to J. R. Millman), JDRF (5-CDA-2017-391-A-N to J. R. Millman) and startup funds from the Washington University School of Medicine Department of Medicine to J. R. Millman. N.J.H. was supported by a JDRF Advanced Postdoctoral Fellowship (3-APF-2020-930-A-N). M.I. was supported by Rita Levi-Montalcini Postdoctoral Fellowship in Regenerative Medicine and the NIH (T32DK007120). M.M.M. was supported by the NIH (T32GM139774). J. R. Miller was supported by a Washington University BioSURF award. D.A.V.-P. was supported by the NSF Graduate Research Fellowship Program (DGE-2139839 and DGE-1745038). L.V.-C. was supported by the NIH (F31DK125068). Microscopy was performed through the Washington University Center for Cellular Imaging, which is supported by the Washington University School of Medicine, the Children's Discovery Institute (CDI-CORE-2015-505) and the Foundation for Barnes-Jewish Hospital

(3770). Microscopy analysis was supported by the Washington University Diabetes Research Center (P30DK020579) and Center of Regenerative Medicine. We thank the Genome Technology Access Center at the McDonnell Genome Institute at Washington University School of Medicine for help with genomic analysis, which is supported by NIH (P30CA91842) support to the Siteman Cancer Center and by ICTS/CTSA (UL1TR002345) from the National Center for Research Resources (NCRR). We thank N. Udomkittivorakul for the graphics design, D. Melton (Harvard University) for the HUES8 cell line and L. Barret (Broad Institute of MIT and Harvard University) for the CRISPRa VPR cell line.

## Author contributions

P.A. and J. R. Millman designed all experiments and performed all in vivo experiments. P.A. and M.I. performed all computational analysis and associated cell culture. P.A., E.M., M.M.M., M.D.S., M.I., D.A.V.-P., J. R. Miller, S.E.G. and L.V.-C. performed in vitro experiments. P.A., N.J.H., M.I. and J. R. Millman wrote the manuscript. All authors revised and approved the manuscript.

## Competing interests

P.A., N.J.H., L.V.-C. and J. R. Millman are inventors on related patents and patent applications. J. R. Millman is/has served as a consultant/ employee for Sana Biotechnology. L.V.-C. is currently employed by Sana Biotechnology. J. R. Millman and L.V.-C. have stock or stock options in Sana Biotechnology. The remaining authors declare no competing interests.

## Additional information

**Extended data** is available for this paper at https://doi.org/10.1038/s41556-023-01150-8.

**Correspondence and requests for materials** should be addressed to Jeffrey R. Millman.

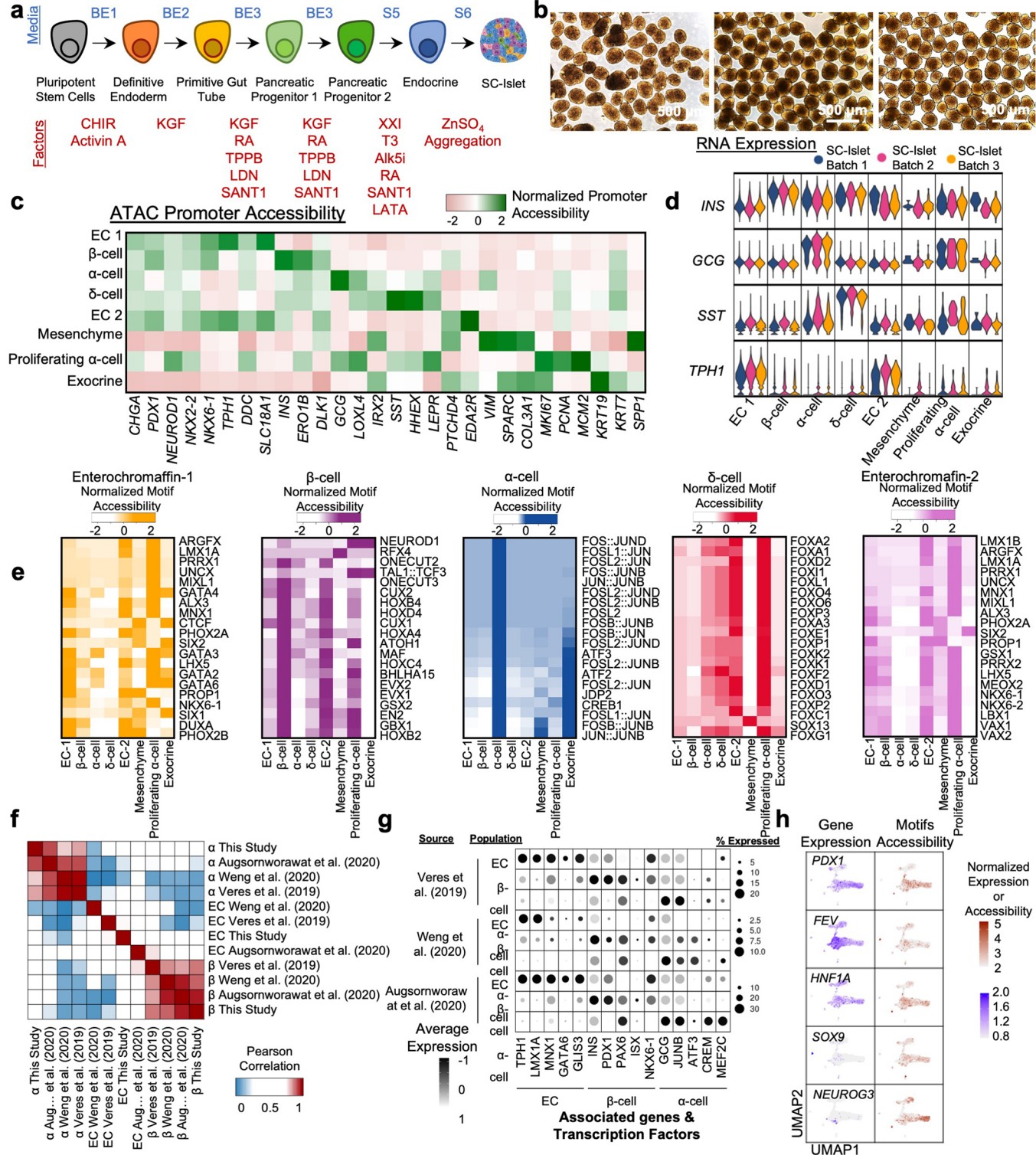

**Extended Data Fig. 1 | Single-cell Multiomic ATAC and gene expression characterization of SC-islets. a**, Schematic diagram outlining the differentiation protocol for generating SC-islets. **b**, Brightfield images of differentiated SC-islets from the three batches used for multiomic sequencing. Individual image representative of 1 sample. **c**, Heatmap showing ATAC promoter accessibility of markers associated with each cell type (29526 cells from 3 independent differentiations; integration of all samples). **d**, Violin plots for INS, GCG, SST, and TPH1 gene expression in SC-islet cell types from different datasets of different differentiation batches. **e**, Heatmaps showing the top 20 enriched motifs in SC-EC1, SC-β, SC-α, SC-δ, and SC-EC2 cells. **f**, Pearson correlation analysis using the top 2000 variable genes comparing correlation of SC-β, SC-α, and SC-EC in SC-islets from this study and other literatures using other differentiation protocols. SC-β, and SC-α have similar gene expression profiles across multiple datasets. **g**, Dot plots showing the upregulation, by gene expression, of identified active transcription factors in SC-β, SC-α, and SC-EC cells of datasets from other studies. **h**, Feature plot showing gene expression of progenitor transcription factors and chromatin accessibility of their binding motifs. EC, enterochromaffin cells.

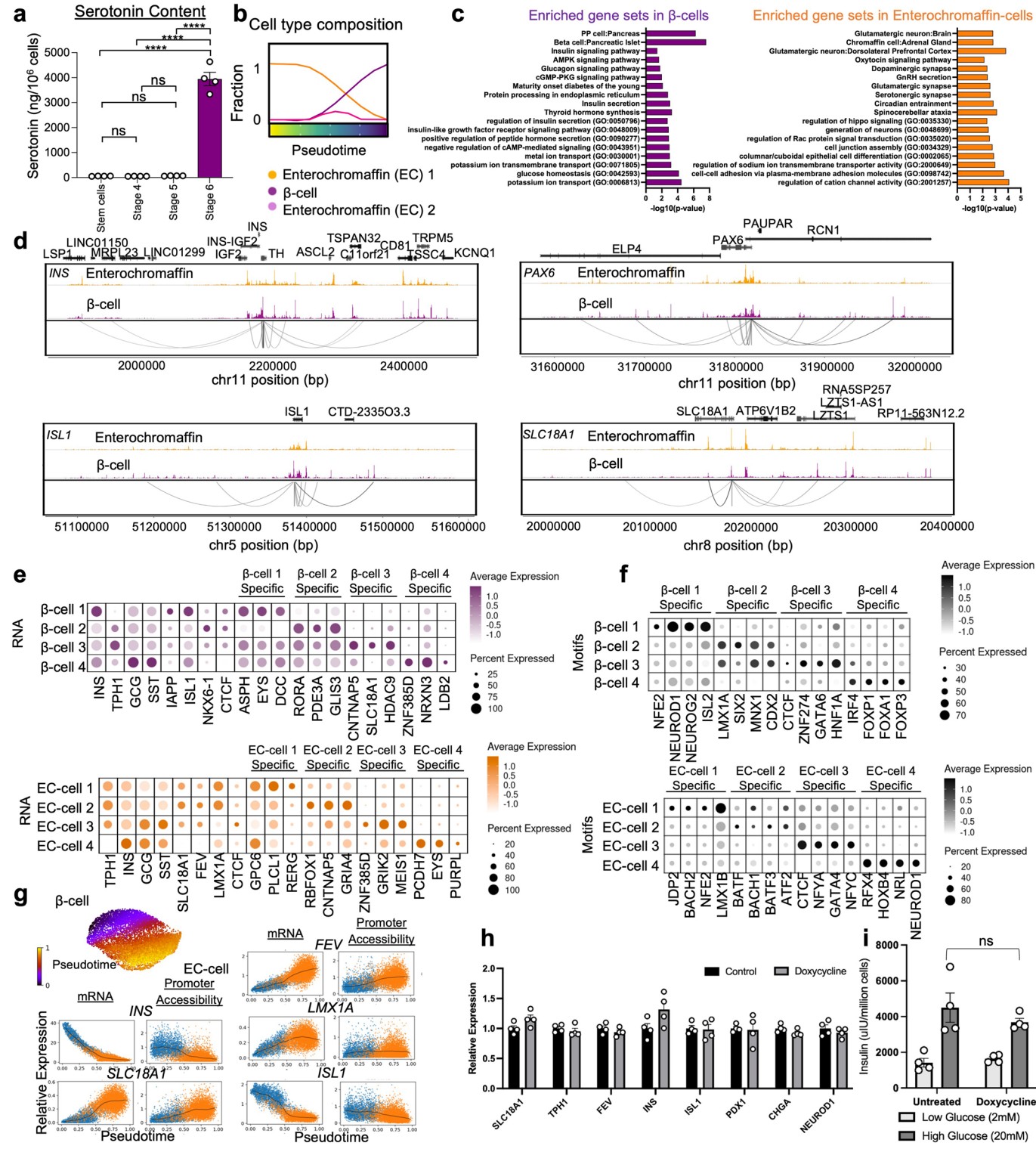

Extended Data Fig. 2 | See next page for caption.

**Extended Data Fig. 2 | Comparing SC-β and SC-EC cells. a**, Protein quantification plot showing mean ± s.e.m. (by ELISA, n = 4 biologically independent samples) of serotonin content in stem cells and stages 4–6 of differentiation (Stem cells vs Stage 4, P = 1.00, Stem cells vs Stage 5, P = 1.00, Stem Cells vs Stage 6, P = 6.17 × 10⁻¹⁰, Stage 4 vs Stage 5, P = 1.00, Stage 4 vs Stage 6, P = 6.06 × 10⁻¹⁰, Stage 5 vs Stage 6, P = 6.42 × 10⁻¹⁰). Statistical significances were assessed by one-way ANOVA with Tukey's multiple comparison testing, reporting adjusted P-value. **b**, Composition of cell types represented on the trajectory analysis. **c**, Gene set enrichment of analysis showing enrichment of gene sets in segments of trajectory representing SC-β and SC-EC cells. Non-adjusted p-values were computed using two-sided Fisher exact test. **d**, ATAC peaks show chromatin accessibility pattern differences and highlight cis-regulatory elements around the INS, ISL1, PAX6, and SLC18A1 gene regions. **e**, Dot plots showing marker genes and unique genes associated with subpopulations identified from the reclustering of SC-β and SC-EC cell populations. **f**, Dot plots showing unique motif chromatin accessibility in subpopulations of SC-β and

SC-EC cells. **g**. Trajectory analysis of the EC cell and β cell population using 'MultiVelo' package showing gene expression and promoter accessibility along. The trajectory analysis validates expression features of EC cell and β cell genes identified in the previous analysis with 'Monocle' package (Subset of 21317 cells from 3 independent differentiations; integration of all samples). **h**, qPCR analysis, plotting mean ± s.e.m. (n = 4 biologically independent samples), of non-transduced control cells comparing doxycycline effects on the expression of EC cell and β cell associated genes. All comparisons show no statistical significance. (*SLC18A1*, P = 0.095; *TPH1*, P = 0.45; *FEV*, P = 0.31; *INS*, P = 0.090; *ISL1*, P = 0.86; *PDX1*, P = 0.86; *CHGA*, P = 0.13; *NEUROD1*, P = 0.37). Statistical significance was assessed by unpaired two-sided t-test. **i**, Glucose stimulated insulin secretion assay, plotting mean ± s.e.m. (by ELISA, n = 4 biologically independent samples), of differentiated non-transduced control SC-islets. Treatment of doxycycline on non-transduced control shows no significant changes is insulin secretion (P = 0.38). Statistical significances were assessed using unpaired two-sided t-test. EC, enterochromaffin cells.

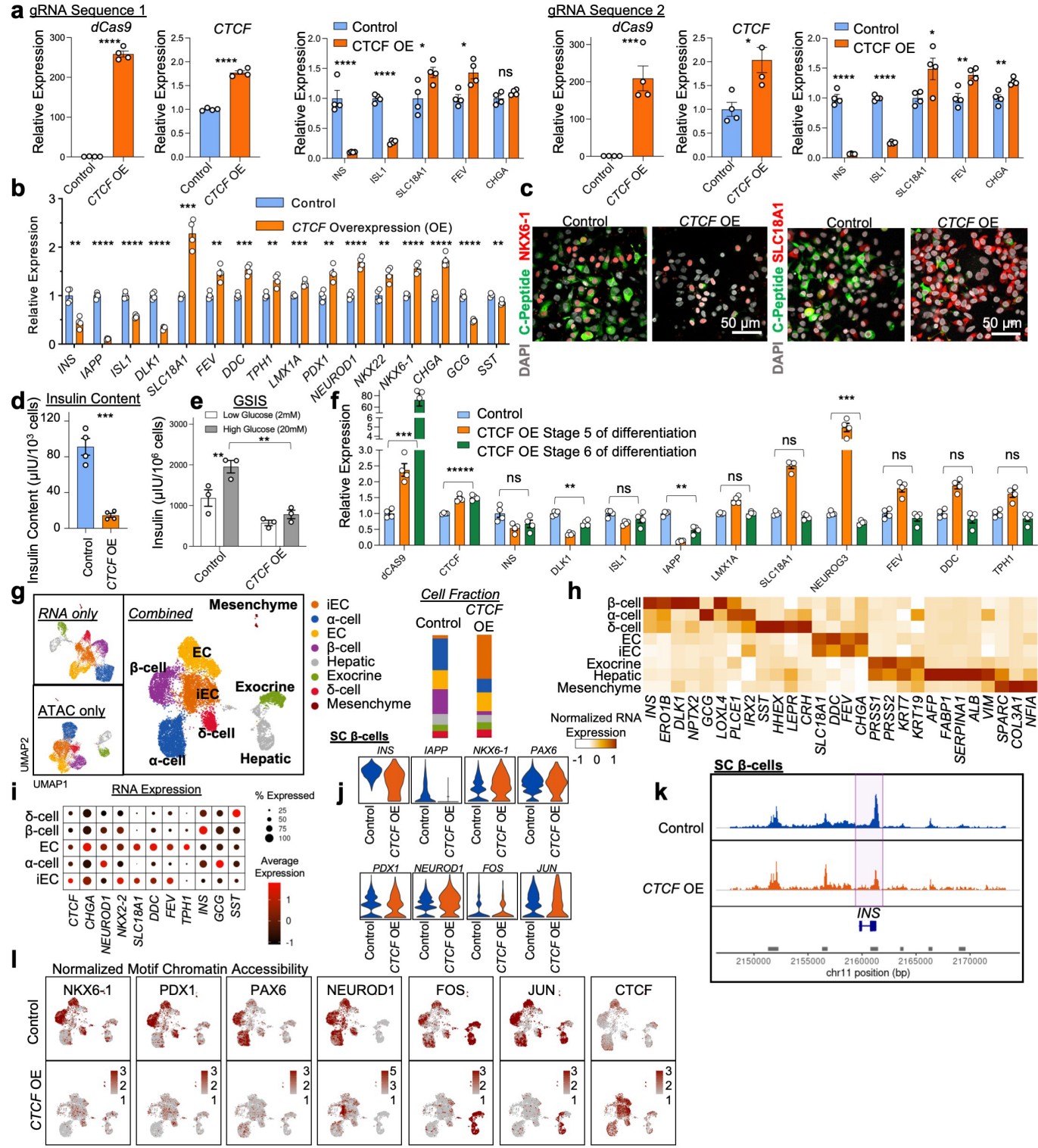

**Extended Data Fig. 3 | See next page for caption.**

**Extended Data Fig. 3 | CTCF induction during differentiation using doxycycline inducible CRISPRa stem cells. Control represents cells without doxycycline treatment. a**, qPCR, plotting mean ± s.e.m. (n = 4 biologically independent samples), showing upregulation of dCas9 (gRNA sequence 1, P = 4.2 × 10⁻⁸; gRNA sequence 2, P = 7.8 × 10⁻⁴), *CTCF* overexpression (gRNA sequence 1, P = 1.4 × 10⁻⁶; gRNA sequence 2, P = 0.014), and changes in expression (gRNA sequence 1, *INS* (P = 5.6 × 10⁻⁴) *ISL1* (P = 1.1 × 10⁻⁶), *SLC18A1* (P = 0.034), *FEV* (P = 0.012), *CHGA* (ns, P = 0.12); gRNA sequence 2, *INS* (P = 4.4 × 10⁻⁶), *ISL1* (P = 7.8 × 10⁻⁸), *SLC18A1* (P = 0.042), *FEV* (P = 0.0069), *CHGA* (P = 0.0050), upon doxycycline. Statistical significance assessed by unpaired two-sided t-test. **b**, qPCR analysis, plotting mean ± s.e.m. (n = 4 biologically independent samples), of *CTCF* overexpressed (*INS*, P = 0.0010; *IAPP*, P = 1.5 × 10⁻⁷; *ISL1*, P = 3.0 × 10⁻⁵; *DLK1*, P = 3.7 × 10⁻⁶), EC cells (*SLC18A1*, P = 1.6 × 10⁻⁴; *FEV*, P = 0.0019; *DDC*, P = 1.2 × 10⁻⁴; *TPH1*, P = 0.0047; *LMX1A*, P = 4.6 × 10⁻⁴; *PDX1*, P = 0.0031; *NEUROD1*, P = 6.2 × 10⁻⁵; *NKX2-2*, P = 0.0047; *NKX6-1*, P = 7.5 × 10⁻⁵; *CHGA*, P = 6.8 × 10⁻⁵; *GCG*, P = 5.9 × 10⁻⁶; *SST*, P = 0.0046). Statistical significance assessed by unpaired two-sided t-test. **c**, Immunocytochemistry of SC-islets with *CTCF* overexpression showing C-peptide (green), NKX6-1 (red), and SLC18A1 (red). Individual image representative of 6 biologically independent samples. **d**, Protein quantification plot showing mean ± s.e.m. (by ELISA, n = 4 biologically independent samples)

after CTCF overexpression (P = 2.2 × 10⁻⁴). Statistical significances assessed by unpaired two-sided t-test. **e**, Insulin secretion, plotting mean ± s.e.m. (by ELISA, n = 3 biologically independent samples), of control (P = 0.0095) and *CTCF* overexpression (P = 0.0032). Statistical significance assessed using paired two-sided t-test and unpaired two-sided t-test respectively. **f**, qPCR analysis, plotting mean ± s.e.m. (n = 4 biologically independent samples), comparing CTCF overexpression during Stage 5 of protocol or at Stage 6 of protocol (*dCAS9*, P = 6.13 × 10⁻⁴; *CTCF*, P = 2.0 × 10⁻⁵; *INS*, P = 0.10; *DLK1*, P = 0.0023; *ISL1*, P = 0.18; *IAPP*, P = 3.3 × 10⁻⁴; *LMX1A*, P = 0.97; *SLC18A1*, P = 0.12; *NEUROG3*, P = 9.3 × 10⁻⁴; *FEV*, P = 0.30; *DDC*, P = 0.27; *TPH1*, P = 0.21) Statistical significance assessed by unpaired two-sided t-test. **g**, UMAPs (12467 cells from 2 independent biological samples; integration of all samples) showing cell types in SC-islets induced with *CTCF* overexpression. **h**, Heatmap showing normalized gene expression associated with *CTCF* experiment. **i**, Dot plots highlighting expression of *CTCF* and endocrine-associated genes. **j**, Violin plot of β-cell associated gene expressions of CTCF overexpression in the SC-β cell population. **k**, Chromatin accessibility around *INS* genomic region of SC-β cells. **l**, Feature plots showing decreased accessibility of β-cell motifs and increased accessibility to CTCF motifs in endocrine populations with CTCF overexpression. EC, enterochromaffin cells; ns, not significant.

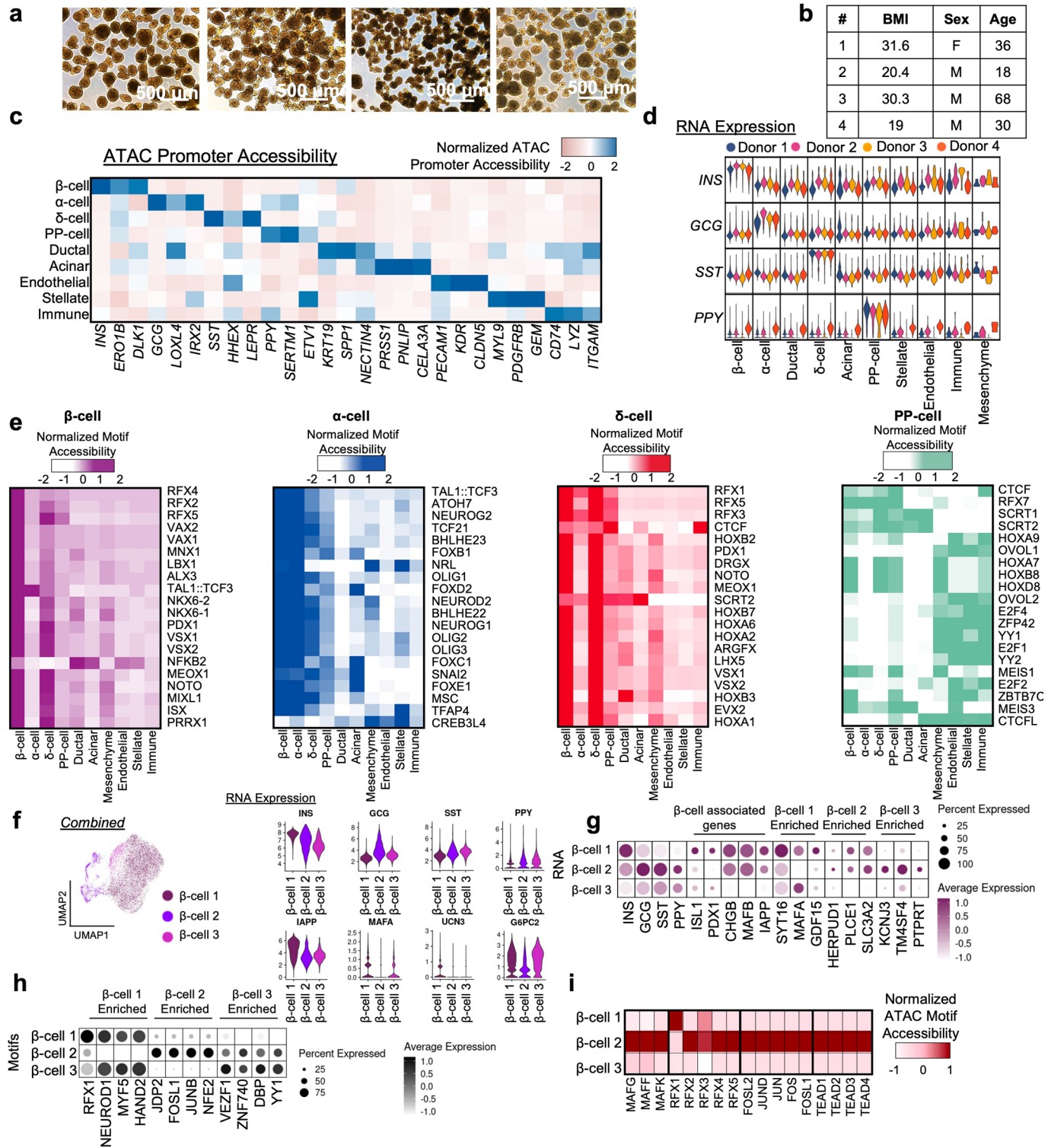

**Extended Data Fig. 4 | Single-cell Multiomic ATAC and gene expression characterization of primary huma islets. a**, Brightfield images of primary islets used for multiomic sequencing. Individual image representative of 1 sample. **b**, Brief primary islet donor information. **c**, Heatmap showing ATAC promoter accessibility of markers associated with each cell type (30202 cells from 4 independent biological samples; integration of all samples). **d**, Violin plots for INS, GCG, SST, and PPY gene expression in primary islet cell types from different donors. **e**, Heatmaps showing the top 20 enriched motifs in primary β, α, δ, and PP cells. **f**, Re-clustering analysis of primary β cells highlighting heterogeneity by variations in mature and polyhormonal gene expressions. **g**, Dotplots showing marker genes and unique genes associated with subpopulations identified from the re-clustering of primary β cells. **h**, Dotplots showing unique motif chromatin accessibility in in subpopulations primary β cells. **i**, Chromatin accessibility of transcription factor binding sites of MAF, RFX, FOS/JUN, and TEAD motif family in primary β cell subpopulations. PP, Pancreatic progenitors.

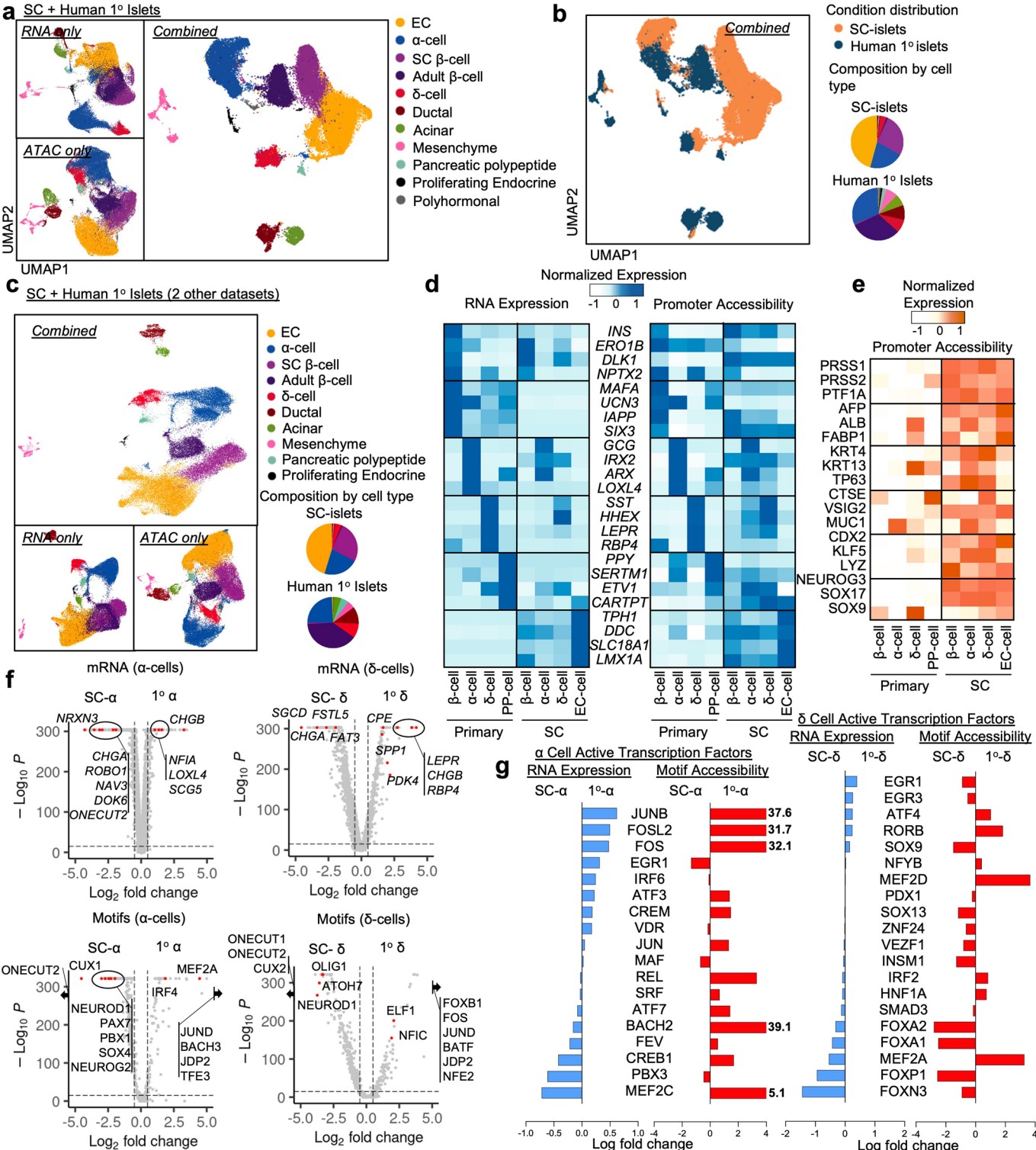

**Extended Data Fig. 5 | Multiomic sequencing comparison of SC-islets and primary human islets. a**, Integrative UMAP showing cells from SC-islets and primary islets (47566 cells from 5 independent biological samples; 3 SC-islets, 2 representative primary islets- donor #1 and #2). **b**, Integrative UMAP plotted by SC-islet or primary islet condition. Pie charts show composition information of cell types identified. **c**, Validation analysis showing integrative UMAP of cells from SC-islets and other primary islet datasets (41688 cells from 5 independent biological samples; 3 SC-islets, 2 representative primary islets- donor #3 and #4). Pie charts show composition information of cell types identified. **d**, Validation analysis showing heatmap highlighting and comparing identity (β, adult β, α, δ, PP and EC) associated gene expression and ATAC promoter accessibility of endocrine cells in SC-islets and primary islets (Human islet donors #3 and

#4). **e**, Validation analysis showing heatmap highlighting and comparing off-target identity (Exocrine, hepatic, esophagus, stomach, intestinal, pancreatic progenitor) associated gene expression and ATAC promoter accessibility in SC-islet cells and primary islet cells (Human islet donors #3 and #4). **f**, Differential gene expression (top) and motif chromatin accessibility analysis (right) for α-cells (left) and δ-cells (right). Statistical significance was assessed by Wilcoxon rank sum test for RNA expression and logistic regression for motif chromatin accessibility. **g**, Bar graphs showing fold change comparing SC and primary α and δ cell populations, showing gene expression and motif chromatin accessibility of identified transcription factors associated with the respective cell types. EC, enterochromaffin cells.

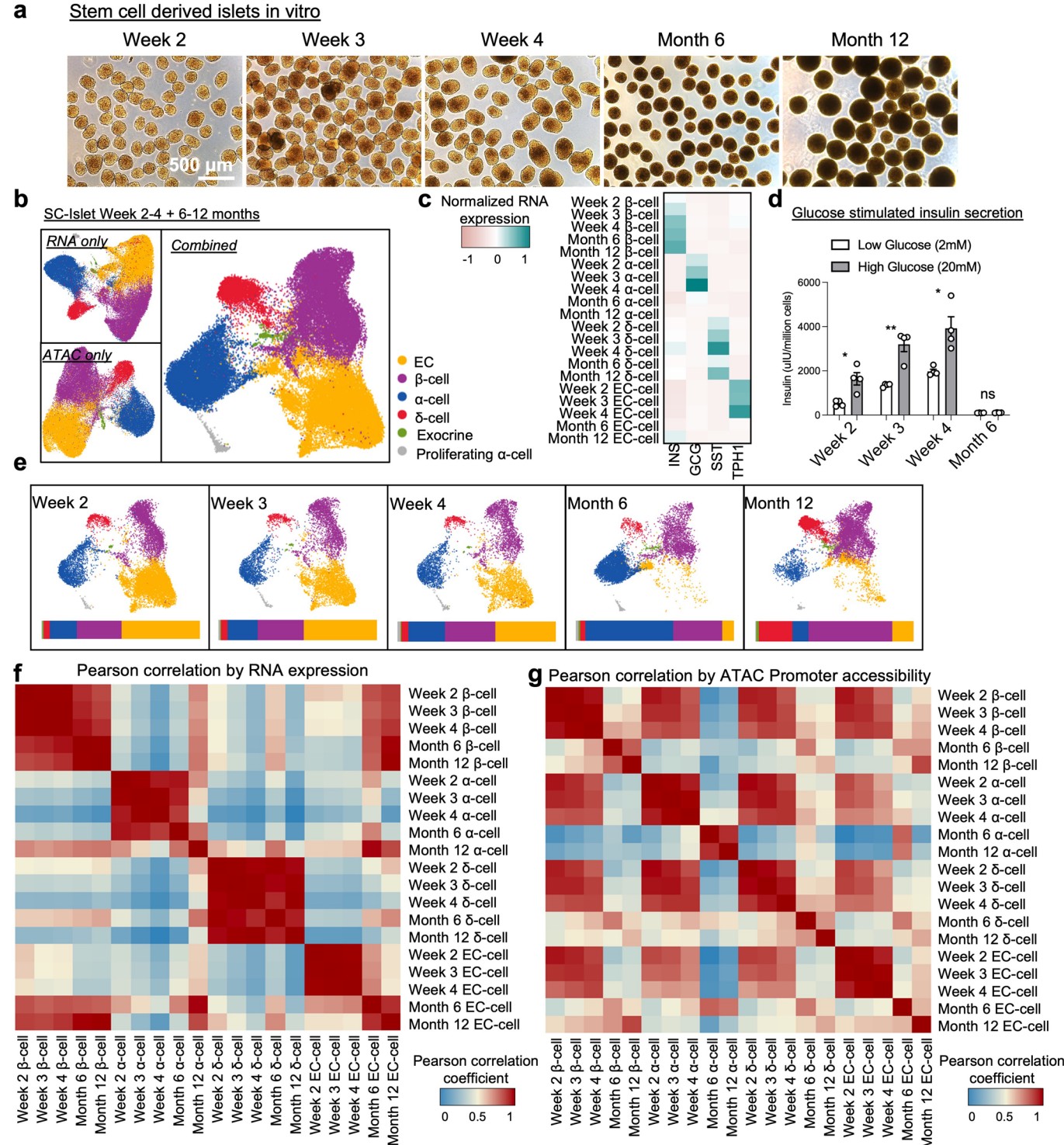

**Extended Data Fig. 6 | Time course characterization and analysis of long-term in vitro SC-islets. a**, Bright field images of SC-islets from week 2, week 3, week 4, month 6, and month 12 of in vitro culture. Individual image representative of 1 sample. **b**, Integrative UMAP of in vitro SC-islet cells including all time points. **c**, Heatmap showing increase, or decrease, of marker gene expression in SC-islet cells throughout time (40332 cells from 5 independent biological samples, 1 from each timepoint). **d**, Glucose stimulated insulin secretion assay, plotting mean ± s.e.m. (by ELISA, n = 4 biologically independent samples), of SC-islets cultured in vitro at week 2 (P = 0.020), week 3 (P = 0.0080), week 4 (P = 0.046), and month 6 (P = 0.27) time point. Statistical significances were assessed using paired two-sided t-test. **e**, UMAP and bar plots showing composition information of cell types from integrated time course datasets. **f** and **g**, Pearson correlation to compare all timepoints, of cell types, (**f**) using top 2000 most variably expressed genes (mRNA), and (**g**) using top 2000 most variably accessible promoters from ATAC. EC, enterochromaffin; ns, not significant.

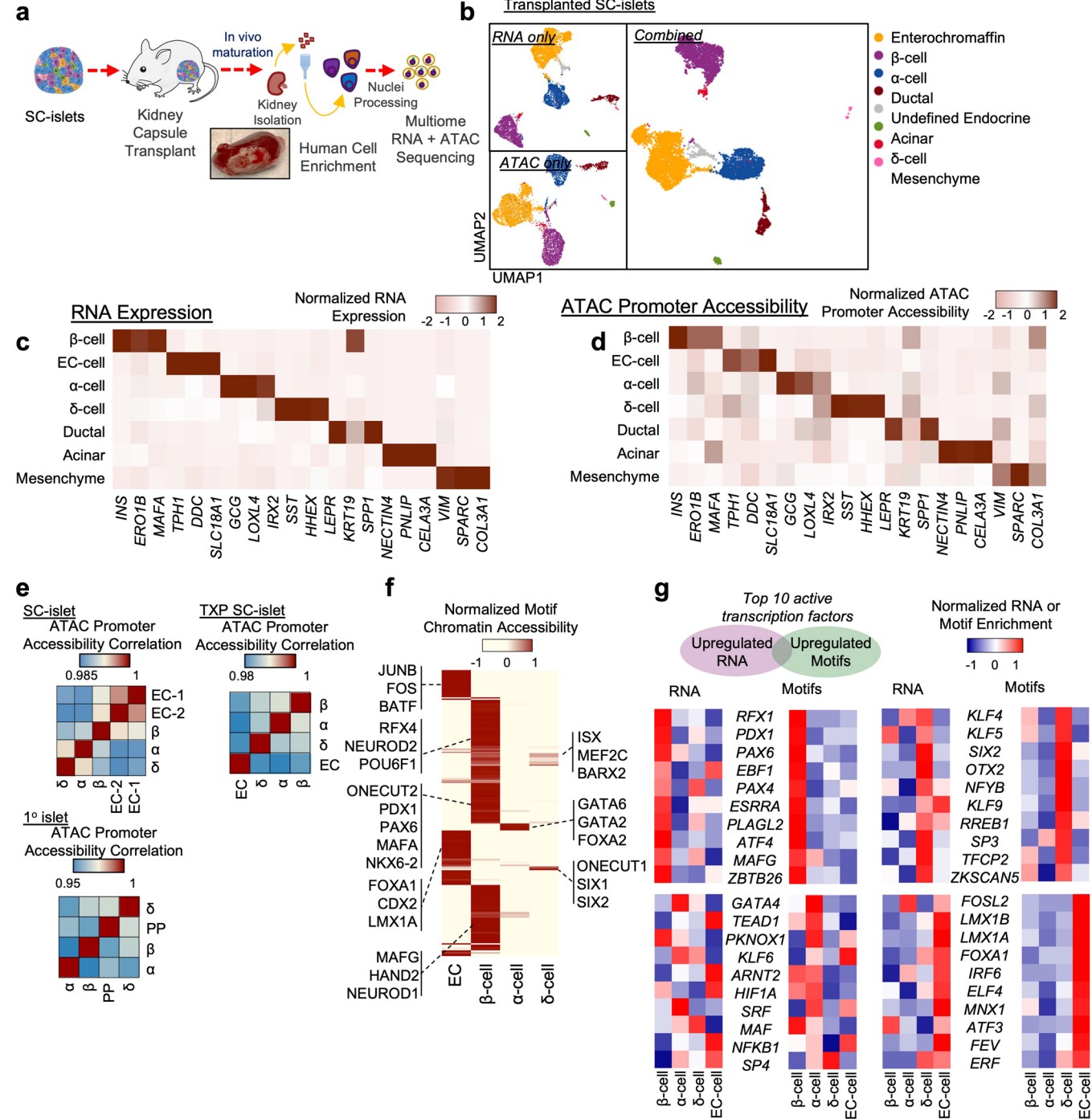

**Extended Data Fig. 7 | Single-cell Multiomic ATAC and gene expression characterization of transplanted SC-islets. a**, Schematics of SC-islet transplantation and retrieval of graft after 6 months in vivo. **b**, UMAP and identification of transplanted SC-islet cells using gene expression and chromatin information (7162 cells from 3 independent biological samples; integration of all samples). **c** and **d**, Heatmap showing gene expression (**c**) and ATAC promoter accessibility (**d**) of markers associated with each cell type. **e**, Pearson correlation analysis using top 2000 most variable ATAC promoter accessibility of key

endocrine populations from SC-islets, transplanted SC-islets, and primary human islets. This analysis highlights the distinctiveness of chromatin identity acquired in cell types from SC-islets after transplantation. **f**, Heatmap showing the top 200 variable motifs within endocrine cell populations and highlighting motif markers for each in vivo cell type. **g**, Heatmaps highlighting gene expression and ATAC motif accessibility of top 10 active transcription factors co-enriched with both features in transplanted SC-β, SC-α, SC-δ, and SC-EC cells.

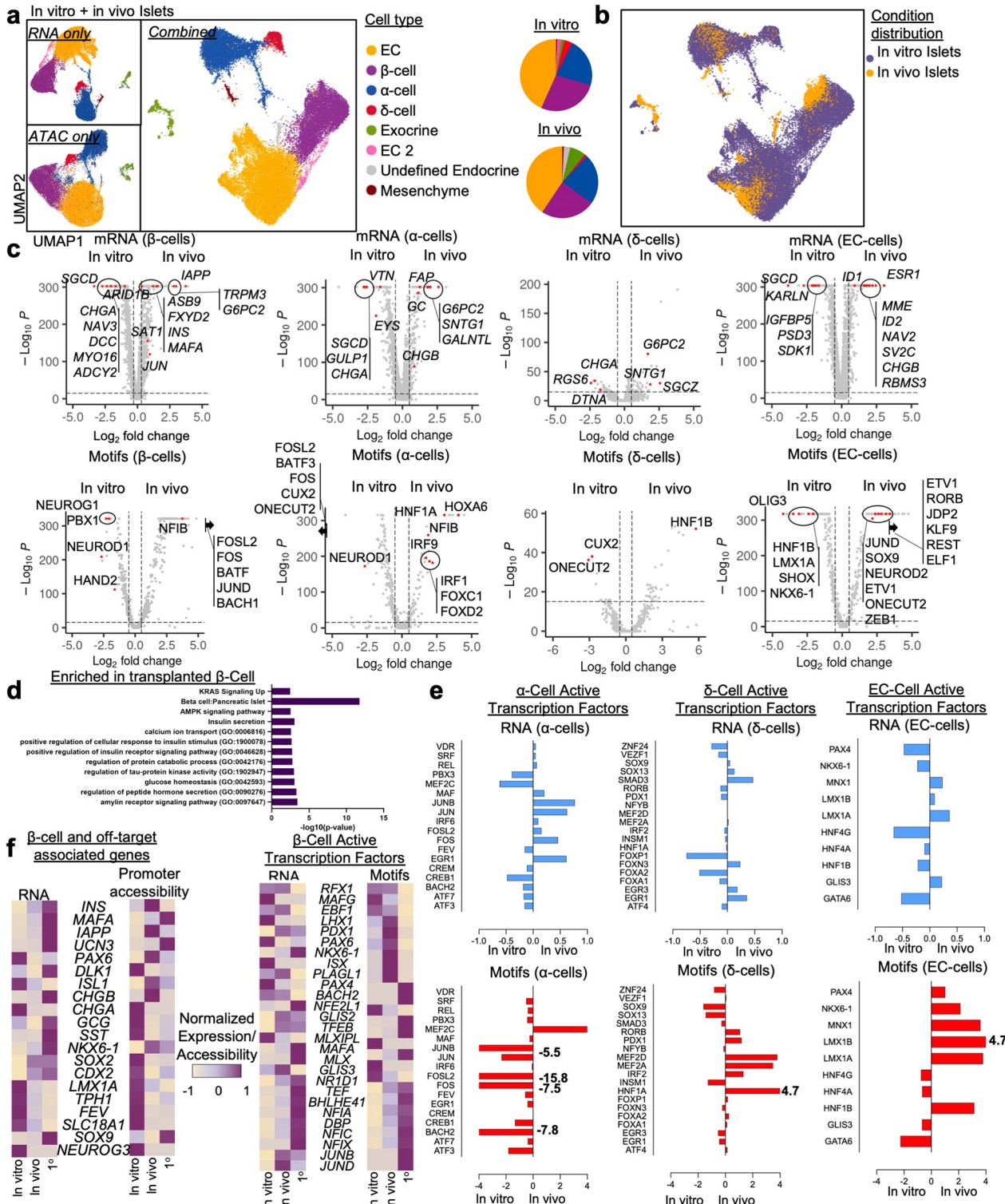

**Extended Data Fig. 8 | Multiomic sequencing comparison of in vitro SC-islets and transplanted in vivo SC-islets. a**, Integrative UMAP clustering showing cells from in vitro SC-islets and in vivo SC-islets using gene expression and chromatin information (36688 cells from 6 independent biological samples; 3 *in vitro* SC-islets, 3 *in vivo* SC-islets; integration of all samples). The pie chart shows composition of cell types in each condition. **b**, UMAP showing distribution of SC-islet cells from in vitro or after transplanted in vivo condition. This plot highlights the separation of clusters from transplanted in vivo SC-β and SC-EC cells. **c**, Differential gene expression analysis (top) and motif chromatin accessibility analysis (bottom) of SC-β, SC-α, SC-δ and SC-EC cell populations from in vitro SC-islets and in vivo SC-islets. Statistical significance was assessed

by two-sided Wilcoxon rank sum test for RNA expression and two-sided logistic regression for motif chromatin accessibility. **d**, Gene set enrichment analysis showing enrichment of gene sets comparing in vitro SC-β cells and vivo SC-β cells. Non-adjusted p-values were computed using two-sided Fisher exact test. **e**, Bar graphs showing fold change differences of cell type associated active transcription factors in SC-α, SC-δ and SC-EC cells from in vitro and in vivo conditions. **f**, Heatmap comparing gene expression, promoter accessibility, or motif chromatin accessibility in β-cells from in vitro SC-islets, in vivo islets, and primary islets (26697 cells from 5 independent biological samples; 1 representative *in vitro* SC-islets, 1 representative primary islets, 3 *in vivo* SC-islets). EC, enterochromaffin.

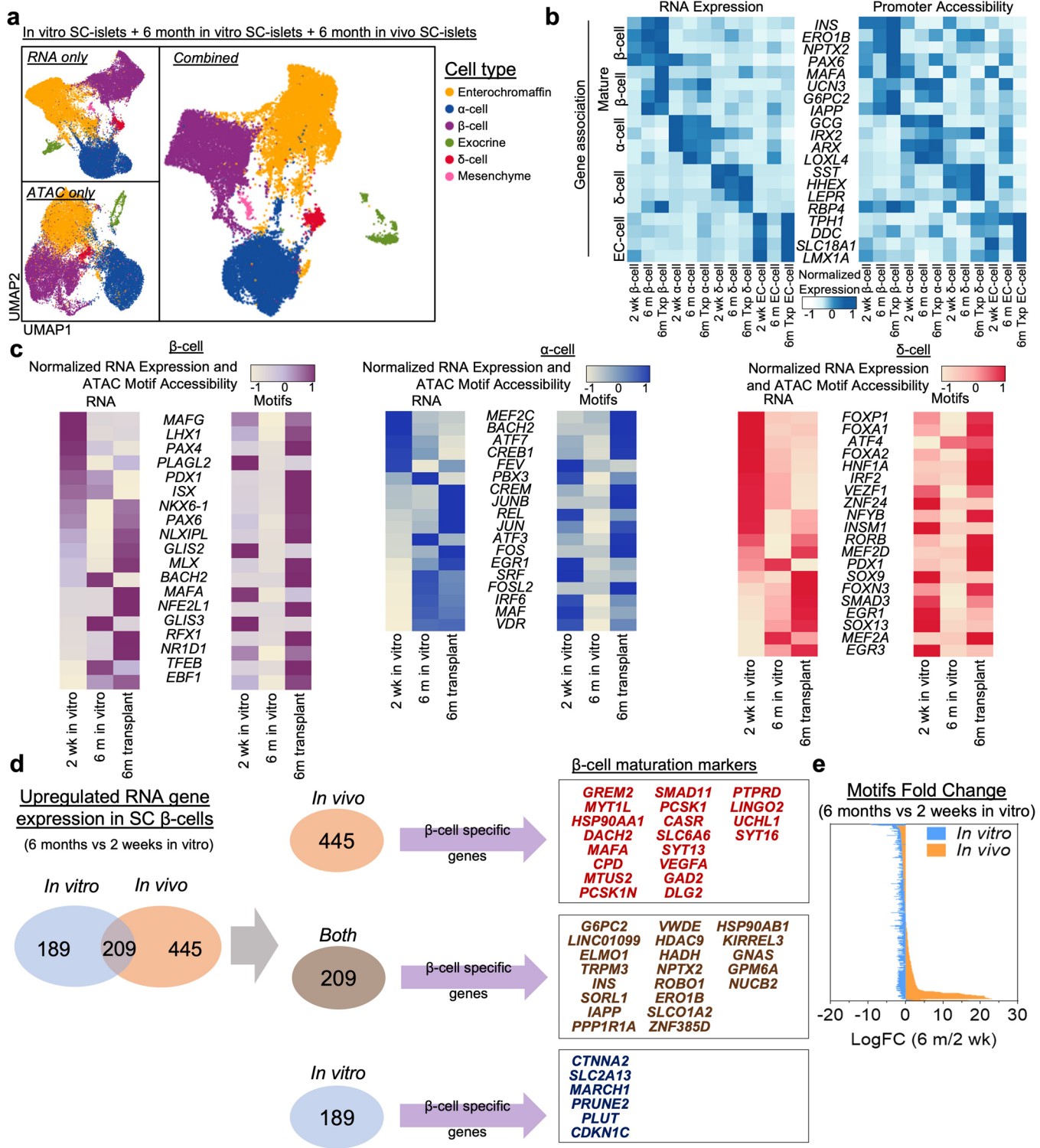

**Extended Data Fig. 9 | Multiomic sequencing comparison of week 2 in vitro SC-islets with 6 months in vitro SC-islets and 6 months in vivo SC-islets. a,** Integrative UMAP clustering showing cells from 2 weeks in vitro SC-islets, 6 months in vitro SC-islets, and 6 months in vivo SC-islets using gene expression and chromatin information (24491 cells from 5 independent biological samples; 1 representative week 2 SC-islets, month 6 SC-islets, 3 month 6 *in vivo* SC-islets). **b,** Heatmap showing gene expression and promoter accessibility of gene markers for SC-β, SC-α, SC-δ, and SC-EC cells. **c,** Gene expression and motifs accessibility

of active transcription factors associated with SC-β, SC-α, and SC-δ identity. **d,** Gene list showing markers for β cell identity 6 months in vitro, in vivo or both. Initial number of genes represent upregulated genes in the 6 month conditions (vitro and vivo) when compared to 2 week SC-islets. List of genes was cross-referenced with β cell upregulated genes from primary human islets. **e,** Plot highlighting greater increase of motifs accessibility from 6 months in vivo SC-β cells when compared in 6 months in vitro.

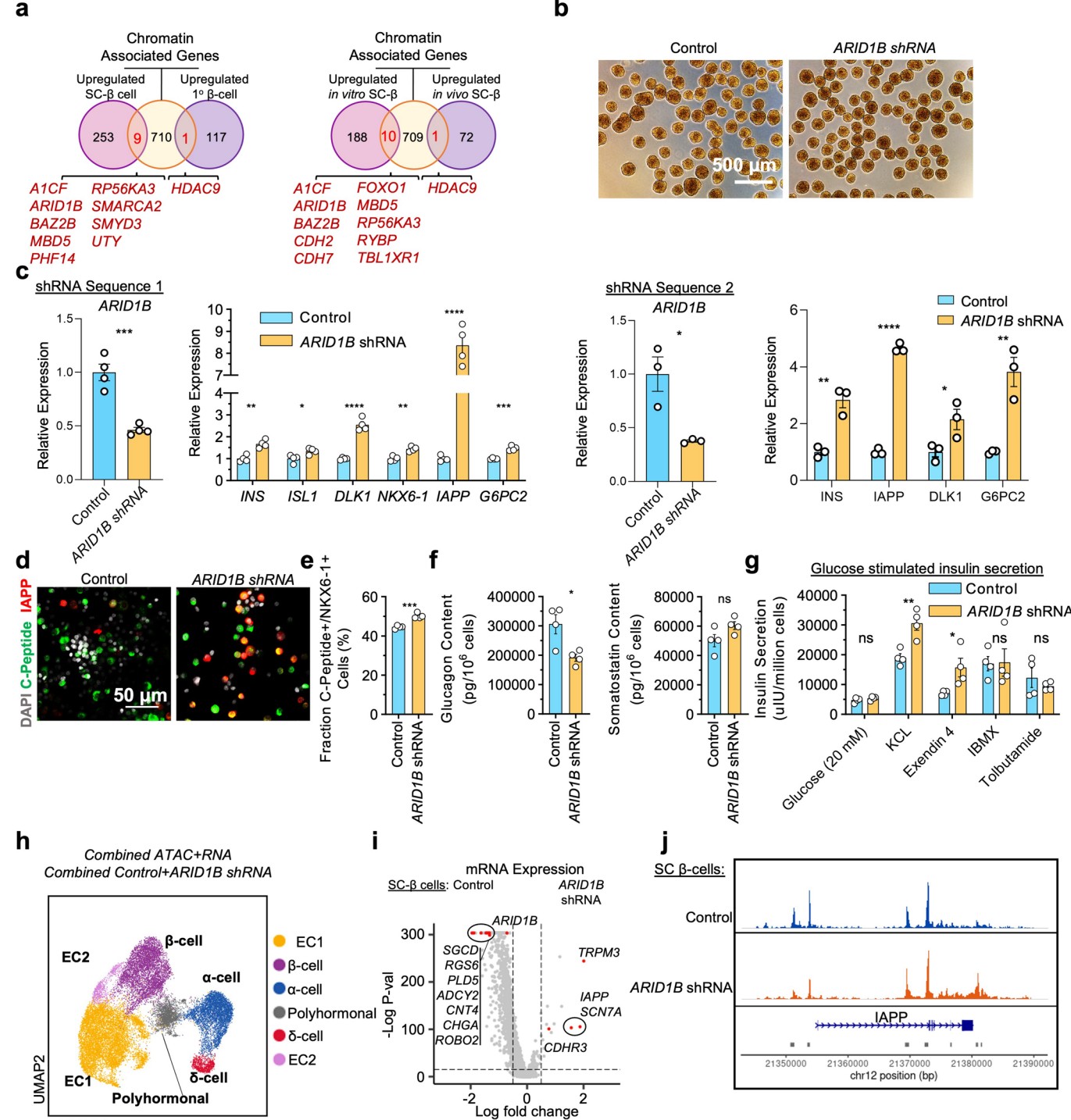

**Extended Data Fig. 10 | See next page for caption.**

**Extended Data Fig. 10 | ARID1B knockdown increases expression of identity genes and chromatin features in SC-β cells. a**, Cross reference map of upregulated SC-β cell and primary β-cell genes (left) or transplanted SC-β cell genes (right) with chromatin associated genes to highlight regulators associated with each cell states. Individual image representative of 1 sample. **b**, Brightfield images of SC-islets transfected with lentivirus carrying ARID1B shRNA. **c**, qPCR analysis, plotting mean ± s.e.m. (n = 4 biologically independent samples for shRNA sequence 1; n = 3 biologically independent samples for shRNA sequence 2), showing reduced expression of *ARID1B* using lentiviruses resulting in increased β cell identity gene expressions. (shRNA sequence 1: ARID1B, P = 5.3 × 10$^{-4}$; *INS*, P = 0.0014; *ISL1*, P = 0.022; *DLK1*, P = 4.3 × 10$^{-5}$; *NKX6-1*, P = 0.0015; *IAPP*, P = 3.0 × 10$^{-6}$; *G6PC2*, P = 2.1 × 10$^{-4}$) Two shRNA sequences were tested for validation of results. (shRNA sequence 2: ARID1B, P = 0.018; *INS*, P = 0.0035; *IAPP*, P = 5.9 × 10$^{-6}$; *DLK1*, P = 0.045, G6PC2, P = 0.0054) Statistical significance was assessed by unpaired two-sided t-test. **d**, Confocal fluorescent images showing increased expression of amylin in SC-islets with *ARID1B* knockdown. Individual image representative of 5 biologically independent samples. **e**, Flow cytometry analysis, plotting mean ± s.e.m. (n = 4 biologically independent samples), of SC-islets with *ARID1B* shRNA showing increased

fraction of cells with C-peptide expression (P = 2.2 × 10$^{-5}$) Statistical significance was assessed by unpaired two-sided t-test. **f**, ELISA quantification, plotting mean ± s.e.m. (n = 4 biologically independent samples) of glucagon (P = 0.017), and somatostatin (ns, P = 0.087) content. Statistical significance was assessed by unpaired two-sided t-test. **g**, Glucose stimulated insulin secretion assay, plotting mean ± s.e.m. (by ELISA, n = 4 biologically independent samples), comparing insulin secretion at high glucose (20 mM) stimulation from control (GFP shRNA) and ARID1B shRNA SC-islets in presence of various secretagogues. (Glucose (20 mM), P = 0.27; KCL, P = 0.049; Exendin 4, P = 0.035; IBMX, P = 0.94; Tolbutamide, P = 0.43) Statistical significances were assessed using paired two-sided t-test. **h**, Single-cell multiomic sequencing UMAPs of *ARID1B* knockdown SC-Islets, showing identified cell types, and cell knockdown condition using both gene expression and ATAC information (21969 cells from 2 independent biological samples; integration of all samples). **i**, Differential gene expression analysis of SC-β cells comparing control and *ARID1B* shRNA. Statistical significance was assessed by Wilcoxon rank sum test. **j**, Chromatin accessibility around the *IAPP* genomic region of SC-β cells, showing increased peak signals with *ARID1B* shRNA. EC, enterochromaffin; ns, not significant.

# Reporting Summary

## Statistics

For all statistical analyses, confirm that the following items are present in the figure legend, table legend, main text, or Methods section.

| n/a | Confirmed | |
|---|---|---|
| ☐ | ☒ | The exact sample size (*n*) for each experimental group/condition, given as a discrete number and unit of measurement |
| ☐ | ☒ | A statement on whether measurements were taken from distinct samples or whether the same sample was measured repeatedly |
| ☐ | ☒ | The statistical test(s) used AND whether they are one- or two-sided<br>*Only common tests should be described solely by name; describe more complex techniques in the Methods section.* |
| ☒ | ☐ | A description of all covariates tested |
| ☐ | ☒ | A description of any assumptions or corrections, such as tests of normality and adjustment for multiple comparisons |
| ☐ | ☒ | A full description of the statistical parameters including central tendency (e.g. means) or other basic estimates (e.g. regression coefficient) AND variation (e.g. standard deviation) or associated estimates of uncertainty (e.g. confidence intervals) |
| ☐ | ☒ | For null hypothesis testing, the test statistic (e.g. *F*, *t*, *r*) with confidence intervals, effect sizes, degrees of freedom and *P* value noted<br>*Give P values as exact values whenever suitable.* |
| ☒ | ☐ | For Bayesian analysis, information on the choice of priors and Markov chain Monte Carlo settings |
| ☒ | ☐ | For hierarchical and complex designs, identification of the appropriate level for tests and full reporting of outcomes |
| ☐ | ☒ | Estimates of effect sizes (e.g. Cohen's *d*, Pearson's *r*), indicating how they were calculated |

*Our web collection on statistics for biologists contains articles on many of the points above.*

## Software and code

Policy information about availability of computer code

| Data collection | Single-cell sequencing data was collected using the 10X Chromium and Illumina NovaSeq 6000. Flow cytometry data was collected using the LSR Fortessa and BD FACSDiva software. qPCR data was collected from the QuantStudio6 Pro using Design & Analysis 2.6.0. Fluorescent images were collected from the Zeiss Cell Discoverer confocal 7. |
|---|---|
| Data analysis | Single-cell multiomics sequencing data was analyzed using Cell Ranger ARC v2.0, and R studio 1.3.1093 (R v 4.03). We used packages Seurat 4.01, Signac 1.3.0, Monocle3 1.0, and chromVAR 1.12.0 . Flow cytometry was analyzed using FlowJo v10.8.1. |

For manuscripts utilizing custom algorithms or software that are central to the research but not yet described in published literature, software must be made available to editors and reviewers. We strongly encourage code deposition in a community repository (e.g. GitHub). See the Nature Portfolio guidelines for submitting code & software for further information.

## Data

Policy information about availability of data

All manuscripts must include a data availability statement. This statement should provide the following information, where applicable:
- Accession codes, unique identifiers, or web links for publicly available datasets
- A description of any restrictions on data availability
- For clinical datasets or third party data, please ensure that the statement adheres to our policy

Sequencing data that support the findings of this study have been deposited in the Gene Expression Omnibus (GEO) under accession code GSE199636. Source data

## Human research participants

Policy information about studies involving human research participants and Sex and Gender in Research.

| | |
|---|---|
| Reporting on sex and gender | Three male and one female donor for human islets were used. |
| Population characteristics | Donors for human islets ranged from 19-31.6 BMI and 18-68 years of age. |
| Recruitment | Prodo laboratories were involved in the recruitment of donors. We were not part of this process. |
| Ethics oversight | Prodo laboratories were involved in the recruitment of donors. We were not part of this process. |

Note that full information on the approval of the study protocol must also be provided in the manuscript.

# Field-specific reporting

Please select the one below that is the best fit for your research. If you are not sure, read the appropriate sections before making your selection.

☒ Life sciences    ☐ Behavioural & social sciences    ☐ Ecological, evolutionary & environmental sciences

For a reference copy of the document with all sections, see nature.com/documents/nr-reporting-summary-flat.pdf

# Life sciences study design

All studies must disclose on these points even when the disclosure is negative.

| | |
|---|---|
| Sample size | Sample sizes for single-nuclei multiome were determined based on 10x genomics recommended instructions. Target sequencing cell number was 10000, unless samples have lower cell count. Sample sizes in other experiments were based on previous experiences and studies with citations included in this study. No statistical methods were used to pre-determine sample sizes but our sample sizes are similar to those reported in previous publications (Pagliuca et al., Veres et al., Augsornworawat et al., Chiou et al., Hogrebe et al.). |
| Data exclusions | In single-nuclei mutiomics sequencings, low quality sequenced cells were removed with low RNA counts (nCount_RNA < 1000) and low ATAC counts (nCount_ATAC < 1000); excessively high RNA counts (ranging > 40000 − 50000) and excessively high ATAC counts (ranging > 40000 − 50000); nucleosome signal > 1.25, and TSS enrichment < 2. Mouse cells were excluded by cells expressing TTC36. |
| Replication | All results presented were replicated across multiple independent differentiations at least 3 times. Human islet datasets were obtained from different donors. In vitro experiments have been validated and performed by multiple personnels. Transplant datasets came from multiple mice. |
| Randomization | Randomization is not applicable in this study because there are no human participants and no clinical trials. |
| Blinding | Data collection and analysis were not performed blind to the conditions of the experiments. Blinding was not used in this study because there are no human participants that is subject to bias. |

# Reporting for specific materials, systems and methods

We require information from authors about some types of materials, experimental systems and methods used in many studies. Here, indicate whether each material, system or method listed is relevant to your study. If you are not sure if a list item applies to your research, read the appropriate section before selecting a response.

## Materials & experimental systems

| n/a | Involved in the study |
|---|---|
| ☐ | ☒ Antibodies |
| ☐ | ☒ Eukaryotic cell lines |
| ☒ | ☐ Palaeontology and archaeology |
| ☐ | ☒ Animals and other organisms |
| ☒ | ☐ Clinical data |
| ☒ | ☐ Dual use research of concern |

## Methods

| n/a | Involved in the study |
|---|---|
| ☒ | ☐ ChIP-seq |
| ☐ | ☒ Flow cytometry |
| ☒ | ☐ MRI-based neuroimaging |

# Antibodies

| | |
|---|---|
| Antibodies used | C-peptide (DSHB GN-ID4-S; 1:300)<br>RRID: AB_2255626<br>GN-ID4 was deposited to the DSHB by Madsen, O.D. (DSHB Hybridoma Product GN-ID4)<br><br>IAPP (Sigma Aldrich; PA5-84142; 1:300)<br>RRID: AB_2806862<br><br>NKX6-1 (DSHB; F55A12-S; 1:100)<br>RRID: AB_532379<br>F55A12 was deposited to the DSHB by Madsen, O.D. (DSHB Hybridoma Product F55A12)<br><br>SLC18A1 (Sigma Aldrich; HPA063797;  1:300)<br>Manufacturer validation: immunohistochemistry. Additional validation by the Human Protein Atlas (HPA) project.<br><br>Secondary Antibodies Company Part #<br>Anti-rat alexa fluor 488 (Invitrogen, cat. no. A21208; RRID: AB_141709)<br>Anti-rabbit alexa fluor 488 (Invitrogen, cat. no. A21206; RRID: AB_2535792)<br>Anti-mouse alexa fluor 594 (Invitrogen, cat. no. A21203; RRID: AB_141633)<br>Anti-rabbit alexa fluor 647 (Invitrogen, cat. no. A31573; RRID: AB_2536183)<br>Anti-rat PE (Jackson ImmunoResearch, cat. no. 712-116-153; RRID: AB_2340657) |
| Validation | All antibodies used have been validated on the supplier web page and used previously in other literatures. (Hogrebe et al., Nature 169 2021; Augsornworawat et al., Cell reports 32 8 2020; Veres et al., Nature 569 7756 2019; Pagliuca et al., Cell 159 2 2014) |

# Eukaryotic cell lines

Policy information about cell lines and Sex and Gender in Research

| | |
|---|---|
| Cell line source(s) | HUES8 hESC line was provided by Douglas Melton (Harvard University)<br>H1 hESC line was provided by Lindy Barrett (Broad Institute) with permission from WiCell |
| Authentication | All lines have been authenticated with DNA fingerprinting. |
| Mycoplasma contamination | Hues 8 and H1 hESC  tested negative  for mycoplasma as tested by Washington Universicty GEiC. |
| Commonly misidentified lines<br>(See ICLAC register) | No cell lines listed used. |

# Animals and other research organisms

Policy information about studies involving animals; ARRIVE guidelines recommended for reporting animal research, and Sex and Gender in Research

| | |
|---|---|
| Laboratory animals | 7 week old, male, NOD.Cg-Prkdcscid Il2rgtm1wjl/SzJ (NSG) mice (Jackson Laboratories; 005557). Mice were housed in an ambient facility with 30-70% humidity and a 12-hr light/dark cycle and were fed a chow diet. |
| Wild animals | No wild animals used in this study. |
| Reporting on sex | In vivo experiments involved male mice only. |
| Field-collected samples | No field-collected samples in this study. |
| Ethics oversight | Animal studies were performed by unblinded individuals in accordance with the Washington University International Care and Use Committee (IACUC) guidelines (Approval 21-0240). |

Note that full information on the approval of the study protocol must also be provided in the manuscript.

# Flow Cytometry

## Plots

Confirm that:

☒ The axis labels state the marker and fluorochrome used (e.g. CD4-FITC).

☒ The axis scales are clearly visible. Include numbers along axes only for bottom left plot of group (a 'group' is an analysis of identical markers).

☒ All plots are contour plots with outliers or pseudocolor plots.

☒ A numerical value for number of cells or percentage (with statistics) is provided.

## Methodology

| | |
|---|---|
| Sample preparation | Cells were single-cell dispersed by washing with PBS and adding 0.2 mL TrypLE/cm2 for 10minutes at 37C. Cells were fixed for 30 minutes in 4% PFA at 4C followed by incubation in ICC solution for 45 minutes in 4C. Primary antibodies were prepared in ICC solution and incubated on cells overnight at 4C. The following day, cells were washed with ICC solution and incubated for 2hr in secondary antibody at 4C. Cells were washed twice with PBS and filtered before running on an LSR Fortessa flow cytometer (BD Bioscience). FlowJo was used for analysis. |
| Instrument | LSR Fortessa flow cytometer (BD Bioscience) |
| Software | FlowJo v10.8.1 was used for analysis |
| Cell population abundance | Flow cytometry populations were greater than 30%. |
| Gating strategy | Gating strategy example is shown in Supplementary Information. |

☒ Tick this box to confirm that a figure exemplifying the gating strategy is provided in the Supplementary Information.

