## [Peer Review File · Nature Cell Biology]

Peer Review Information

Journal: Nature Cell Biology

Manuscript Title: Defining the chromatin and transcriptional landscape of stem cell-derived islets

Corresponding author name(s): Professor Jeffrey Millman

Editorial Notes:

Reviewer Comments & Decisions:

Decision Letter, initial version:
--

*Please delete the link to your author homepage if you wish to forward this email to co-authors.

Dear Professor Millman,

Your manuscript, "Defining the chromatin and transcriptional landscape of stem cell-derived islets", has now been seen by 3 referees, who are experts in beta cell biology, stem cell-derived islets, scRNAseq (referee 1); human pancreatic islets, chromatin, ATACseq (referee 2); and pancreatic islets, stem cell-derived beta cells, scRNAseq (referee 3). As you will see from their comments (attached below) they find this work of potential interest, but have raised substantial concerns, which in our view would need to be addressed with considerable revisions before we can consider publication in Nature Cell Biology.

Nature Cell Biology editors discuss the referee reports in detail within the editorial team, including the chief editor, to identify key referee points that should be addressed with priority, and requests that are overruled as being beyond the scope of the current study. To guide the scope of the revisions, I have listed these points below. I should stress that the referees' concerns point to a premature dataset and these points would need to be addressed with experiments and data, and reconsideration of the study for this journal and re-engagement of referees would depend on strength of these revisions.

In particular, it would be essential to:

(A) Ensure your genome-wide data become publicly available and accessible to the referees, so that they can properly evaluate your datasets.

(B) Increase the amount of donors used to generate the dataset and clarify the number of replicates for SC-islets and number of cells used for the different analyses, as indicated by referee #2:

"The approach is interesting, but the study should use human islets from a larger number of human donors to make it a valuable resource and it is also difficult to follow the number of replicates for SC-islets and number of cells used for different analyses throughout the manuscript".

"...this study only includes pancreatic islets from 2 human donors, which is a major limitation since there is variability between human donors due to age, sex, therapy etc and are these 2 donors representative for the stem cell donors".

(C) Provide a better description of the methods used.

Referee #1 says:

"I would appreciate a more detailed description of the bioinformatics analysis of the multiome data in the methods part as this is quite novel territory and it deserves a more detailed description. Recent efforts (for example see here: PMID: 35761403) use deep-learning algorithms for the analysis of multiomics datasets from different sequencing batches and it is not clear in the manuscript if additional methods to Seurat was used especially for integration of the data and batch corrections. Before publication the methods part of the work needs to be comprehensively described".

Referee #2 says:

"Overall, a better description of their methods and analyses should be provided. What exact input was used for respective analysis? Currently, one needs to guess a lot".

"The authors use Monocle of data presented in sup table 5 for analysis but this is not described in the manuscript".

"For data presented in sup table 6, how many cells were included in each SC-beta cell subpopulation. What exact comparison was done for the data in sup table 6, 4 different subgroups seem to be compared?"

"Human islets were provided by Prodo. In the methods on page 24, I miss important information about number of donors, including their BMI, use of pharmacotherapy, and days in culture. I found some information in Sup table 2.2 but this should be clearer in the paper".

"Throughout the method section, please include what cells type and species was used for each method. For example, in the "Multiomics sequencing analysis workflow" all of a sudden, the following statement occurs "In addition, mouse cells were removed by removing cells showing high..." And it is not very clear to the reader why the authors write about mice when one expects human cells, but it is not at all mentioned in this section. It is possible to understand further down, but overall a better method section is desirable".

"In the real-time PCR section and sup table 1.7, it is unclear which endogenous control was used? Multiple once? Did the authors test if endogenous controls were regulated by the comparison they studied?"

"The mouse transplantation experiments need further/better description throughout the manuscript (results, legends and methods), how many mice were used (3?). Sup table 2 gives some information that suggests 3 mice, and gives a cell number after filtering, but overall, it is difficult to follow numbers and methods".

Referee #3 says:

"It is unclear how S6 stage cells can be maintained in culture for 12 months. Whether they have been passaged? How often was the passage? The information was not found in the text or methods".

(D) Provide a better description of the statistics and replicates used, as indicated by referee #2:

"I miss numbers and replicates throughout the result section and also in the legends (only included in a few places). It is difficult to find numbers used for each experimental part".

"In each legend to both figures and sup tables please include what statistical method has been used and numbers compared (it is only done in a few cases). In particular, sup table 3 needs better explanation. How did the authors correct for multiple testing? The order of the sup tables needs to be corrected in the text. Also clarify abbreviations in tables".

(E) Add a 4-week time point in your analyses, as indicated by referee #3:

"Figure 5, the authors compared the 6-month transplanted SC-islets to those before transplantation, and the data suggest that transplanted SC-islets were more similar to their primary cell counterparts. How about the comparison between 4-week transplanted SC-islets to those before transplantation? As 4-week in vitro at S6 is the prime time for SC- β cells, this may have different results. The authors stated on p. 16 that, in SC- β cells, expression of many β -cell transcription factors and their associated motifs were upregulated at week 4 of in vitro culture, but most of these motifs and many transcripts decreased in the long term.

Adding a 4-week time point will also illustrate the changes in transcriptomic/chromatin accessibility during SC-islet maturation in vivo".

(F) Enhance the impact of the dataset by highlighting functionality issues and expanding your analyses, as indicated by referee #1:

"On that note and since the EC cell population might have some function during islet cell differentiation, can the authors assess the functionality of this population during in vitro stem cell differentiation? Is there an ELISA that can detect serotonin to measure serotonin production in these preparations?"

"It seems that pancreatic progenitors are not detectable in this multiome approach even following the 2 weeks of stage 6 differentiation. Since all the resources are available could the authors assess the multiome profile of key endocrinogenesis transcription factors at this stage, for example NEUROG3-FEV-HNF1 etc"

"An interesting point in chromatin accessibility and CHIP-Seq studies are the definition of the promoter region. Promoters are always classified kind of arbitrarily and can differ between studies (as it is impossible to verify promoters for all genes experimentally). In this manuscript why was 2kb upstream of TSS chosen? Additionally, can the ATAC-Seq data of the present study be used for enhancer-promoter interaction prediction or simply enhancer biology? Enhancer data could really broaden the significance of the current resource to include more information on epigenetic regulation of cell fates".

(G) All other referee concerns pertaining to strengthening existing data, providing controls, methodological details, clarifications and textual changes, should also be addressed.

(H) Finally, please pay close attention to our guidelines on statistical and methodological reporting (listed below) as failure to do so may delay the reconsideration of the revised manuscript. In particular please provide:

We would be happy to consider a revised manuscript that would satisfactorily address these points, unless a similar paper is published elsewhere, or is accepted for publication in Nature Cell Biology in the meantime.

- ensure that it conforms to our format instructions and publication policies (see below and www.nature.com/nature/authors/).

- provide a point-by-point rebuttal to the full referee reports verbatim, as provided at the end of this letter.

- provide the completed Editorial Policy Checklist (found here <https://www.nature.com/authors/policies/Policy.pdf>), and Reporting Summary (found here <https://www.nature.com/authors/policies/ReportingSummary.pdf>). This is essential for reconsideration of the manuscript and these documents will be available to editors and referees in the event of peer review. For more information see <http://www.nature.com/authors/policies/availability.html> or contact me.

Nature Cell Biology is committed to improving transparency in authorship. As part of our efforts in this

direction, we are now requesting that all authors identified as 'corresponding author' on published papers create and link their Open Researcher and Contributor Identifier (ORCID) with their account on the Manuscript Tracking System (MTS), prior to acceptance. ORCID helps the scientific community achieve unambiguous attribution of all scholarly contributions. You can create and link your ORCID from the home page of the MTS by clicking on 'Modify my Springer Nature account'. For more information please visit www.springernature.com/orcid.

[Redacted]

We would like to receive a revised submission within six months. We would be happy to consider a revision even after this timeframe, however if the resubmission deadline is missed and the paper is eventually published, the submission date will be the date when the revised manuscript was received.

We hope that you will find our referees' comments, and editorial guidance helpful. Please do not hesitate to contact me if there is anything you would like to discuss.

Best wishes,

Stelios

Stylios Lefkopoulos, PhD
He/him/his
Associate Editor
Nature Cell Biology
Springer Nature
Heidelberger Platz 3, 14197 Berlin, Germany

E-mail: stylios.lefkopoulos@springernature.com

Twitter: @s_lefkopoulos

Reviewers' Comments:

Reviewer #1:

Remarks to the Author:

Review for manuscript: "Defining the chromatin and transcriptional landscape of stem cell-derived islets"

A: Summary of key results: The work from Augsornworawat et al uses a multiomics approach (single-cell RNA-Seq and single-cell ATAC-Seq) to characterize the differentiation protocol of hESCs towards beta-cells to be used for cell replacement therapy. The work uses both stem cell-derived (SC)-islets,

human primary islets and SC-islets following transplantation and maturation in mice. The work builds on the initial observations of the resource and goes as far as to validate certain transcription factors on cell fate decisions in vitro including CTCF and ARID1B genetic manipulations.

B: Originality and significance: Overall, this study is an important resource for the beta-cell differentiation field opening new avenues for exploiting the dataset for increasing beta-cell maturation from stem-cell preparations. The multiomics characterization and the number of datasets reported are results of a great amount of resources and work from the authors. Regarding novelty, this is one of the first two studies employing a multiomics approach to study beta-cell differentiation from hESC, the second being published as a preprint very recently:

<https://www.biorxiv.org/content/10.1101/2022.09.25.509355v1>. Both studies advance the field and report similar findings. In fairness to the authors of the preprint article, their work captures more differentiation stages of the hESC differentiation protocol, giving the opportunity for more in-depth characterization of the process and integrate data from both childhood and adult human donors giving a more comprehensive view of the human islet multiome. On the other hand, the novelty of the current manuscript is the multiome integration of the prolonged maturation steps of the islet in vitro and after transplantation in mice. Another important biological advancement of this manuscript stemming from measuring transcriptome and chromatin accessibility of the same cell is the reporting of the transcription factor multiome data between the different datasets that can lead to advancements in the maturation of cell types in future studies.

C: Data and methodology: The authors use a standard workflow for sequencing and bioinformatics analysis of their multiple datasets. I would appreciate a more detailed description of the bioinformatics analysis of the multiome data in the methods part as this is quite novel territory and it deserves a more detailed description. Recent efforts (for example see here: PMID: 35761403) use deep-learning algorithms for the analysis of multiomics datasets from different sequencing batches and it is not clear in the manuscript if additional methods to Seurat was used especially for integration of the data and batch corrections. Before publication the methods part of the work needs to be comprehensively described.

I could not access the raw data of the study (GEO database accession) which are locked by the authors so I cannot comment on that. The data are presented clearly in the figures and the figure legends are informative to follow the logic of the presentation.

Some suggestions for this section:

- Typically dead cells are removed computationally based on mitochondrial content but I do not see such filtering in the methods section. Did the authors not use such filtering and if so why?
- Trajectory analysis on Figure 2a was done using Monocle 3 but it would be worth trying different trajectory analysis tools as they can give vastly different results.
- Human donor information: Are there any additional information regarding the donor's clinical characteristics that might affect the sequencing results (medication etc)? Also how were the islets treated? Were they cultured prior to dissociation and for how long? This can be important information as efforts are made to standardize processes benefiting human islet biology (see commentary here: PMID: 35953581). Also the ethical approval information is not mentioned in the manuscript and it would be important point to clarify that and if there was written consent from donors.

Appropriate use of statistics and treatment of uncertainties: Statistical tests are reported in the manuscript as requested.

Conclusions: As a resource paper, the conclusions suggested by the authors are based on their reported data. Suggested improvements to strengthen the conclusions are presented in the following section.

Suggested improvements:

Main points to improve

1. The second biorxiv manuscript with similar approach mentioned above shows that enterochromaffin cells are actually a human embryonic islet population characterized by CDX2 transcription factor and it is not a by-product of the differentiation protocol. This data needs to be discussed in the discussion section of the paper to put the work of this resource in the perspective of the field and how the multiome data can further study the function of this population during development.
2. On that note and since the EC cell population might have some function during islet cell differentiation, can the authors assess the functionality of this population during in vitro stem cell differentiation? Is there an ELISA that can detect serotonin to measure serotonin production in these preparations?
3. In the CTCF experiment, it is a bit unclear what the control condition refers to. Is it treated with doxycycline or untreated? Doxycycline is shown to affect mitochondrial biology and it would be good to include such control in non-CAS9 hESC and check at least the insulin secretion and beta vs EC differentiation phenotypes.
4. Since the paper is a resource effort it would be nice to have an online resource that the data can be accessed easily by the diabetes community, if possible.
5. It seems that pancreatic progenitors are not detectable in this multiome approach even following the 2 weeks of stage 6 differentiation. Since all the resources are available could the authors assess the multiome profile of key endocrinogenesis transcription factors at this stage, for example NEUROG3-FEV-HNF1 etc.
6. An interesting point in chromatin accessibility and CHIP-Seq studies are the definition of the promoter region. Promoters are always classified kind of arbitrarily and can differ between studies (as it is impossible to verify promoters for all genes experimentally). In this manuscript why was 2kb upstream of TSS chosen? Additionally, can the ATAC-Seq data of the present study be used for enhancer-promoter interaction prediction or simply enhancer biology? Enhancer data could really broaden the significance of the current resource to include more information on epigenetic regulation of cell fates.
6. Since the end goal of this approach for diabetes therapies would be to differentiate iPSCs to beta-cells and the authors have elegantly showed with their improved protocol and increased differentiation efficiency of iPSC to beta-cells, would it possible to perform multiomics characterization of an iPSC-derived sc-islets? That would greatly benefit the significance of this resource for the community, if resources and time allow for such big experiment.

Minor points

1. Proliferating a-cell as cluster might require more stringent clustering parameters. Are there not proliferating beta-cells or delta cells? Why is this so distinct?
2. In vivo transplantation of sc-islets improved maturation identity but not cell type composition. Why is that? Is the multiomics characterization a more robust picture of the differentiation status compared to traditional sc-RNA-seq and immunostaining methodologies?
3. Cluster proximity on the 2-dimensional plot does not mean lineage proximity. There is an ongoing debate on this issue and it would be important to rephrase or justify using computational explanation why the authors think that the 2-dimensional space reflects the closeness of the multidimensionality of the data. Trajectory analysis is more suited to answer such questions.

References: - Format the reference list consistently as sometimes the manuscript gives the abbreviations for the journals and some time the full names in certain cases misspelled.

- Reference 7 and 48 is the same paper. Please check the reference list for further inconsistencies.

- Reference the papers sorting for GP2+ cells at progenitor stage together with the 8-12 references for completion in the Introduction. That is reference 53 of the current manuscript and include the Cell Reports work form Henrik Semb's laboratory for completion: PMID: 28380361

- Ackermann et al paper is missing from the reference and the discussion. It should be included in the manuscript as it was one of the first to use bulk ATAC and RNA-Seq to study human endocrine populations PMID: 26977395

Clarity and context: The paper is clearly written and conveys the main points of the work. A minor comment:

Page 11: Problem with sentence structure starting with Unlike SC-islets. Please re-phrase.

Reviewer #2:

Remarks to the Author:

Punn Augsornworawat et al used sc/sn-RNA seq combined with sc/sn-ATAC seq (10X) to study stem cell-derived islets and compared these with results using the same methods for human islets from 2 donors. The approach is interesting, but the study should use human islets from a larger number of human donors to make it a valuable resource and it is also difficult to follow the number of replicates for SC-islets and number of cells used for different analyses throughout the manuscript. Overall, a better description of their methods and analyses should be provided. What exact input was used for respective analysis? Currently, one needs to guess a lot. Specific limitations and comments are found below.

1. I miss numbers and replicates throughout the result section and also in the legends (only included in a few places). It is difficult to find numbers used for each experimental part.
2. Page 6, result section: "Each of the identified populations had high expression of genes traditionally associated with their cell type (Fig. 1c)." This statement expects the reader to know what genes are traditionally expressed in different cell types?
3. In each legend to both figures and sup tables please include what statistical method has been used and numbers compared (it is only done in a few cases). In particular, sup table 3 needs better explanation. How did the authors correct for multiple testing? The order of the sup tables needs to be corrected in the text. Also clarify abbreviations in tables.
4. The authors use Monocle of data presented in sup table 5 for analysis but this is not described in the manuscript.
5. For data presented in sup table 6, how many cells were included in each SC-beta cell subpopulation. What exact comparison was done for the data in sup table 6, 4 different subgroups seem to be compared?
6. Human islets were provided by Prodo. In the methods on page 24, I miss important information about number of donors, including their BMI, use of pharmacotherapy, and days in culture. I found some information in Sup table 2.2 but this should be clearer in the paper. Also, this study only includes pancreatic islets from 2 human donors, which is a major limitation since there is variability between human donors due to age, sex, therapy etc and are these 2 donors representative for the stem cell donors?
7. Throughout the method section, please include what cells type and species was used for each method. For example, in the "Multiomics sequencing analysis workflow" all of a sudden, the following

statement occurs “In addition, mouse cells were removed by removing cells showing high...” And it is not very clear to the reader why the authors write about mice when one expects human cells, but it is not at all mentioned in this section. It is possible to understand further down, but overall a better method section is desirable.

8. In the real-time PCR section and sup table 1.7, it is unclear which endogenous control was used? Multiple once? Did the authors test if endogenous controls were regulated by the comparison they studied?

9. The mouse transplantation experiments need further/better description throughout the manuscript (results, legends and methods), how many mice were used (3?). Sup table 2 gives some information that suggests 3 mice, and gives a cell number after filtering, but overall, it is difficult to follow numbers and methods.

Reviewer #3:

Remarks to the Author:

Augsornworawat et al. performed multiomic assessments on SC- β cells using integrated sc-ATACseq and sc-RNAseq. They compared the chromatin accessibility between SC- β and SC-EC, SC- β and primary β , and long-term cultured cell populations and their post-transplantation counterparts. This approach improved the resolution of single cell transcriptome and revealed new signature genes in each population. In addition, they provided two examples, with signature genes being modulated. Overexpression of CTCF, a signature of SC-EC, during stage 5 leads to drastic reductions in the expression of β -cell identity genes. Knockdown of ARID1B, whose expression is lower in primary beta and long-term culture SC- β , leads to increased SC- β cell yields.

This is the first sc-ATACseq of SC-beta cells. My only criticism is that the conclusions reached in this study are mostly expected. Nevertheless, I appreciate the thoroughness of the investigation. Data revealed in this study provide a valuable resource to identify deficiencies in the chromatin and transcriptional landscape of SC-islet cell types, which may contribute to further improving the current ES- β differentiation protocol.

Specific

1, Figure 5, the authors compared the 6-month transplanted SC-islets to those before transplantation, and the data suggest that transplanted SC-islets were more similar to their primary cell counterparts. How about the comparison between 4-week transplanted SC-islets to those before transplantation? As 4-week in vitro at S6 is the prime time for SC- β cells, this may have different results. The authors stated on p. 16 that, in SC- β cells, expression of many β -cell transcription factors and their associated motifs were upregulated at week 4 of in vitro culture, but most of these motifs and many transcripts decreased in the long term.

Adding a 4-week time point will also illustrate the changes in transcriptomic/chromatin accessibility during SC-islet maturation in vivo.

2, It is unclear how S6 stage cells can be maintained in culture for 12 months. Whether they have been passaged? How often was the passage? The information was not found in the text or methods.

FINANCIAL AND NON-FINANCIAL COMPETING INTERESTS – the authors must include one of three declarations: (1) that they have no financial and non-financial competing interests; (2) that they have financial and non-financial competing interests; or (3) that they decline to respond, after the Author Contributions section. This statement will be published with the article, and in cases where financial

and non-financial competing interests are declared, these will be itemized in a web supplement to the article. For further details please see <https://www.nature.com/licenceforms/nrg/competing-interests.pdf>.

Methods should be written concisely, but should contain all elements necessary to allow interpretation and replication of the results. As a guideline, Methods sections typically do not exceed 3,000 words. The Methods should be divided into subsections listing reagents and techniques. When citing previous methods, accurate references should be provided and any alterations should be noted. Information must be provided about: antibody dilutions, company names, catalogue numbers and clone numbers for monoclonal antibodies; sequences of RNAi and cDNA probes/primers or company names and catalogue numbers if reagents are commercial; cell line names, sources and information on cell line identity and authentication. Animal studies and experiments involving human subjects must be reported in detail, identifying the committees approving the protocols. For studies involving human subjects/samples, a statement must be included confirming that informed consent was obtained. Statistical analyses and information on the reproducibility of experimental results should be provided in a section titled "Statistics and Reproducibility".

All Nature Cell Biology manuscripts submitted on or after March 21 2016 must include a Data availability statement at the end of the Methods section. For Springer Nature policies on data availability see <http://www.nature.com/authors/policies/availability.html>; for more information on this particular policy see <http://www.nature.com/authors/policies/data/data-availability-statements-data-citations.pdf>. The Data availability statement should include:

- Accession codes for primary datasets (generated during the study under consideration and designated as "primary accessions") and secondary datasets (published datasets reanalysed during the study under consideration, designated as "referenced accessions"). For primary accessions data should be made public to coincide with publication of the manuscript. A list of data types for which submission to community-endorsed public repositories is mandated (including sequence, structure, microarray, deep sequencing data) can be found here <http://www.nature.com/authors/policies/availability.html#data>.
- Unique identifiers (accession codes, DOIs or other unique persistent identifier) and hyperlinks for datasets deposited in an approved repository, but for which data deposition is not mandated (see here for details <http://www.nature.com/sdata/data-policies/repositories>).

- At a minimum, please include a statement confirming that all relevant data are available from the authors, and/or are included with the manuscript (e.g. as source data or supplementary information), listing which data are included (e.g. by figure panels and data types) and mentioning any restrictions on availability.
- If a dataset has a Digital Object Identifier (DOI) as its unique identifier, we strongly encourage including this in the Reference list and citing the dataset in the Methods.

We recommend that you upload the step-by-step protocols used in this manuscript to the Protocol Exchange. More details can found at www.nature.com/protocolexchange/about.

All imaging data should be accompanied by scale bars, which should be defined in the legend. Cropped images of gels/blots are acceptable, but need to be accompanied by size markers, and to retain visible background signal within the linear range (i.e. should not be saturated). The boundaries of panels with low background have to be demarked with black lines. Splicing of panels should only be considered if unavoidable, and must be clearly marked on the figure, and noted in the legend with a statement on whether the samples were obtained and processed simultaneously. Quantitative comparisons between samples on different gels/blots are discouraged; if this is unavoidable, it should only be performed for samples derived from the same experiment with gels/blots were processed in parallel, which needs to be stated in the legend.

The total number of Supplementary Figures (not including the “unprocessed scans” Supplementary Figure) should not exceed the number of main display items (figures and/or tables (see our Guide to Authors and March 2012 editorial <http://www.nature.com/ncb/authors/submit/index.html#suppinfo>; <http://www.nature.com/ncb/journal/v14/n3/index.html#ed>). No restrictions apply to Supplementary Tables or Videos, but we advise authors to be selective in including supplemental data.

GUIDELINES FOR EXPERIMENTAL AND STATISTICAL REPORTING

REPORTING REQUIREMENTS – To improve the quality of methods and statistics reporting in our papers we have recently revised the reporting checklist we introduced in 2013. We are now asking all life sciences authors to complete two items: an Editorial Policy Checklist (found here <https://www.nature.com/authors/policies/Policy.pdf>) that verifies compliance with all required editorial policies and a reporting summary (found here <https://www.nature.com/authors/policies/ReportingSummary.pdf>) that collects information on experimental design and reagents. These documents are available to referees to aid the evaluation of the manuscript. Please note that these forms are dynamic ‘smart pdfs’ and must therefore be downloaded and completed in Adobe Reader. We will then flatten them for ease of use by the reviewers. If you would like to reference the guidance text as you complete the template, please access these flattened versions at <http://www.nature.com/authors/policies/availability.html>.

Author Rebuttal to Initial comments

Reviewer #1:

[Reviewers comments in black]

[Author comments in dark blue]

We thank the reviewer for the feedback.

[Reviewer]

Remarks to the Author:

Review for manuscript: "Defining the chromatin and transcriptional landscape of stem cell-derived islets"

A: Summary of key results: The work from Augsornworawat et al uses a multiomics approach (single-cell RNA-Seq and single-cell ATAC-Seq) to characterize the differentiation protocol of hESCs towards beta-cells to be used for cell replacement therapy. The work uses both stem cell-derived (SC)-islets, human primary islets and SC-islets following transplantation and maturation in mice. The work builds on the initial observations of the resource and goes as far as to validate certain transcription factors on cell fate decisions in vitro including CTCF and ARID1B genetic manipulations.

B: Originality and significance: Overall, this study is an important resource for the beta-cell differentiation field opening new avenues for exploiting the dataset for increasing beta-cell maturation from stem-cell preparations. The multiomics characterization and the number of datasets reported are results of a great amount of resources and work from the authors. Regarding novelty, this is one of the first two studies employing a multiomics approach to study beta-cell differentiation from hESC, the second being published as a preprint very recently: <https://www.biorxiv.org/content/10.1101/2022.09.25.509355v1>. Both studies advance the field and report similar findings. In fairness to the authors of the preprint article, their work

captures more differentiation stages of the hESC differentiation protocol, giving the opportunity for more in-depth characterization of the process and integrate data from both childhood and adult human donors giving a more comprehensive view of the human islet multiome. On the other hand, the novelty of the current manuscript is the multiome integration of the prolonged maturation steps of the islet in vitro and after transplantation in mice. Another important biological advancement of this manuscript stemming from measuring transcriptome and chromatin accessibility of the same cell is the reporting of the transcription factor multiome data between the different datasets that can lead to advancements in the maturation of cell types in future studies.

[Authors]

We now include this citation and say:

“We expect that future comparisons of the data presented here and other multiomic approaches⁶⁰ will produce novel insights into the nature of SC-islet identity and biology.”

Also, we now state in the discussion:

“ Interestingly, another group ⁶⁰ in a recent pre-print using a different single-cell multiomic approach suggested that SC-EC cells resemble a pre- β cell population in the pancreas rather than the enterochromaffin cells found in the intestine. While we have observed reduced SC-EC and other endocrine cell type numbers after long term (>6 months) in vitro culture, our transplanted SC-islets retain a high percentage of the SC-EC cell population after 6 months in vivo. Furthermore, we did not detect strong traces of β cell features in this transplanted SC-EC population, suggesting that the SC-EC and SC- β cell populations become more distinct over time as they mature in vivo. As SC-EC cells may be detrimental to SC- β cell function ⁹, or at the least are unlikely to be providing a benefit for SC-islets in a diabetes cell replacement therapy, understanding how to reduce or eliminate this off-target cell population is most likely desired.”

We acknowledge that there are likely important comparisons to make in our study with that recent preprint. This current manuscript also has its own pre-print that precedes the noted preprint. As many important details, such as scientific conclusions, often change between prepublication and peer-reviewed publication, we believe that a more detailed comparison should occur once both have passed peer review and been formally published.

[Reviewer]

C: Data and methodology: The authors use a standard workflow for sequencing and bioinformatics analysis of their multiple datasets. I would appreciate a more detailed description of the bioinformatics analysis of the multiome data in the methods part as this is quite novel territory and it deserves a more detailed description. Recent efforts (for example see here: PMID: 35761403) use deep-learning algorithms for the analysis of multiomics datasets from

different sequencing batches and it is not clear in the manuscript if additional methods to Seurat was used especially for integration of the data and batch corrections. Before publication the methods part of the work needs to be comprehensively described.

[Authors]

We have added an a more detailed description of the bioinformatics analysis in the Methods section: Processing and filtering of multiomics sequencing data; Dataset normalization, integration, and assay build; Analysis of multiomics datasets; Trajectory analysis sections. These sections now describe all relevant information, including used packages, functions, methods, databases, and parameters. Additionally, supplemental table 2 now includes parameters used for filtering cells.

[Reviewer]

I could not access the raw data of the study (GEO databse accession) which are locked by the authors so I cannot comment on that. The data are presented clearly in the figures and the figure legends are informative to follow the logic of the presentation.

[Authors]

We have arranged for access of the raw data for the reviewer for the purposes of performing this review. Our datasets are available at <https://www.ncbi.nlm.nih.gov/geo/query/acc.cgi> under GSE199636. Reviewer access can be gained by using the following token: apwxsksqhknhcv. We will of course open this database to the public as soon as this study is published.

[Reviewer]

Some suggestions for this section:

- Typically dead cells are removed computationally based on mitochondrial content but I do not see such filtering in the methods section. Did the authors not use such filtering and if so why?

[Authors]

We have clarified this in the “Single-nuclei sample preparation and sequencing” section of the methods, saying:

“Cells collected were processed and delivered to the McDonnell Genome Institute at Washington University for library preparation and sequencing. Samples were processed by dispersing cells into a single-cell suspension using TrypLE for 10 min at 37°C and quantified for viability using the Vi-Cell XR (Beckman Coulter). Prior to proceeding, all samples were ensured to have >90% viability to minimize dead cell carry over in sequencing.”

Additionally, we have also clarified our cell filtering strategy in the “Processing and filtering of multiomic sequencing data” section of the methods, now saying:

“We attempted to remove dead cells from the dataset by excluding cells with very low RNA yields because the multiomics platform relies on single-nuclei, which does not contain valid cytoplasmic mitochondrial content. Low quality cells including doublets, dead cells, and poor sequencing depth cells were filtered by removing cells with low RNA counts ($nCount_RNA < 1000$) and low ATAC counts ($nCount_ATAC < 1000$); high RNA counts (ranging $> 40000 - 50000$) and high ATAC counts (ranging $> 40000 - 50000$); nucleosome signal > 1.25 , and TSS enrichment < 2 .”

We also now also include parameters used for filtering all datasets in Supplementary Table S2.3.

These are standard approaches for handling dead cells for single-nuclei sequencing.

[Reviewer]

-Trajectory analysis on Figure 2a was done using Monocle 3 but it would be worth trying different trajectory analysis tools as they can give vastly different results.

[Authors]

We added new trajectory analyses to Extended Data Fig. 2g using another trajectory analysis tool. This data is consistent with our findings using Monocle3. We now state in the results:

“Additionally, we performed trajectory analysis using another tool⁴⁴ to validate our observations and found that the dynamic expression of key SC-EC and SC- β cell genes were consistent with our previous analysis (Extended Data Fig. 2g).”

Our use of the monocle package highlights a valuable example on the application of cellular trajectory to assess and delineate non-exclusive populations that allow us to identify fate driving regulators, such as CTCF. As many common packages utilize mRNA alone, development and utilization of other packages that allows for evaluation of trajectories using both chromatin and mRNA information can greatly enhance findings and novel insights in other contexts.

[Reviewer]

- Human donor information: Are there any additional information regarding the donor’s clinical characteristics that might affect the sequencing results (medication etc)? Also how were the islets treated? Were they cultured prior to dissociation and for how long? This can be important information as efforts are made to standardize processes benefiting human islet biology (see commentary here: PMID: 35953581). Also the ethical approval information is not mentioned in

the manuscript and it would be important point to clarify that and if there was written consent from donors.

[Authors]

We now provide details on human islets donors in Supplemental Table 2 and added text to the methods to better reference the table. We have also expanded our methods in the “SC-islet and primary islet cell culture” section showing details for our human islet culture methods, now stating:

“Donor details, including BMI, health status, age, sex, and race can be found in Supplemental Table 2.2. Our study consists of 4 total donors. Upon arrival, islets were transferred into wells of a 6-well plate on an orbital shaker at 100 RPM and maintained with 4mL per well of CMRL 1066 Supplemented media (Corning; 99-603-CV) with 10% heat inactivated FBS (Gibco; 26140-079). Primary human islets were submitted for sequencing within 2 days after arrival.”

We also provide details on the human islet supplier including the ethical approval information, now stating:

“Primary human islets were acquired as clusters and shipped from Prodo Laboratories. These islets have been refused for human islet transplants and meet specific criteria for research use. Consent information can be found on their website (<https://prodolabs.com/human-islets-for-research>).”

[Reviewer]

Appropriate use of statistics and treatment of uncertainties: Statistical tests are reported in the manuscript as requested.

Conclusions: As a resource paper, the conclusions suggested by the authors are based on their reported data. Suggested improvements to strengthen the conclusions are presented in the following section.

Suggested improvements:

Main points to improve

1. The second biorxiv manuscript with similar approach mentioned above shows that enterochromaffin cells are actually a human embryonic islet population characterized by CDX2 transcription factor and it is not a by-product of the differentiation protocol. This data needs to be discussed in the discussion section of the paper to put the work of this resource in the perspective of the field and how the multiome data can further study the function of this population during development.

[Authors]

We now discuss this citation, saying:

“A prior study discovered that enterochromaffin cells are generated during SC-islet differentiations in vitro and suggested that SC-EC and SC- β cells are distinct populations that arise from the same pancreatic progenitor population⁹. Our use of Monocle3 analysis allowed us to obtain greater resolution of these cell identities by simultaneously analyzing both the transcriptional and chromatin state of each cell rather than mRNA alone. Other trajectory analysis tools that take account of both chromatin and mRNA⁴⁴, can facilitate, and lead new insights on cell fate decisions within SC-islet cell types and cellular maturation states using our datasets. Our data suggests that the SC-EC and SC- β cells generated from these directed differentiation protocols are not completely exclusive cell populations by the end of the in vitro differentiation protocol. Instead, they comprise a continuum of endocrine cell types that contain varying degrees of both enterochromaffin and β -cell features, suggesting that the current directed differentiation methodology is insufficient for fully specifying each endocrine cell type. Interestingly, increased DNA binding motif accessibility for the chromatin regulator CTCF was highly correlative with increased SC-EC cell features. CTCF-overexpression blocked differentiation to pancreatic endocrine and further redirected cell fate selection toward the enteroendocrine lineage, supporting the notion that chromatin accessibility is a major regulator of the fate decision between SC-EC and SC- β cells along this gradient. Interestingly, another group⁶⁰ in a recent pre-print using a different single-cell multiomic approach suggested that SC-EC cells resemble a pre- β cell population in the pancreas rather than the enterochromaffin cells found in the intestine. While we have observed reduced SC-EC and other endocrine cell type numbers after long term (>6 months) in vitro culture, our transplanted SC-islets retain a high percentage of the SC-EC cell population after 6 months in vivo. Furthermore, we did not detect strong traces of β cell features in this transplanted SC-EC population, suggesting that the SC-EC and SC- β cell populations become more distinct over time as they mature in vivo. As SC-EC cells may be detrimental to SC- β cell function⁹, or at the least are unlikely to be providing a benefit for SC-islets in a diabetes cell replacement therapy, understanding how to reduce or eliminate this off-target cell population is most likely desired.”

We are wary of conducting a more in-depth comparison to this preprint before it has gone through peer-review and without us having access to their sequencing data. The value of our and the other group's datasets will be best realized in the field after they have both fully refereed through peer-review and public accessibility of the relevant datasets. A big difference that we noticed between their and our approaches was that they separately performed single-cell RNA and single-cell ATAC sequencing and then integrating the results with certain assumptions. In contrast, our approach is to perform both single-cell RNA+ATAC sequencing in the same cell. We can't be clear about how these different approaches affect the data at this time.

[Reviewer]

2. On that note and since the EC cell population might have some function during islet cell differentiation, can the authors assess the functionality of this population during in vitro stem cell

differentiation? Is there an ELISA that can detect serotonin to measure serotonin production in these preparations?

[Authors]

We have added ELISA quantification of serotonin in Extended Data Figure 2A. In the text, we now say:

“The serotonin-producing SC-EC cells comprise an off-target cell population that arises during in vitro SC-islet differentiation protocols but that does not positively contribute to tissue function^{9, 41, 42} (Fig. 1b). Detection of significant amounts of serotonin occurred only at the end of the differentiation protocol (Extended Data Fig. 2a). While it is not known how serotonin affects in vitro differentiation to SC- β cells, a study has reported its involvement in regulating β cell mass during pregnancy⁴³.”

[Reviewer]

3. In the CTCF experiment, it is a bit unclear what the control condition refers to. Is it treated with doxycycline or untreated? Doxycycline is shown to affect mitochondrial biology and it would be good to include such control in non-CAS9 hESC and check at least the insulin secretion and beta vs EC differentiation phenotypes.

[Authors]

We have clarified the control in the text, saying: *“Control represents cells without doxycycline treatment.”* We now also state this in the figure captions.

We have added data investigating the effect of doxycycline on the cells in Extended Data Fig. 2g and h. In the text, we now say:

“By treating the differentiating CRISPRa cells with doxycycline in the absence of a guide RNA (gRNA), we observed no significant changes in identity nor function when compared to untreated control, ascertaining that doxycycline has no negative impact on differentiation nor function of SC-islets (Extended Data Fig. 2h and i).”

[Reviewer]

4. Since the paper is a resource effort it would be nice to have an online resource that the data can be accessed easily by the diabetes community, if possible.

[Authors]

The dataset will become publicly available after publication. We have also provided a template code for investigators to analyze these datasets. In the Data availability and code availability section of the text, we now state:

“Single-nuclei multiomic sequencing analysis and source data can be found in the supplementary tables. Fluorescent images for quantification can be found in Supplementary Figure 2. Individual flow cytometry data can be found in Supplementary 4. All sequencing datasets have been deposited in the Gene Expression Omnibus (GEO) – NCBI under accession number GSE199636.”

And

“Codes used for analyzing single-nuclei multiomic sequencing are available on <https://github.com/punnaug>.”

[Reviewer]

5. It seems that pancreatic progenitors are not detectable in this multiome approach even following the 2 weeks of stage 6 differentiation. Since all the resources are available could the authors assess the multiome profile of key endocrinogenesis transcription factors at this stage, for example NEUROG3-FEV-HNF1 etc.

[Authors]

We now have this analysis in Extended Data figure 1. In the text, we now say:

“In addition to exploring transcription factors associated with different endocrine cell identities, we also investigated transcription factors specifically implicated during endocrine cell development by RNA expression and motif accessibility (Extended Data Fig. 1h). Of note, we observed FEV being upregulated in the SC-EC population, HNF1A in all endocrine populations, and PDX1 expressed predominantly in SC-EC and SC-β cells by both RNA and chromatin motif accessibility. We also observed a mismatch of NEUROG3 in the final SC-β cell population, where there was no detectable RNA but enriched motif accessibility .”

This differentiation protocol does not have progenitor cells during stage 6 of the protocol. Stage 5 of the protocol is designed to differentiate pancreatic progenitors to endocrine cells.

[Reviewer]

6. An interesting point in chromatin accessibility and CHIP-Seq studies are the definition of the promoter region. Promoters are always classified kind of arbitrarily and can differ between studies (as it is impossible to verify promoters for all genes experimentally). In this manuscript why was 2kb upstream of TSS chosen? Additionally, can the ATAC-Seq data of the present study be used for enhancer-promoter interaction prediction or simply enhancer biology? Enhancer data could really broaden the significance of the current resource to include more information on epigenetic regulation of cell fates.

[Authors]

We have clarified why 2kb was chosen in the text, now saying:

“Promoter accessibilities were determined from ATAC information using “GeneActivity” by taking account of 2000 base pairs upstream of the transcription start site, as this is the default recommended parameter and commonly used in other single-nuclei ATAC studies^{91, 92}.”

In other words, we chose 2kb because that is the convention in the field and it appears to sufficiently cover the promoter regions of most genes (Stuart et al., Nature Methods 2021; Ma et al., Cell 183 4, 2020).

We expanded our promoter region analysis in Supplemental Table 4 by covering lists of differential promoter accessibilities in key cell types, which we hope to be a valuable resource.

We performed additional analyses looking at cis-regulatory elements that could potentially define epigenetic regulation of cell fates. We focused the analyses by assessing ATAC peak patterns in SC-beta and SC EC comparisons in Extended Data figure 2d. In the text, we now say:

“ATAC peaks around SC-β cell and SC-EC cell marker genes also highlight differences in chromatin accessibility patterns, some of which were predicted to be cis-regulatory elements (Extended Data Fig. 2d).”

We also assessed potential cis-regulatory elements relating to β cell identity, showing cis-regulatory elements in genes associated with maturation (MAFA, UCN3) in figure 4f. In the text, we now state:

“Further analysis of the chromatin accessibility of MAFA and UCN3 (Fig. 4f), which are known to be transcriptionally upregulated in β-cells^{9, 16, 50-52}, demonstrated that predicted cis-regulatory elements present around the MAFA and UCN3 gene regions differed considerably between SC-β and primary β-cells.”

[Reviewer]

6. Since the end goal of this approach for diabetes therapies would be to differentiate iPSCs to beta-cells and the authors have elegantly showed with their improved protocol and increased differentiation efficiency of iPSC to beta-cells, would it possible to perform multiomics characterization of an iPSC-derived sc-islets? That would greatly benefit the significance of this resource for the community, if resources and time allow for such big experiment.

[Authors]

We now discuss this, saying:

" Finally, SC-islets have been generated from patients with diabetes via induced pluripotent stem cells (iPSCs) for the study of diabetes pathology and consideration as an autologous cell source for β -cell replacement therapy ^{15, 84-89}. In all of these examples, a multiomic approach, including using the datasets presented here, could help determine how these modulations affect cell identity within SC-islets with greater resolution, facilitating finer control of differentiation parameters. Furthermore, additional multiomic sequencing of iPSC-derived SC-islets could elucidate crucial information on the importance of iPSC derivation methods and starting somatic cell types for SC-islet differentiation strategies."

We chose to focus on human embryonic stem (ES) cells because the trials by ViaCyte and Vertex are using this type of stem cell. Given that the current differentiation protocol produces SC-islets that are remarkably similar between ES and iPS cells, we felt that investigation of iPS cells would be better handled by a separate dedicated study to more rigorously investigate key relevant parameters, such as donor characteristics (e.g. healthy vs diabetic), derivation method (e.g. viral vs mRNA vs chemical), and starting somatic cell type (e.g. fibroblast vs PBMCs vs kidney epithelial).

[Reviewer]

Minor points

1. Proliferating α -cell as cluster might require more stringent clustering parameters. Are there not proliferating beta-cells or delta cells? Why is this so distinct?

[Authors]

In the text, we now state:

"Consistent with previous studies ^{15, 16, 34}, we also identified proliferating SC- α cells containing high transcript counts and promoter accessibilities for proliferating genes, including MKI67, PCNA, and MCM2, in addition to α -cells markers (Fig. 2c and Extended Data Fig. 1c)."

We and others have observed this population in prior single-cell RNA sequencing studies, and they are indeed quite distinct across parameters. We and others have not seen similar distinct proliferating beta or delta-cell populations. We do not know why the alpha cells are 'special' in this regard.

[Reviewer]

2. In vivo transplantation of sc-islets improved maturation identity but not cell type composition. Why is that? Is the multiomics characterization a more robust picture of the differentiation status compared to traditional sc-RNA-seq and immunostaining methodologies?

[Authors]

We added an additional analysis to compare cell composition in vitro vs after transplantation (Extended Data Fig. 8a). In the text, we now say:

"Comparison of these 6-month transplanted SC-islets to those before transplantation revealed that transplanted SC-islets acquired gene expression and chromatin accessibility signatures that were more similar to their primary cell counterparts, including increased INS, MAFA, and IAPP expression in SC-β cells (Extended Data Fig. 8a, b and c, Supplementary Table 11). These improvements in cell identity were also reflected in the chromatin accessibility around the INS gene, which exhibited diminished peak signals in non-β cell populations (Fig. 5f). There were, however, no major shifts in composition, suggesting this was set by the end of the in vitro differentiation process and that trans-differentiation was likely not occurring (Extended Data Fig. 8a)."

By the end of the in vitro differentiation protocol, the composition has been locked since we have pushed them into terminal cell fates. Rather, it seems that the in vivo environment provides signals that reinforces the identities specified during in vitro differentiation, although it is currently unknown what those specific signals are. Our multiomics analysis, which includes chromatin information, highlights motif markers for each cell type and does provide more complete information about the state of the cells after transplantation. Specifically, key identity motif markers are retained in transplanted islets, while off-target motifs close overtime in vivo. The chromatin information provides us with new aspects for characterizing these cell identities.

[Reviewer]

3. Cluster proximity on the 2-dimensional plot does not mean lineage proximity. There is an ongoing debate on this issue and it would be important to rephrase or justify using computational explanation why the authors think that the 2-dimensional space reflects the closeness of the multidimensionality of the data. Trajectory analysis is more suited to answer such questions.

[Authors]

We agree that distances of clusters do not infer lineage proximity and removed the inaccurate language. We now use trajectory analysis to study this in Figure 2 and associated text.

[Reviewer]

References: - Format the reference list consistently as sometimes the manuscript gives the abbreviations for the journals and some time the full names in certain cases misspelled.
- Reference 7 and 48 is the same paper. Please check the reference list for further inconsistencies.

[Authors]

We have corrected this error.

[Reviewer]

- Reference the papers sorting for GP2+ cells at progenitor stage together with the 8-12 references for completion in the Introduction. That is reference 53 of the current manuscript and include the Cell Reports work from Henrik Semb's laboratory for completion: PMID: 28380361

[Authors]

We have corrected this error.

[Reviewer]

- Ackermann et al paper is missing from the reference and the discussion. It should be included in the manuscript as it was one of the first to use bulk ATAC and RNA-Seq to study human endocrine populations PMID: 26977395

[Authors]

We have corrected this oversight and added the reference.

[Reviewer]

Clarity and context: The paper is clearly written and conveys the main points of the work. A minor comment:
Page 11: Problem with sentence structure starting with Unlike SC-islets. Please re-phrase.

[Authors]

We have corrected this sentence.

Reviewer #2:

[Reviewers comments in black]

[Author comments in dark blue]

We thank the reviewer for the feedback.

[Reviewer]

Remarks to the Author:

Punn Augsornworawat et al used sc/sn-RNA seq combined with sc/sn-ATAC seq (10X) to study stem cell-derived islets and compared these with results using the same methods for human islets from 2 donors. The approach is interesting, but the study should use human islets from a larger number of human donors to make it a valuable resource and it is also difficult to follow the number of replicates for SC-islets and number of cells used for different analyses throughout the manuscript. Overall, a better description of their methods and analyses should be provided. What exact input was used for respective analysis? Currently, one needs to guess a lot. Specific limitations and comments are found below.

[Authors]

We have added an a more detailed description of the bioinformatics analysis in the Methods section: Processing and filtering of multiomics sequencing data; Dataset normalization, integration, and assay build; Analysis of multiomics datasets; Trajectory analysis sections. These sections now describe all relevant information, including used packages, functions, methods, databases, and parameters. Additionally, supplemental table 2 now includes parameters used for filtering cells.

The comments on human islet donors and replicate and cell numbers are addressed in detail later in this document in our response to reviewer 2.

[Reviewer]

1. I miss numbers and replicates throughout the result section and also in the legends (only included in a few places). It is difficult to find numbers used for each experimental part.

[Authors]

We now provide this information throughout the main text and figure legends, including cell number and replicates. Additionally, supplement table 2 now provides details on which datasets were used in each single-cell figure. We also now provide this in the "Statistics and reproducibility" section, saying :

"This study contains multiomic sequencing datasets of SC-islets from 3 independent differentiation batches (3 datasets), human islets from 4 donors (4 datasets), SC-islets

after 3 weeks, 4 weeks, 6 months, and 12 months into stage 6 with one 1 replicate each (5 datasets), CTCF CRISPRa differentiations of 1 replicate from each condition (Control and Dox-induced CTCF overexpression; 2 datasets), and ARID1BshRNA SC-islets of 1 replicate from each condition (Control and shARID1B knockdown; 2 datasets). Information on datasets used and sample details can be found in Supplemental Table 2.”

[Reviewer]

2. Page 6, result section: “Each of the identified populations had high expression of genes traditionally associated with their cell type (Fig. 1c).” This statement expects the reader to know what genes are traditionally expressed in different cell types?

[Authors]

We have clarified this in the text, now saying:

“Each of the identified populations had high expression of genes traditionally associated with their cell type (SC-β: INS, ERO1B, DLK1; SC-α: GCG, LOXL4, IRX2; SC-δ: SST, HHX, LEPR; SC-EC: TPH1, DDC, SLC18A1; Fig. 1c). ”

[Reviewer]

3. In each legend to both figures and sup tables please include what statistical method has been used and numbers compared (it is only done in a few cases). In particular, sup table 3 needs better explanation. How did the authors correct for multiple testing? The order of the sup tables needs to be corrected in the text. Also clarify abbreviations in tables.

[Authors]

We now include statistical methods in every figure legend. We also provide details in the "Statistics and reproducibility" section, including statistic tests and correcting for multiple testing, saying:

*“Statistical significance from the multiomic analyses was calculated using the Wilcoxon rank sum test for RNA expression, logistic regression for motif chromatin accessibility, and Bonferroni correction to account for multiple testing. For in vitro experiments, we performed unpaired or paired parametric t-tests (two-sided) and one-way ANOVA with Tukey’s multiple comparison testing to determine significance. All in vitro experiment data points presented are biological replicates and can be found in Supplementary Table 14. Significant values are marked based on p-values using ns > 0.05, * < 0.05, ** < 0.01, *** < 0.001, and **** < 0.0001.”*

In the table legend to supplementary table 3, we now say:

“This set of tables show gene expression upregulated in each cluster for cell type identification. These include datasets for: (3.1) SC-islets, (3.2) Primary human islets, (3.3) Transplanted SC-islets, and (3.4) CTCF CRISPRa SC-islets. Statistical significance was assessed by the Wilcoxon rank sum test.”

The order of the tables has been corrected, and the abbreviations have been clarified in the tables.

[Reviewer]

4. The authors use Monocle of data presented in sup table 5 for analysis but this is not described in the manuscript.

[Authors]

We added the “Trajectory analysis” section under Methods describing our use of Monocle, which now states:

“Trajectory analysis was conducted using SeuratWrappers and the Monocle3 package⁹⁶. We isolated specific populations (Enterochromaffin cells and β -cell) from the integrated multiomic dataset using "subset". A monocle compatible cds object file format was then generated using "as.cell_data_set" containing expression assays, including gene expression, promoter accessibility, and motif enrichment, and condition labels such as identified cell types. Pseudotime within the isolated dataset was computed and determined using the "cluster_cells" and "learn_graph" functions. Trajectories were reestablished by selecting the initial node pseudo-timepoint using "order_cells". Dynamic analysis of gene expression and motif enrichment were obtained by conducting differential expression feature analyses along the pseudotime trajectory. We used "graph_test" with parameter "neighbor_graph" set to "principal_graph". Subsequently, genes or motifs of high differential expression along the pseudotime were filtered by excluding features with low Moran's I scores ($morans_I > 0.05$). Expression values were Z-scored and plotted using "Heatmap" with K means parameter ($km = 2$). Pseudotime information from this analysis was incorporated into the original Seurat object file, transferring pseudotime labels on cells using "AddMetaData" for further trajectory related analyses.”

[Reviewer]

5. For data presented in sup table 6, how many cells were included in each SC-beta cell subpopulation. What exact comparison was done for the data in sup table 6, 4 different subgroups seem to be compared?

[Authors]

We added cell number information for the subpopulation/subgroup analysis in supplemental table 7 (previously called supplemental table 6). We clarified the exact comparisons performed in the table legend, which now says:

“This set of tables show differential gene expression and motifs accessibility analysis of a selected subpopulation compared to the rest of the other subpopulations in SC- β cells, SC-EC cells, and primary β cells. (7.1) SC- β subpopulation 1 (7.2) SC- β subpopulation 2 (7.3) SC- β subpopulation 3 (7.4) SC- β subpopulation 4 (7.5) SC-EC subpopulation 1 (7.6) SC-EC subpopulation 2 (7.7) SC-EC subpopulation 3 (7.8) SC-EC subpopulation 4 (7.9) Primary β subpopulation 1 (7.10) Primary β subpopulation 2 (7.11) Primary β subpopulation 3. (7.12) Composition information showing the number of cells in each subpopulation for SC-islets and primary islets. Statistical significance was assessed by the Wilcoxon rank sum test for RNA expression and logistic regression for motif chromatin accessibility. Avg, average; SC, stem cell derived; EC, enterochromaffin; HI, human islets.”

[Reviewer]

6. Human islets were provided by Prodo. In the methods on page 24, I miss important information about number of donors, including their BMI, use of pharmacotherapy, and days in culture. I found some information in Sup table 2.2 but this should be clearer in the paper.

[Authors]

We have clarified this in the "SC-islet and primary islet cell culture" section in the Methods, now saying:

“Donor details, including BMI, health status, age, sex, and race can be found in Supplemental Table 2.2. Our study consists of 4 total donors. Upon arrival, islets were transferred into wells of a 6-well plate on an orbital shaker at 100 RPM and maintained with 4mL per well of CMRL 1066 Supplemented media (Corning; 99-603-CV) with 10% heat inactivated FBS (Gibco; 26140-079). Primary human islets were submitted for sequencing within 2 days after arrival.”

We now provide details on human islet donors in Supplemental table 2.

[Reviewer]

Also, this study only includes pancreatic islets from 2 human donors, which is a major limitation since there is variability between human donors due to age, sex, therapy etc and are these 2 donors representative for the stem cell donors?

[Authors]

We have added two additional human islet donors, making up a total of 40,892 cells from four human islet donors. We have incorporated this into the primary islet datasets and redid the analysis in figure 3 and extended data figure 4. We now also include the number of cells and number of samples in every figure legend that involves single-cell analyses. These numbers are also in the main text.

Additionally, supplement table 2 provides details on which datasets were used in each single-cell figure. For clarity, we now mention this in the statistics and reproducibility section where we now state:

“Information on datasets used and sample details can be found in Supplemental Table 2.”

Despite variability in donor characteristics, our comparative analysis with SC-islets shows that the results are consistent across donors (Extended data fig. 5c-g). However, these donors were not intended to represent population variation nor potential stem cell donors. We address this limitation in the text, now saying:

“While the four islet donors analyzed here may not encompass all possible variability found across the general population, these data nevertheless provide a valuable resource to identify particular deficiencies in the chromatin and transcriptional landscape of SC-islet cell types.”

[Reviewer]

7. Throughout the method section, please include what cells type and species was used for each method. For example, in the “Multiomics sequencing analysis workflow” all of a sudden, the following statement occurs “In addition, mouse cells were removed by removing cells showing high....” And it is not very clear to the reader why the authors write about mice when one expects human cells, but it is not at all mentioned in this section. It is possible to understand further down, but overall a better method section is desirable.

[Authors]

We edited the methods section to clarify these several points. We moved the “Mouse transplantations and SC-islet cell retrieval” section up in appropriate order. We added the "Processing and filtering of multiomics sequencing data" section to expand on this.

We clarified the section referencing mouse cells, now saying:

“For transplanted SC-islets, excess non-desirable mice cells from host mice were further removed by detecting and filtering out cells expressing kidney marker TTC3616, which is

a gene shared across both mouse and human genome. Cells carrying high expression of TTC36 (> 0.0005 normalized counts) were excluded from the datasets. Additional details including exact inputs for filtering cells and final cell counts from each dataset can be found in Supplementary Table 2 and Supplementary Figure 1.”

[Reviewer]

8. In the real-time PCR section and sup table 1.7, it is unclear which endogenous control was used? Multiple once? Did the authors test if endogenous controls were regulated by the comparison they studied?

[Authors]

We now state the endogenous control in the Real-Time PCR section under Methods, saying:

“The housekeeping genes TBP and GUSB were both used for normalization.”

We selected these genes because they are commonly used in islet and diabetes publications. We used both genes together to prevent or account for potential changes. Here, we show the expression of our genes comparing the use of both or individual housekeeping genes and do not see meaningful expression changes from different controls.

[Reviewer]

9. The mouse transplantation experiments need further/better description throughout the manuscript (results, legends and methods), how many mice were used (3?). Sup table 2 gives some information that suggests 3 mice, and gives a cell number after filtering, but overall, it is difficult to follow numbers and methods.

[Authors]

We have expanded our “Mouse transplantations and SC-islet cell retrieval” section under the Methods to provide greater detail. In it, we now say:

“A total of 9 mice were used to produce 3 samples. Each sample consisted of pooling 3 transplanted kidneys (1 kidney from each mouse) to achieve sufficient cell numbers required for sequencing.”

We now provide the number of cells used in every figure legend with single-cell analysis, including those from transplanted experiments. These numbers are also in the main text of the results section and in supplement table 2.

Reviewer #3:

[Reviewers comments in black]

[Author comments in dark blue]

We thank the reviewer for the feedback.

[Reviewer]

Remarks to the Author:

Augsornworawat et al. performed multiomic assessments on SC- β cells using integrated sc-ATACseq and sc-RNAseq. They compared the chromatin accessibility between SC- β and SC-EC, SC- β and primary β , and long-term cultured cell populations and their post-transplantation counterparts. This approach improved the resolution of single cell transcriptome and revealed new signature genes in each population. In addition, they provided two examples, with signature genes being modulated. Overexpression of CTCF, a signature of SC-EC, during stage 5 leads to drastic reductions in the expression of β -cell identity genes. Knockdown of ARID1B, whose expression is lower in primary beta and long-term culture SC- β , leads to increased SC- β cell yields.

This is the first sc-ATACseq of SC-beta cells. My only criticism is that the conclusions reached in this study are mostly expected. Nevertheless, I appreciate the thoroughness of the investigation. Data revealed in this study provide a valuable resource to identify deficiencies in the chromatin and transcriptional landscape of SC-islet cell types, which may contribute to further improving the current ES- β differentiation protocol.

Specific

1, Figure 5, the authors compared the 6-month transplanted SC-islets to those before transplantation, and the data suggest that transplanted SC-islets were more similar to their primary cell counterparts. How about the comparison between 4-week transplanted SC-islets to those before transplantation? As 4-week in vitro at S6 is the prime time for SC- β cells, this may have different results. The authors stated on p. 16 that, in SC- β cells, expression of many β -cell transcription factors and their associated motifs were upregulated at week 4 of in vitro culture, but most of these motifs and many transcripts decreased in the long term.

Adding a 4-week time point will also illustrate the changes in transcriptomic/chromatin accessibility during SC-islet maturation in vivo.

[Authors]

To address this comment from the reviewer, we transplanted nine mice and attempted to generate in vivo data at the desired early time point. However, we unfortunately experienced technical challenges, described below.

The approach we developed for getting good quality human cells from the mice after several months did not appear to work well at this early time point each of the nine times

that we attempted this. In particular, dispersion of the graft from the mouse kidney was more challenging than we experienced at the later timepoints. Nevertheless, we proceeded with attempting to process and sequence the grafts from these nine mice. Unfortunately, the analysis pipeline of sequencing and alignment by the 10X Cell Ranger software failed for samples representing 6 out of the 9 mice. We worked directly with the manufacturer, 10X Genomics, on this issue but they were unable to resolve it. 10x Genomics attributed this to the poor quality of the cells. Proceeding with data from the three remaining mice, there ended up being only 59 SC-beta cells total. Below represents our analysis with these cells.

We compared SC-beta cells from week 4 in vivo with week 4 vitro and pre-transplanted in vitro cells. Our heatmap analysis (B) shows that the week 4 in vivo beta cells have decreased transcript for multiple beta cell genes, such as *INS*, and *ISL1*. *IAPP* is decreased when compared to week 4 in vivo, but upregulated when compared to pre-transplantation. We have also performed differential analyses comparing the different conditions and highlighted, in volcano plots (C and D), genes and chromatin motif accessibilities upregulated in each comparison and conditions. In summary, there appears to be differences in how SC-beta cells respond to in vitro and in vivo environment in the short term, however the exact nature of this is unclear based on this current dataset.

We have decided not to include this analysis in the manuscript. We believe that the technical difficulties experienced has greatly impacted the robustness of this analysis such that it is unsuitable for inclusion in the manuscript. We suspect that part of the difficulty is that additional time is necessary for the transplanted cells to properly engraft and overcome the acute stresses of transplantation.

Reviewer Figure 1. Multiomics sequencing comparisons of SC-islets from 2 week in vitro, 4 week in vitro and 4 week in vivo. A) Image of isolated kidney with transplanted SC-

islets. B) Heatmap comparing gene expression, promoter accessibility, or motif chromatin accessibility in β -cells. C-D) Differential gene expression analysis (top) and motif chromatin accessibility analysis (bottom) of SC- β cells comparing C) 4 week in vivo vs. 4 week in vitro and D) 4 week in vivo vs. 2 week in vitro.

[Reviewer]

2, It is unclear how S6 stage cells can be maintained in culture for 12 months. Whether they have been passaged? How often was the passage? The information was not found in the text or methods.

[Authors]

We have expanded our methods and now include details on long term culture where we now state:

“These clusters were maintained by aspirating and replacing 3 mL of ESFM every 2 - 3 days. SC-islets in long-term culture were similarly maintained with ESFM for up to 1 year without passaging.”

The cells do not meaningfully proliferate and did not require passaging.

Decision Letter, first revision:

Our ref: NCB-RS49518A

6th March 2023

Dear Jeff,

Thank you for submitting your revised manuscript "Defining the chromatin and transcriptional landscape of stem cell-derived islets" (NCB-RS49518A) and for your additional responses to the last referee comments. As you know, your manuscript has now been seen by the original referees and their comments are below. The reviewers find that the paper has improved in revision, and therefore we'll be happy in principle to publish it in Nature Cell Biology, pending minor revisions to satisfy the referees' final requests and to comply with our editorial and formatting guidelines.

We are now performing detailed checks on your paper and will send you a checklist detailing our editorial and formatting requirements in about one to two weeks. Please do not upload the final materials and make any revisions until you receive this additional information from us. Please note

that, after you receive the checklist and perform the revision per our requests, we expect you to also revise your manuscript per the last referee comments like you have suggested to do in your latest response to referee comments. We also expect you to upload a point-by-point response to the remaining referee comments (the one you provided to me via e-mail will do).

Thank you again for your interest in Nature Cell Biology. Please do not hesitate to contact me if you have any questions.

Best wishes,
Stelios

Stylianos Lefkopoulos, PhD
He/him/his
Associate Editor
Nature Cell Biology
Springer Nature
Heidelberger Platz 3, 14197 Berlin, Germany

E-mail: stylianos.lefkopoulos@springernature.com
Twitter: @s_lefkopoulos

Reviewer #1 (Remarks to the Author):

The authors have sufficiently addressed my questions regarding their work and I think the updated manuscript will be an important resource for the islet biology community.

I have a couple minor remaining comments:

- for completion add this reference (PMID: 31806625) when discussing about serotonin signaling and beta-cells
- perhaps I missed this during the first revision round but in Figure 3b it seems there is an acinar cell population budding out of the a-cell population. Is this accurate and if so do they have distinct signature to the main acinar cluster? No need to expand on that on the manuscript just need to make sure this is not an accidental wrong graph inserted in the panel

Reviewer #2 (Remarks to the Author):

Overall, the authors have responded to most of my comments and particularly they have added human islet data from 2 more donors. Nevertheless, a few comments/questions remain.

1. I cannot find any reference to Supplementary Table 4 in the main text of the ms and after searching not in the supplementary either?
2. Regarding Supplementary Table 2, please include cell count before filtering in S2.4 and a better explanation to the cell filtering parameters in S2.3 (both in the legend to the table and a better explanation in the table).

3. TBP does not seem to be a very good housekeeping gene for islets/beta-cells (DOI: 10.1074/jbc.274.30.21095).

4. As a recourse paper and for QC, I also wonder if the Tape station figures for all sn-ATAC experiments/libraries, showing the nucleosome pattern/sizes for the different experiments, can be added into the supplementary?

Reviewer #3 (Remarks to the Author):

The authors had managed to address the concern raised. I support its publication as a resource.

Decision Letter, final checks:

Our ref: NCB-RS49518A

24th March 2023

Dear Dr. Millman,

Thank you for your patience as we've prepared the guidelines for final submission of your Nature Cell Biology manuscript, "Defining the chromatin and transcriptional landscape of stem cell-derived islets" (NCB-RS49518A). Please carefully follow the step-by-step instructions provided in the attached file, and add a response in each row of the table to indicate the changes that you have made. Please also check and comment on any additional marked-up edits we have proposed within the text. Ensuring that each point is addressed will help to ensure that your revised manuscript can be swiftly handed over to our production team.

In recognition of the time and expertise our reviewers provide to Nature Cell Biology's editorial process, we would like to formally acknowledge their contribution to the external peer review of your manuscript entitled "Defining the chromatin and transcriptional landscape of stem cell-derived islets". For those reviewers who give their assent, we will be publishing their names alongside the published article.

Nature Cell Biology offers a Transparent Peer Review option for new original research manuscripts submitted after December 1st, 2019. As part of this initiative, we encourage our authors to support increased transparency into the peer review process by agreeing to have the reviewer comments, author rebuttal letters, and editorial decision letters published as a Supplementary item. When you submit your final files please clearly state in your cover letter whether or not you would like to participate in this initiative. Please note that failure to state your preference will result in delays in accepting your manuscript for publication.

Cover suggestions

As you prepare your final files we encourage you to consider whether you have any images or illustrations that may be appropriate for use on the cover of Nature Cell Biology.

Nature Cell Biology has now transitioned to a unified Rights Collection system which will allow our Author Services team to quickly and easily collect the rights and permissions required to publish your work. Approximately 10 days after your paper is formally accepted, you will receive an email in providing you with a link to complete the grant of rights. If your paper is eligible for Open Access, our Author Services team will also be in touch regarding any additional information that may be required to arrange payment for your article.

Please note that *Nature Cell Biology* is a Transformative Journal (TJ). Authors may publish their research with us through the traditional subscription access route or make their paper immediately open access through payment of an article-processing charge (APC). Authors will not be required to make a final decision about access to their article until it has been accepted. Find out more about Transformative Journals

Please use the following link for uploading these materials:
[Redacted]

Best regards,

Jonathon Comfort
Staff
Nature Cell Biology

On behalf of

Stylianos Lefkopoulos, PhD
He/him/his
Associate Editor
Nature Cell Biology
Springer Nature
Heidelberger Platz 3, 14197 Berlin, Germany

E-mail: stylianos.lefkopoulos@springernature.com
Twitter: @s_lefkopoulos

Reviewer #1:

Remarks to the Author:

The authors have sufficiently addressed my questions regarding their work and I think the updated manuscript will be an important resource for the islet biology community.

I have a couple minor remaining comments:

- for completion add this reference (PMID: 31806625) when discussing about serotonin signaling and beta-cells
- perhaps I missed this during the first revision round but in Figure 3b it seems there is an acinar cell population budding out of the a-cell population. Is this accurate and if so do they have distinct signature to the main acinar cluster? No need to expand on that on the manuscript just need to make sure this is not an accidental wrong graph inserted in the panel

Reviewer #2:

Remarks to the Author:

Overall, the authors have responded to most of my comments and particularly they have added human islet data from 2 more donors. Nevertheless, a few comments/questions remain.

1. I cannot find any reference to Supplementary Table 4 in the main text of the ms and after searching not in the supplementary either?
2. Regarding Supplementary Table 2, please include cell count before filtering in S2.4 and a better explanation to the cell filtering parameters in S2.3 (both in the legend to the table and a better explanation in the table).
3. TBP does not seem to be a very good housekeeping gene for islets/beta-cells (DOI: 10.1074/jbc.274.30.21095).
4. As a recourse paper and for QC, I also wonder if the Tape station figures for all sn-ATAC experiments/libraries, showing the nucleosome pattern/sizes for the different experiments, can be added into the supplementary?

Reviewer #3:

Remarks to the Author:

The authors had managed to address the concern raised. I support its publication as a resource.

Author Rebuttal, first revision:

[Reviewer comments in black]

[Authors comments in dark blue]

We thank the authors for the continued feedback.

Reviewer #1

The authors have sufficiently addressed my questions regarding their work and I think the updated manuscript will be an important resource for the islet biology community.

I have a couple minor remaining comments:

[Authors] Thank you

- for completion add this reference (PMID: 31806625) when discussing about serotonin signaling and beta-cells

[Authors] We have added this reference (#43 in our revised text).

- perhaps I missed this during the first revision round but in Figure 3b it seems there is an acinar cell population budding out of the a-cell population. Is this accurate and if so do they have distinct signature to the main acinar cluster? No need to expand on that on the manuscript just need to make sure this is not an accidental wrong graph inserted in the panel

[Authors] These cells are indeed part of the same acinar population. Populations can be split when visualized using UMAPs due to the multi-dimensional reduction of data onto two-dimensional figures. To avoid confusion for the reader, we adjusted the clustering parameters to avoid splitting the acinar population on the UMAP. Below is how Fig. 3a-b looks:

Reviewer #2

Overall, the authors have responded to most of my comments and particularly they have added human islet data from 2 more donors. Nevertheless, a few comments/questions remain.

[Authors] Thank you

1. I cannot find any reference to Supplementary Table 4 in the main text of the ms and after searching not in the supplementary either?

[Authors] Supplementary Table 4 is referenced in the Results section, “Multiomic SC-islet analysis improves cell identity resolution” subsection, in the sentence that now says: “Analysis of gene expression and chromatin accessibility both individually and in combination enabled us to identify specific islet cell types, including SC- β cells (Fig. 1b and c, Extended Data Fig. 1c and d, Supplementary Table 3 and 4; 29526 cells from 3 independent differentiations).” This was in the main text on page 5 line 111 and page 12 line 271 of the previous submission.

2. Regarding Supplementary Table 2, please include cell count before filtering in S2.4 and a better explanation to the cell filtering parameters in S2.3 (both in the legend to the table and a better explanation in the table).

[Authors] We updated the supplementary table to include cell counts before filtering and additionally provide clarification in the legends.

In the supplemental table legend, we now state:

“(2.3) Cell filtering parameters used to exclude low quality cells and mouse cells from each data set. Cells with exceedingly low and high RNA and ATAC fragment counts, and high nucleosome signal were excluded to minimize potential doublets and dead cells. For transplanted samples, we have excluded host cells by filtering out cells with high expression of kidney marker TTC36.”

3. TBP does not seem to be a very good housekeeping gene for islets/beta-cells (DOI: 10.1074/jbc.274.30.21095).

[Authors] Our gene expression data is not meaningfully changed using or not using TBP as a housekeeping gene. Below we show real-time PCR data using TBP alone, GUSB alone, or both together showing not meaningful difference:

Our reported real-time PCR experiments use two housekeeping genes, TBP and GUSB, to normalize data. The use of two minimizes the likelihood that variation in the housekeeping gene meaningfully changes the results. TBP is a well-accepted housekeeping gene in the field commonly used by many expert groups (D’Amour/ViaCyte, Huisin, Millman, Bonner-Weir, Egli). See PMIDs 17053790, 23251699, 32094658, 26207953, 24227685. It is unclear how the article cited by the reviewer supports their assertion that TBP is not a good housekeeping gene, as the data appears unrelated.

4. As a recourse paper and for QC, I also wonder if the Tape station figures for all sn-ATAC experiments/libraries, showing the nucleosome pattern/sizes for the different experiments, can be added into the supplementary?

[Authors] We have added the requested information into Supplementary Figure 1.

Reviewer #3

The authors had managed to address the concern raised. I support its publication as a resource.

[Authors] Thank you

Final Decision Letter:

Dear Jeff,

I am pleased to inform you that your manuscript, "Single nuclei multiomics of human stem cell-derived islets identifies deficiencies in lineage specification", has now been accepted for publication in Nature Cell Biology. Congratulations to you and the whole team!

Once your paper has been scheduled for online publication, the Nature press office will be in touch to confirm the details. An online order form for reprints of your paper is available at <https://www.nature.com/reprints/author-reprints.html>. All co-authors, authors' institutions and

authors' funding agencies can order reprints using the form appropriate to their geographical region.

Please note that *Nature Cell Biology* is a Transformative Journal (TJ). Authors may publish their research with us through the traditional subscription access route or make their paper immediately open access through payment of an article-processing charge (APC). Authors will not be required to make a final decision about access to their article until it has been accepted. Find out more about Transformative Journals

If you have not already done so, we strongly recommend that you upload the step-by-step protocols used in this manuscript to the Protocol Exchange (www.nature.com/protocolexchange), an open online resource established by Nature Protocols that allows researchers to share their detailed experimental know-how. All uploaded protocols are made freely available, assigned DOIs for ease of citation and are fully searchable through nature.com. Protocols and Nature Portfolio journal papers in which they are used can be linked to one another, and this link is clearly and prominently visible in the online versions of both papers. Authors who performed the specific experiments can act as primary authors for the Protocol as they will be best placed to share the methodology details, but the Corresponding Author of the present research paper should be included as one of the authors. By uploading your Protocols to Protocol Exchange, you are enabling researchers to more readily reproduce or adapt the methodology you use, as well as increasing the visibility of your protocols and papers. You can also establish a dedicated page to collect your lab Protocols. Further information can be found at www.nature.com/protocolexchange/about

With kind regards,
Stelios

Stylios Lefkopoulos, PhD
He/him/his
Associate Editor
Nature Cell Biology
Springer Nature
Heidelberger Platz 3, 14197 Berlin, Germany

E-mail: stylios.lefkopoulos@springernature.com
Twitter: @s_lefkopoulos
